# Impaired adaptation of learning to contingency volatility in internalizing psychopathology

Christopher Gagne[1,2], Ondrej Zika[3], Peter Dayan[2,4], Sonia J Bishop[1,5,6]*

[1]Department of Psychology, UC Berkeley, Berkeley, United States; [2]Max Planck Institute for Biological Cybernetics, Tübingen, Germany; [3]Max Planck Institute for Human Development, Berlin, Germany; [4]University of Tübingen, Tübingen, Germany; [5]Wellcome Centre for Integrative Neuroimaging, University of Oxford, FMRIB, John Radcliffe Hospital, Oxford, United Kingdom; [6]Helen Wills Neuroscience Institute, UC Berkeley, Berkeley, United States

**Abstract** Using a contingency volatility manipulation, we tested the hypothesis that difficulty adapting probabilistic decision-making to second-order uncertainty might reflect a core deficit that cuts across anxiety and depression and holds regardless of whether outcomes are aversive or involve reward gain or loss. We used bifactor modeling of internalizing symptoms to separate symptom variance common to both anxiety and depression from that unique to each. Across two experiments, we modeled performance on a probabilistic decision-making under volatility task using a hierarchical Bayesian framework. Elevated scores on the common internalizing factor, with high loadings across anxiety and depression items, were linked to impoverished adjustment of learning to volatility regardless of whether outcomes involved reward gain, electrical stimulation, or reward loss. In particular, high common factor scores were linked to dampened learning following better-than-expected outcomes in volatile environments. No such relationships were observed for anxiety- or depression-specific symptom factors.

*For correspondence: sbishop@berkeley.edu

## Introduction

Many of the situations we encounter in daily life are characterized by uncertainty. From small choices to large lifetime decisions, we rarely can know for sure the consequences that will stem from our actions. The uncertainty that pervades our daily decision-making is a source of greater distress, and even dysfunction, for some individuals than for others. In recognition of this, intolerance of uncertainty has been proposed as a core feature of generalized anxiety disorder (*Freeston et al., 1994*; *Dugas et al., 1998*; *Dugas et al., 2001*) and, more recently, a transdiagnostic marker of internalizing psychopathology more broadly (*Gentes and Ruscio, 2011*; *Carleton et al., 2012*; *Boswell et al., 2013*). However, these accounts are largely based on patients' self-reported behavioral and emotional responses to exposure to uncertainty rather than experimental and computational investigations of the cognitive processes involved in choice under uncertainty. We address this in the current study. Here, we test the possibility that a deficit in adapting decision-making to uncertainty might be linked to internalizing psychopathology, in general, and observed across both aversive and reward-based learning.

Within the computational literature, advances have been made in formalizing different sources of uncertainty and their effects on decision making (*Yu and Dayan, 2005*; *Behrens et al., 2007*; *Nassar et al., 2012*; *Payzan-LeNestour et al., 2013*). One source of uncertainty is noise in the relationship between actions and outcomes, for instance when an action only leads to a given outcome on a proportion of the occasions it is performed. A second source of uncertainty is non-stationarity,

or volatility, in the underlying causal structure; for example, when action-outcome contingencies switch and an action that was primarily associated with a given outcome becomes predominantly associated with another. Such volatility leads to uncertainty around the estimate of outcome probability; this can be conceived of as second-order uncertainty (*Bach et al., 2011*). Crucially, the level of second-order uncertainty determines the optimal response to unexpected outcomes. When contingencies are volatile, and hence second-order uncertainty is high, participants should adjust their probability estimates more rapidly than when contingencies are stable and second-order uncertainty is low. *Behrens et al., 2007* reported that healthy adult participants are indeed able to adjust their rate of updating, that is, their learning rate, to match contingency volatility in this manner.

Failure to adapt correctly to the source of uncertainty present in a given situation may result in inaccurate predictions and suboptimal decisions. This might well be a source of distress and dysfunction in everyday life. In a prior study, we examined the relationship between trait anxiety and adaptation of decision-making to contingency volatility (*Browning et al., 2015*). Modeling participants' performance on an aversive version of Behrens and colleagues' probabilistic decision-making task, using electrical stimulation as outcomes, we found that trait anxiety was associated with reduced adaptation of learning rate to volatility. This finding raises the possibility that poor adjustment of probabilistic learning to volatility might reflect an inability to differentially respond to different forms of uncertainty and that this, in turn, might be an important marker of psychopathology.

Critically, this initial work leaves open the question of whether impaired adaptation of learning rate to volatility is specific to anxiety or a marker of internalizing psychopathology more broadly. Scores on the Spielberger Trait Anxiety Inventory (STAI, *Spielberger et al., 1983*), the measure of anxiety used in *Browning et al., 2015*, are typically elevated across patients with both anxiety and depressive disorders and correlate highly with scores on measures of depression such as the Beck Depression Inventory (BDI, *Beck et al., 1961*). Establishing whether impairment in adapting decision-making to volatility is specific to anxiety, or more broadly linked to internalizing psychopathology, requires a way of partitioning symptom variance into components that disentangle variance common to both anxiety and depression from that specific to anxiety or to depression. Bifactor analysis provides a principled method for achieving this goal. It has been used extensively to study the structure of internalizing symptomology (*Clark et al., 1994*; *Steer et al., 1995*; *Zinbarg and Barlow, 1996*; *Steer et al., 1999*; *Simms et al., 2008*; *Steer et al., 2008*; *Brodbeck et al., 2011*) and has consistently revealed a substantial amount of shared variance, often termed 'general distress' or 'negative affect' (*Clark and Watson, 1991*; *Clark et al., 1994*). In addition, separate specific factors for depression and anxiety are consistently observed, with the depression-specific factor tapping symptoms of anhedonia (*Clark et al., 1994*; *Steer et al., 1999*; *Steer et al., 2008*) and anxiety-specific factors tapping symptoms of anxious arousal (*Clark et al., 1994*; *Steer et al., 1999*; *Steer et al., 2008*) and worry (*Brodbeck et al., 2011*). Although bifactor modeling of internalizing symptoms is well established, it has not, to date, been used to inform studies of anxiety- and depression-related deficits in decision-making. Using bifactor analysis to estimate scores for each participant on latent dimensions of internalizing symptoms, we can investigate whether impoverished adjustment of learning rate to volatility is primarily linked to general factor scores (i.e. to symptom variance common to both anxiety and depression) or to anxiety-specific or depression-specific factor scores.

A second question of specificity pertains to the breadth of impairment in adjusting learning rate to volatility. There is evidence that the neural substrate, and affective consequences, of learning from unexpected outcomes may vary both with domain (reward versus punishment; *Boureau and Dayan, 2011*) and with whether outcomes are better or worse than expected (*Frank et al., 2007*; *Cox et al., 2015*; *Eldar et al., 2016*; *Palminteri and Pessiglione, 2017*). If impoverished adaptation of decision-making to volatility is potentially a core feature of internalizing psychopathology, it is important to establish if such impairment is, or is not, observed regardless of whether choice involves potential aversive outcomes (e.g. shock), reward gain or reward loss and both when outcomes exceed and fall short of our expectations.

Our aims were hence as follows. First, to use bifactor analysis of item-level responses to measures of internalizing symptoms to determine whether impaired adjustment of learning rate to volatility is linked to symptom variance common to both anxiety and depression (i.e. to general factor scores) or to symptom variance specific to anxiety or to depression. Second, to determine whether any such impairment is domain general (i.e. observed both when actions involves the pursuit of rewarding

outcomes and the avoidance of punishing outcomes) or domain specific. And third, to determine whether this impairment is observed equally for learning from both positive and negative prediction errors (i.e. better-than-expected and worse-than-expected outcomes) or is differentially linked to one or the other.

## Results

### Outline

We addressed these aims through two experiments. The first experiment was conducted in our on-site testing facilities. The participant sample comprised patients diagnosed with major depressive disorder (MDD) or generalized anxiety disorder (GAD), healthy control participants screened to be free of any psychiatric diagnoses, and a community sample with naturally varying levels of symptoms (for further details see Materials and methods: Self-Report Measures). Participants completed two versions of the probabilistic decision-making under volatility task (*Behrens et al., 2007*; *Browning et al., 2015*). In one version, they chose between two shapes that were associated with potential receipt of electrical stimulation of varying magnitude. In the other version, the outcomes involved possible receipt of financial rewards of varying magnitude. In each case, the probability that a given shape would result in receipt of shock or reward had to be learned across trials. Action-outcome contingencies were stable in one period of the task and volatile in the other (see Materials and methods).

The second experiment was conducted online using Amazon's Mechanical Turk platform. Here, we used reward gain and reward loss versions of the probabilistic decision-making under volatility task. As detailed further below, we used the same bifactor decomposition of internalizing symptoms and the same computational model of task performance across both experiments and all three versions of the volatility task. A hierarchical Bayesian framework was used for model estimation with participants' scores on the latent factors of internalizing symptoms entered as predictors of behavioral model parameter values.

### Experiment 1

#### Estimating latent factors of internalizing symptoms

Eighty-eight participants (51 females, mean age = 27 ± 8 years) took part in experiment 1. This was conducted at the Wellcome Centre for Integrative Neuroimaging (WIN) at the John Radcliffe Hospital with ethical approval obtained from the Oxford Central University Research Ethics Committee (CUREC). Twenty participants had a primary diagnosis of major depressive disorder (MDD), 12 participants had a primary diagnosis of generalized anxiety disorder (GAD), 26 'healthy control' participants of approximately the same age and sex ratio were screened to ensure they were free of any psychiatric diagnosis, and 30 participants were members of the local community with a natural range of internalizing symptoms (see *Appendix 1—table 1* for participant details). Participants who met criteria for any psychiatric diagnoses apart from anxiety or depressive disorders were excluded, as were participants with neurological conditions and those currently taking psychoactive medications or recreational drugs (see Materials and methods). Data from two control participants was excluded; one was a result of equipment failure; debriefing indicated the second misunderstood the task. This left 86 participants in total.

Participants completed a battery of standardized questionnaires comprising the Spielberger State-Trait Anxiety Inventory (STAI form Y; *Spielberger et al., 1983*), the Beck Depression Inventory (BDI; *Beck et al., 1961*), the Mood and Anxiety Symptoms Questionnaire (MASQ; *Clark and Watson, 1995*; *Watson and Clark, 1991*), the Penn State Worry Questionnaire (*Meyer et al., 1990*), the Center for Epidemiologic Studies Depression Scale (CESD; *Radloff, 1977*), and the 90-item Eysenck Personality Questionnaire (EPQ; *Eysenck and Eysenck, 1975*). These questionnaires were selected to measure a range of depressive and anxiety symptoms (e.g. anhedonia, negative mood, negative cognitive biases, worry, somatic symptoms) and to assess trait negative affect more broadly (via inclusion of the EPQ Neuroticism subscale; items from other subscales of the EPQ were not included in the bifactor analysis described below).

We sought to separate symptom variance common to both anxiety and depression from that specific to depression or to anxiety. Bifactor modeling of item level responses provides a simple

approach to achieve this aim, as demonstrated previously within the internalizing literature (*Clark et al., 1994*; *Steer et al., 1995*; *Steer et al., 1999*; *Simms et al., 2008*; *Brodbeck et al., 2011*). Bifactor models decompose the item-level covariance matrix into a general factor and two or more specific factors. Here, we specified a model with one general and two specific factors. This decision drew on our theoretical aim, namely to separate symptom variance common to both anxiety and depression from that specific to depression and that specific to anxiety, and was informed by prior tripartite models of internalizing psychopathology (e.g. *Clark and Watson, 1991*). It was also supported by the results of eigenvalue decomposition of the covariance matrix. Only the first three eigenvalues were reliably distinguishable from noise—this was determined by comparison of the eigenvalues in descending order against eigenvalues obtained from a random normal matrix of equivalent size (*Humphreys and Montanelli, 1975*; *Floyd and Widaman, 1995*), see *Figure 1—figure supplement 1* and Materials and methods for more details.

The Schmid-Leiman (SL) procedure was used to estimate the loadings of individual questionnaire items on each factor (*Schmid and Leiman, 1957*). This procedure performs oblique factor analysis followed by a higher order factor analysis on the lower order factor correlations to extract a general factor. All three factors are forced to be orthogonal to one another, which allows for easier interpretability. In line with previous findings (*Clark et al., 1994*; *Steer et al., 1995*; *Zinbarg and Barlow, 1996*; *Steer et al., 1999*; *Simms et al., 2008*; *Steer et al., 2008*; *Brodbeck et al., 2011*), the general factor had high loadings (>0.4) for multiple anxiety-related and depression-related items and moderately high loadings (>0.2) across almost all items. One specific factor had high loadings (>0.4) for questions related to anhedonia and depressed mood. The other specific factor had high loadings (>0.4) for questions related to worry and anxiety.

We validated this factor structure by conducting a confirmatory bifactor analysis on item-level responses to the same set of questionnaires completed by an independent online sample (n = 199). Participants were students at UC Berkeley (120 females, mean age = 20 ± 4). This group was fairly distinct from our first sample, being more homogenous in age and educational status and less homogenous in ethnicity, and not including individuals recruited to meet diagnosis for either GAD or MDD. Evaluating the fit of the factor structure obtained from experiment 1 in this second dataset is a strong test of its generalizability. In the confirmatory bifactor analysis, we used diagonally weighted least squares estimation and constrained the factor structure so that items were only allowed to load on a factor for which they had a loading of >0.2 in experiment 1 (see Materials and methods: Exploratory Bifactor Analysis). This constrained model showed a good fit to the data from this new participant sample, comparative fit index (CFI) = 0.962.

As a convergent analysis, we conducted an unconstrained (i.e. exploratory, not confirmatory) bifactor analysis in this second participant sample to see if a similar factor structure would emerge to that obtained in experiment 1. We again specified one common and two specific factors. The factor loadings obtained were highly congruent with the factor loadings obtained from the bifactor analysis in experiment 1 (cosine-similarity was 0.96 for the general factor loadings, 0.81 for the depression-specific factor loadings, and 0.77 for the anxiety-specific factor loadings). Congruence in factor loadings was assessed after matching the two specific factors according to the similarity of their loading content.

Factor loadings from experiment 1 were used to calculate factor scores for all participants from experiment 1 (n = 86) and the confirmatory factor analysis sample (n = 199). The resultant scores are plotted in *Figure 1a–b*. As an additional check of construct validity, participants' scores on these factors were correlated with summary (scale or subscale) scores for the standardized questionnaires administered, *Figure 1c*. The questionnaires to which items belonged were not specified during the fitting of the bifactor model, hence these summary scores provide an independent measure of the construct validity of the latent factors extracted from the bifactor analysis. As can be seen in *Figure 1c*, participants' scores on the general factor correlated strongly (r > 0.60) with summary scores for all the questionnaire measures, indicating that the general factor is indeed tapping variance linked to both anxiety and depressive symptoms. Scores on the depression-specific factor correlated strongly with scores for the MASQ anhedonia subscale (MASQ-AD; r = 0.72) and the STAI depression subscale (STAIdep; r = 0.53), and scores on the anxiety-specific factor correlated strongly with scores for the Penn State Worry Questionnaire (PSWQ; r = 0.76). This indicates that the two specific factors extracted from the bifactor analysis do indeed capture anxiety- and depression-related symptoms, respectively, as intended, and that these factors explain variance above that

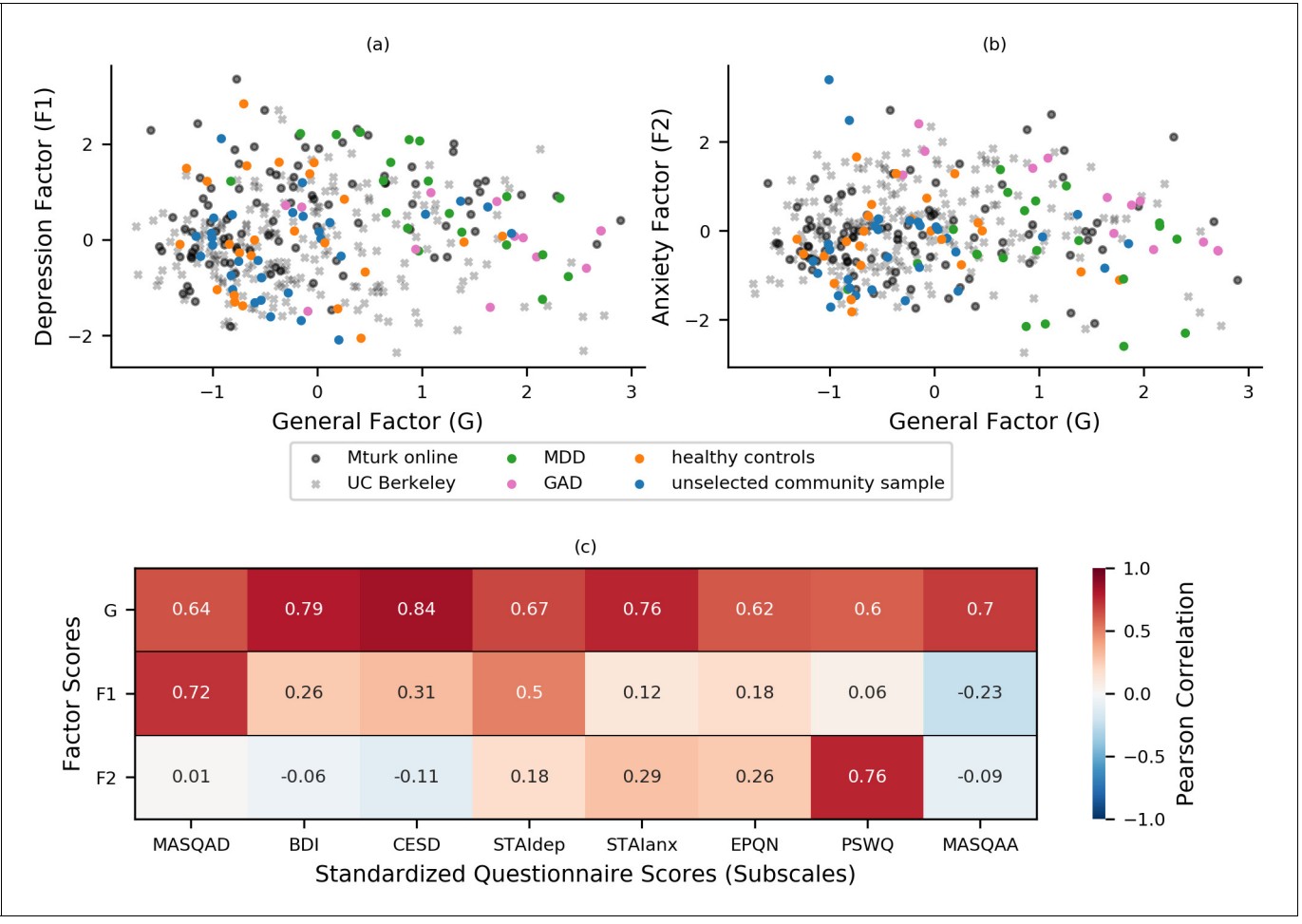

**Figure 1.** Bifactor analysis of internalizing symptoms. (**a-b**) Bifactor analysis of item-level scores from the STAI, BDI, MASQ, PSWQ, CESD, and EPQ-N (128 items in total) revealed a general 'negative affect' factor (xaxis) and two specific factors: one depression-specific (left panel, y-axis) and one anxiety-specific (right panel, y-axis). The initial bifactor analysis was conducted in a sample (n = 86) comprising participants diagnosed with MDD, participants diagnosed with GAD, healthy control participants and unselected community participants. The factor solution showed a good fit in a separate sample of participants (n = 199) recruited and tested online through UC Berkeley's participant pool (x). Item loadings on a sub-set of questionnaires were used to calculate factor scores for a third set of participants recruited and tested online through Amazon's Mechanical Turk (n = 147), see Experiment 2. It can be seen that both online samples show a good range of scores across the general and two specific factors that encompass the scores shown by patients with GAD and MDD. (**c**) Factor scores were correlated with summary scores for questionnaire scales and subscales to assess the construct validity of the latent factors. This was conducted using a combined dataset comprising data from both the exploratory (n = 86) and confirmatory (n = 199) factor analyses. Scores on the general factor correlated highly with all questionnaire summary scores, scores on the depression-specific factor correlated highly with measures of depression, especially anhedonic depression, and scores on the anxiety-specific factor correlated particularly highly with scores for the PSWQ. MASQAD = Mood and Anxiety Symptoms Questionnaire (anhedonic depression subscale); BDI = Beck Depression Inventory; CESD = Center for Epidemiologic Studies Depression Scale; STAIdep = Spielberger State-Trait Anxiety Inventory (depression subscale); STAIanx = Spielberger State-Trait Anxiety Inventory (anxiety subscale); EQN-N = Eysenck Personality Questionnaire (Neuroticism subscale); PSWQ = Penn State Worry Questionnaire; MASQAA = Mood and Anxiety Symptoms Questionnaire (anxious arousal subscale); MDD = major depressive disorder; GAD = generalized anxiety disorder.

The online version of this article includes the following figure supplement(s) for figure 1:

**Figure supplement 1.** Scree plot for the eigenvalue decomposition of the covariance matrix of individual items from the battery of internalizing symptom measures.

**Figure supplement 2.** Correlation matrices for internalizing questionnaire scales and latent factors from the bifactor analysis.

explained by the general factor. The latter conclusion can be drawn since the specific factors are orthogonal to the general factor and therefore their correlations with scale and subscale scores reflect independently explained variance. This can be further demonstrated by regressing variance explained by scores on the general factor out of scale and subscale scores and then examining the

relationship between residual scores for each scale with scores on the two specific factors. As shown in *Figure 1—figure supplement 2*, after removing variance explained by general factor scores, nearly all the remaining variance in PSWQ scores could be captured by the anxiety-specific factor and nearly all the remaining variance in MASQ-AD scores could be captured by scores on the depression-specific factor.

As outlined earlier, we applied a bifactor model to item-level symptom responses as we sought to tease apart symptom variance common to anxiety and depression versus unique to anxiety or depression. With model selection, both the extent to which a given model can address the aim of the experiment and the fit of the given model to the data are important considerations. In addition to assessing the absolute fit of the bifactor solution in the confirmatory factor analysis (CFA) dataset, we can also consider its fit relative to that of alternate models. The bifactor model reported here showed a better fit to the novel (CFA) dataset than a 'flat' correlated two-factor model, a hierarchical three factor model with the higher order factor constrained to act via the lower level factors, and a unifactor model created by retaining only the general factor and neither of the specific factors (see Appendix 2: Additional Factor Analyses for further details). We note that none of these alternate models would enable us to separate symptom variance common to anxiety and depression versus unique to anxiety and depression, as desired.

## Computational modeling of task performance: measuring the impact of block type (volatile, stable), task version (reward, aversive), and relative outcome value (good, bad) on learning rate

Participants completed both reward gain and aversive versions of a probabilistic decision-making under volatility task (*Behrens et al., 2007*; *Browning et al., 2015*). Full task details are provided in *Figure 2* and in the Materials and methods. In short, participants were asked to choose between the same two shapes repeatedly across trials. On each trial, one of the two shapes resulted in reward receipt or shock receipt; the nature of the outcome depended on the version of the task. When making their choice, participants were instructed to consider both the magnitude of the reward or shock associated with each shape, which was shown to participants inside each shape and varied across trials, and the probability that each shape would result in reward or shock receipt. The outcome probability could be learned across trials, using the outcomes received. During the stable task period, the

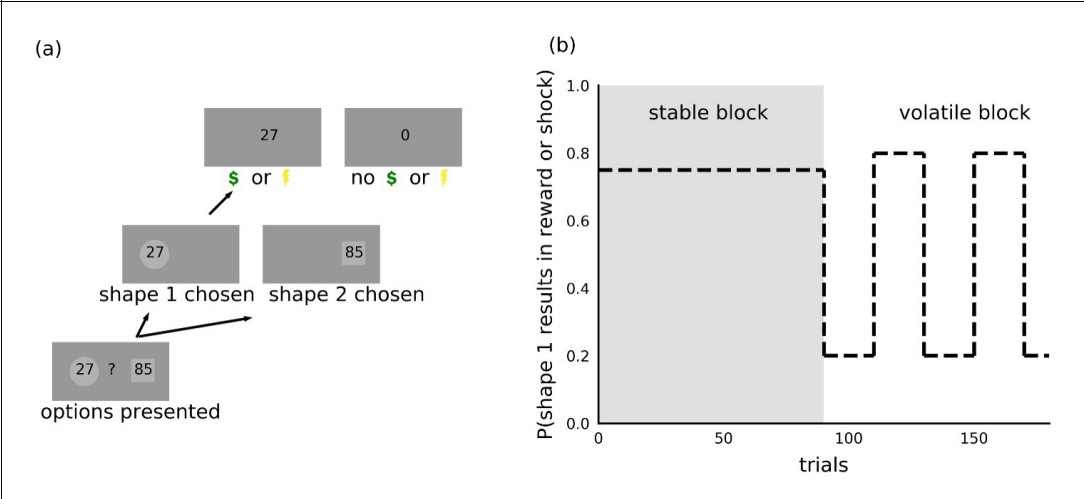

**Figure 2.** Task. (a) On each trial, participants chose between two shapes. One of the two shapes led to receipt of shock or reward on each trial, the nature of the outcome depending on the version of the task. The magnitude of the potential outcome was shown as a number inside each shape and corresponded to the size of the reward in the reward version of the task or intensity of the electric shock in the aversive version of the task. (b) Within each task, trials were organized into two 90-trial blocks. During the stable block, one shape had a 75% probability of resulting in reward or shock receipt; the other shape resulted in shock or reward receipt on the remaining trials. During the volatile block, the shape with the higher probability (80%) of resulting in shock or reward receipt switched every 20 trials. Participants were instructed to consider the magnitude of the potential outcome, shown as a number inside each shape, as well as the probability that the outcome would occur if the shape was chosen.

outcome probability was fixed such that one shape had a 75% probability of resulting in reward or shock receipt if chosen and the other 25%. During the volatile task period, the shape with the higher probability of shock or reward receipt switched every 20 trials (see Materials and methods for further details).

We fitted participants' choice behavior using alternate versions of simple reinforcement learning models. We focused on models that were parameterized in a sufficiently flexible manner to capture differences in behavior between experimental conditions (block type: volatile versus stable; task version: reward gain versus aversive) and differences in learning from better or worse than expected outcomes. We used a hierarchical Bayesian approach to estimate distributions over model parameters at an individual- and population-level with the latter capturing variation as a function of general, anxiety-specific, and depression-specific internalizing symptoms. Given our a priori interest in assessing the variance in choice behavior explained by the three internalizing factors, participants' scores for the three factors were included in the estimation procedure for all models. We compared the fits of different models using Pareto smoothed importance sampling to approximate leave-one-out cross-validation accuracy (PSIS-LOO; *Vehtari et al., 2017*). PSIS-LOO is a popular method with which to estimate out of sample prediction accuracy as it is less computationally expensive than evaluating exact leave-one-out or k-fold cross-validation accuracy (*Gelman et al., 2013*). We note that comparing models using WAIC (*Watanabe and Opper, 2010*), an alternative penalization-based criterion for hierarchical Bayesian models, resulted in identical model rankings for our dataset. We elaborate briefly on model comparison here. Full details are provided in the Materials and methods; a summary of model comparison results is also provided in *Appendix 3—table 1*.

The models considered were informed by prior work (*Lau and Glimcher, 2005*; *Behrens et al., 2007*; *Ito and Doya, 2009*; *Li and Daw, 2011*; *Berns and Bell, 2012*; *Akaishi et al., 2014*; *Browning et al., 2015*; *Donahue and Lee, 2015*; *Mkrtchian et al., 2017*; *Aylward et al., 2019*). Each of the models included a parameter to allow for individual differences in the weighting of outcome probabilities against outcome magnitudes, an inverse temperature parameter to allow for differences in how noisily participants made choices as a function of this weighted combination of probability and magnitude, and a learning rate parameter that captured the extent to which participants adjusted probability estimates given the unexpectedness of the previous trial's outcome. We parameterized dependence on experimental conditions by additive and interactive factors. As one example, the learning rate $\alpha$ was divided into a baseline learning rate ($\alpha_{baseline}$), a difference in learning rates between the volatile and stable blocks ($\alpha_{volatile-stable}$), a difference in learning rates between the reward gain and aversive versions of the volatility task ($\alpha_{reward-aversive}$), and the two-way interaction of those differences.

Parametrizing the effects of our experimental conditions in the manner described above allowed us to test how these effects varied as a function of between-participant differences in internalizing symptomatology. We parameterized the dependence on internalizing symptoms by adjusting the mean of the Bayesian hierarchical prior according to the general, depression-specific, and anxiety-specific factor scores of each participant, using population-level weights $\{\beta_g, \beta_d, \beta_a\}$, respectively. These weights differed for each parameter component (e.g. $\alpha_{volatile-stable}$), but we hide this specificity for notational ease. Including participants' scores on the three latent internalizing factors in the estimation procedure in this manner enables us to separate variance linked to internalizing symptoms from noise in participants' choices when estimating model parameters (see Materials and methods: Hierarchical Bayesian Estimation for more details).

We used a form of stage-wise model construction to investigate the manner in which participants integrated outcome probability and outcome magnitude (additive or multiplicative) and the extent to which task performance could be better captured by inclusion of additional model parameters. At each stage of the model construction process, we added or modified a particular component of the model and compared the enriched model to the best model from the previous stage using leave-one-out cross-validation error approximated by Pareto smoothed importance sampling (PSIS-LOO; *Vehtari et al., 2017*).

In the first stage, we compared a model (#1) that combined outcome probability and outcome magnitude multiplicatively (i.e. by calculating expected value, similarly to *Browning et al., 2015*) with a model (#2) that combined outcome probability and outcome magnitude additively; see Materials and methods for full model details. We observed that the additive model fit participants' choice behavior better (model #2 PSIS-LOO = 26,164 versus model #1 PSIS-LOO = 27,801; difference in

PSIS-LOO = −1637; std. error of difference = 241; lower PSIS-LOO is better). This finding is consistent with observations of separate striatal representations for outcome magnitude and probability (*Berns and Bell, 2012*), as well as findings from work with non-human primates where additive models have also been reported to fit choice behavior better than expected value models (*Donahue and Lee, 2015*).

In both of the models in the first stage, all the parameters were divided into a baseline component, a component for the difference between volatile and stable blocks, a component for the difference between reward and aversive task versions, and a component for the interaction of these two experimental factors. During the second stage, we investigated whether task performance was better captured by additionally allowing for differences in learning as a result of positive and negative prediction errors. Specifically, we added a component for relative outcome value (good, bad) and two further components that captured the interaction of relative outcome value with block type (volatile, stable) and with task version (reward, aversive). We added these components for learning rate alone in one model (#3) and for learning rate, mixture weight, and inverse temperature in another model (#5); see Materials and methods for full model details. We defined a good outcome to be the receipt of a reward or the receipt of no shock and a bad outcome to be the receipt of no reward or the receipt of shock. Including effects of relative outcome value for all three model parameters, including the two-way interactions with block type and task version, improved PSIS-LOO (model #5 PSIS-LOO = 25,462 versus model #2 PSIS-LOO = 26,164; difference in PSIS-LOO = −702; std. error of difference = 142; lower PSIS-LOO is better). Adding the three-way interaction of block type, task version and relative outcome value worsened PSIS-LOO slightly (model #6 PSIS-LOO = 25,486 versus model #5 PSIS-LOO = 25,462; difference in PSIS-LOO = 24; std. error of difference = 9), indicating that two-way interactions were sufficient to capture important aspects of behavioral variation.

Additional stages of model comparison revealed that allowing subjective weighting of magnitude differences improved model fit (model #7 PSIS-LOO = 25,154 versus model #5 PSIS-LOO = 25,462; difference in PSIS-LOO = −308; std. error of difference = 104) as did the addition of a choice kernel that captures participants' predisposition to repeating prior choices (model #11 PSIS-LOO = 25,037 versus model #7 PSIS-LOO = 25,154; difference in PSIS-LOO = −117; std. error of difference = 42). Both the subjective magnitude parameter and choice kernel inverse temperature were broken down by task version (reward, aversive); a single choice kernel update rate was used across conditions; see Materials and methods: Stage-wise Model Construction. In contrast to the above parameters, adding a lapse term did not improve model fit nor did allowing outcome probabilities to be separately updated for each shape; see Materials and methods for further details.

The best fitting model (#11) is presented in *Equation 1a-d*. The probability ($p_t$) that shape 1 and not shape 2 would result in reward or shock receipt if chosen is updated on each trial using a prediction error (the difference between the most recent outcome $O_{t-1}$ and the previous estimate $p_{t-1}$) scaled by the learning rate ($\alpha$) (*Equation 1a*).

$$p_t = p_{t-1} + \alpha(O_{t-1} - p_{t-1}) \tag{1a}$$

Next, the estimate of the difference between the two shapes in outcome probability is combined additively with the difference in outcome magnitude using a mixture weight ($\lambda$) (*Equation 1b*). Here, the difference in outcome magnitude is nonlinearly scaled using $r$ to account for potential differences in subjective valuation ($M1_t$ and $M2_t$ denote the magnitude for shapes 1 and shape 2, respectively; note that the sign for the difference in this equation is removed before exponentiating and then restored).

$$v_t = \lambda[p_t - (1 - p_t)] + (1 - \lambda)[M1_t - M2_t]^r \tag{1b}$$

A choice kernel is also updated on each trial using the difference between the previous choice ($C_{t-1}$) and the choice kernel on the previous trial $k_{t-1}$, scaled by an update rate ($\eta$) (*Equation 1c*).

$$k_t = k_{t-1} + \eta(C_{t-1} - k_{t-1}) \tag{1c}$$

Finally, the outcome value and the choice kernel determine the probability that shape 1 was chosen on that trial using a softmax choice rule with two separate inverse temperatures ($\omega$ and $\omega_k$) (*Equation 1d*).

$$P(C_t = 1) = \frac{1}{1 + exp(-(\omega v_t + \omega_k[k_t - (1 - k_t)]))} \tag{1d}$$

To validate the model estimation procedure, we treated participants' estimated parameter values as the ground truth and used them to simulate 10 new datasets for each of the 86 participants. By fitting the model to these simulated datasets, we could compare the ground truth parameter values with the parameters estimated (i.e. recovered) from the simulated data. The recovered parameters from each dataset strongly correlated with the ground truth parameter values; the mean correlation across simulated datasets and across parameters was r = 0.76 (std = 0.15). For learning rate components, the average correlation was r = 0.88 (std = 0.13) (for more methodological details see Materials and methods: Parameter Recovery; for parameter recovery results see *Appendix 4—figure 1* and *Appendix 4—figure 2*). This analysis indicates that individual model parameters were recoverable as desired. For estimates of noise in population-level parameters, see *Appendix 4—figure 3* and *Appendix 4—figure 4*.

## Cross-group results: participants adjust learning rate to contingency volatility

Having selected model #11, we fit this model to participants' choice behavior and estimated distributions over model parameters at an individual- and population-level (as described above and detailed further in the Materials and methods). This included estimating population-level weights $\{\beta_g, \beta_d, \beta_a\}$ that captured the effect of internalizing factor scores upon each parameter component (e.g. $\alpha_{volatile-stable}$).

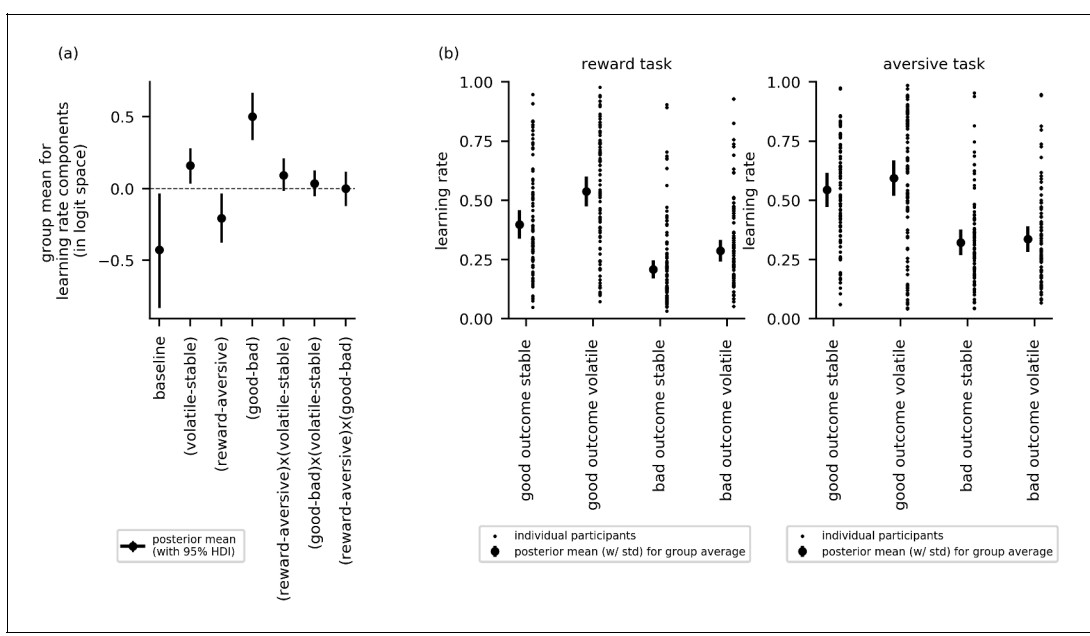

**Figure 3.** Cross-group results from experiment 1 for effects of block type (volatile, stable), task version (reward, aversive), and relative outcome value (good, bad) on learning rate (n = 86). (**a**) This panel shows the posterior means along with the 95% highest posterior density intervals (HDI) for the group means ($\mu$s) for each learning rate component (i.e. for baseline learning rate and the change in learning rate as a function of each within-subject factor and their two-way interactions). The 95% posterior intervals excluded zero for effect of block type upon learning rate (i.e. difference in learning rate for the volatile versus stable task blocks $\alpha_{volatile-stable}$). This was also true for the effect of task version, that is, whether outcomes entailed reward gain or electrical stimulation ($\alpha_{reward-aversive}$) and for the effect of relative outcome value, that is, whether learning followed a relatively good (reward or no stimulation) or relatively bad (stimulation or no reward) outcome ($\alpha_{good-bad}$). Participants showed higher learning rates during the volatile block than the stable block, during the aversive task than the reward task, and on trials following good versus bad outcomes. None of the two-way interactions were statistically credible, that is the 95% posterior included zero. (**b**) In this panel, the learning rate components are combined to illustrate how learning rates changed across conditions. The posterior mean learning rate for individual participants (small dots) and the group posterior mean learning rate (large dots, error bars represent the associated posterior standard deviation) are given for each of the eight conditions; these values were calculated from the posterior distributions of the learning rate components ($\alpha_{baseline}$, $\alpha_{volatile-stable}$, etc.) and the group means ($\mu$s).

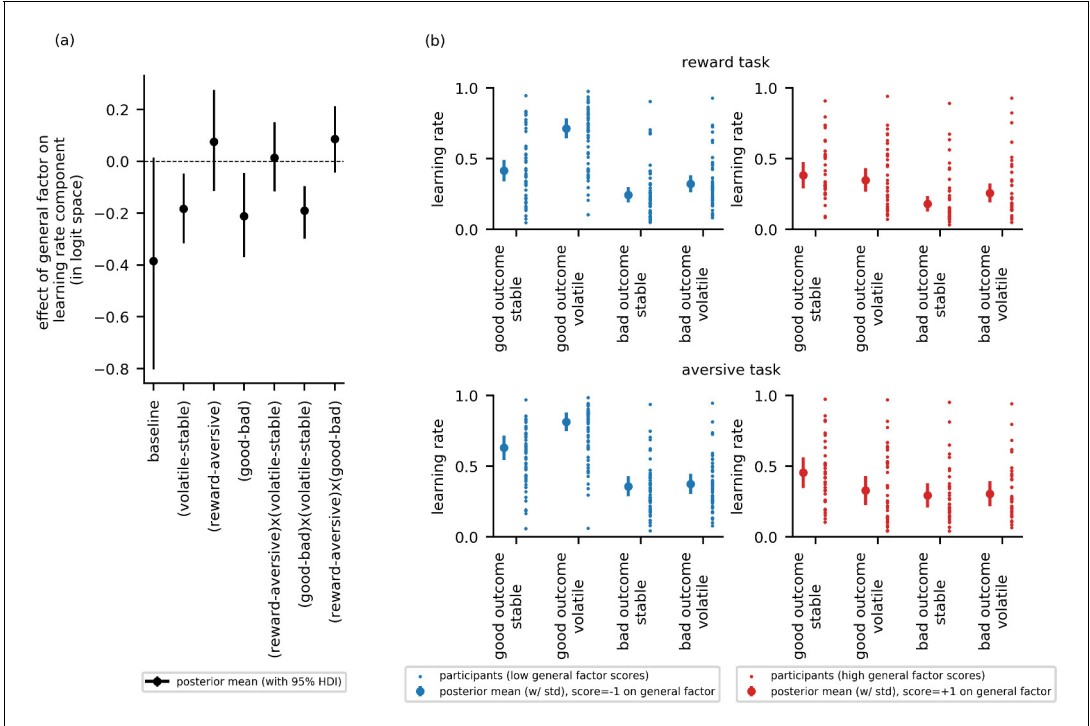

**Figure 4.** Experiment 1: Effect of general factor scores on learning rate (in-lab sample, n = 86). Panel (a) shows posterior means and 95% highest posterior density intervals (HDI) for the effect of general factor scores ($\beta_g$) on each of the learning rate components. General factor scores credibly modulated the extent to which learning rate varied between the stable and volatile task blocks ($\alpha_{volatile-stable}$; $\beta_g$ = −0.18, 95%-HDI = [−0.32,–0.05]), the effect of relative outcome value on learning rate ($\alpha_{good-bad}$; $\beta_g$ = −0.21, 95%-HDI = [−0.37,–0.04]) and the interaction of these factors upon learning rate ($\alpha_{(good-bad)x(volatile-stable)}$; $\beta_g$ = −0.19, 95%-HDI = [−0.3,–0.1]). In each case, the 95% HDI did not include 0. (b) Here, we illustrate learning rate as a function of each within-subject factor and high versus low scores on the general factor of internalizing symptoms. To do this, we calculated the expected learning rate for each within-subject condition associated with scores one standard deviation above ('high', shown in red) or below ('low', shown in blue) the mean on the general factor. It can be seen that the largest difference in learning rates for participants with high versus low general factor scores is on trials following good outcomes during volatile task blocks. This effect is observed across both reward and aversive task versions. Small data points represent posterior mean parameter estimates for individual participants. Large points represent the posterior mean learning rates expected for participants with scores ± 1 standard deviations above or below the mean on the general factor. Error bars represent the posterior standard deviation for these expected learning rates.

The online version of this article includes the following figure supplement(s) for figure 4:

**Figure supplement 1.** Effect of depression-specific and anxiety-specific factors on learning rate and its components (data from experiment 1).

In this section, we report cross-group effects (i.e. across all participants). Here, we used the posterior distributions over the group mean for each learning rate component to examine whether learning rate varied as a function of block type (volatile or stable), task version (reward or aversive), or relative outcome value (i.e. trials following good or bad outcomes). *Figure 3a* shows the posterior means, along with their 95% highest posterior density intervals (HDIs), for each parameter component. If the 95% HDI for a given effect does not cross zero, the effect is considered statistically credible.

The effects of block type, task version and relative outcome value upon learning rate were statistically credible; that is, their HDIs did not cross zero: block type ($\alpha_{volatile-stable}$), μ = 0.16, 95%-HDI = [0.04,0.28]; task version ($\alpha_{reward-aversive}$), μ = −0.21, 95%-HDI = [−0.37, –0.03]; relative outcome value ($\alpha_{good-bad}$), μ = 0.50, 95%-HDI = [0.34,0.67]. Participants had higher learning rates during the volatile block than the stable block, higher learning rates during the aversive task than the reward gain task, and higher learning rates on trials following good versus bad outcomes. None of the two-way interactions were statistically credible: block type by task version ($\alpha_{(volatile-stable)x(reward-aversive)}$), μ = 0.09, 95%-HDI [−0.02, 0.21]; block type by relative outcome value ($\alpha_{(volatile-stable)x(good-bad)}$), μ = 0.04, 95%-HDI [−0.05, 0.13]; task version by relative outcome value ($\alpha_{(reward-aversive)x(good-bad)}$), μ = 0.0, 95%-HDI

[−0.12, 0.12]. *Figure 3b* illustrates differences in learning rates between experimental conditions for the reward and aversive versions of the task, respectively.

## Elevated general factor scores are linked to reduced adjustment of learning rate to volatility, especially following better-than-expected outcomes, across both reward and aversive tasks

To address our first main research question—that is, whether impaired adjustment of learning rate to volatility is linked to symptom variance common to both anxiety and depression or to symptom variance specific to anxiety or to depression—we looked at whether the difference in learning rate between the volatile and stable blocks varied as a function of general factor scores or as a function of anxiety- or depression-specific factor scores. Examining learning rate difference between blocks, $\alpha_{volatile-stable}$, the 95% HDI for the general factor regression coefficient excluded zero, $\beta_g = -0.18$, 95%-HDI=[−0.32,−0.05], *Figure 4a*. Individuals with low scores on the general factor adjusted learning rate between the stable and volatile task blocks to a greater extent than individuals with high general factor scores. Neither anxiety-specific factor scores nor depression-specific factor scores credibly modulated learning rate difference between blocks, $\alpha_{volatile-stable}$, $\beta_a = -0.03$, 95%-HDI = [−0.16, 0.09], $\beta_d = 0.06$, 95%-HDI=[−0.08, 0.19], respectively, *Figure 4—figure supplement 1*. This suggests that the ability to appropriately adjust learning rate to volatility, previously linked to trait anxiety (*Browning et al., 2015*), is actually common to both anxiety and depression and not specific to one or the other.

To address our second research question—that is, whether the relationship between internalizing symptoms and adjustment of learning rate to volatility is domain general (i.e. holds across both aversive and reward task versions) or domain specific—we looked at whether there was an interaction between internalizing factor scores, block type and task version upon learning rate. Estimates for $\alpha_{(reward-aversive)x(volatile-stable)}$ were not credibly modulated by scores on any of the three internalizing factors ($\beta_g = 0.01$, 95%-HDI = [−0.12,0.15]; $\beta_a = 0.05$, 95%-HDI = [−0.07,0.17], $\beta_d = 0.06$, 95%-HDI = [−0.06, 0.17]), see *Figure 4a* and *Figure 4—figure supplement 1*. Estimates for the main effect of task version on learning rate ($\alpha_{reward-aversive}$) also did not vary credibly as a function of internalizing factor scores, $\beta_g = 0.08$, 95%-HDI = [−0.11; 0.28], $\beta_a = -0.11$, 95%-HDI = [−0.29,0.06], $\beta_d = -0.17$, 95%-HDI = [−0.38,0.01], see *Figure 4a* and *Figure 4—figure supplement 1*.

To address our third research question—that is, whether relative outcome value (good or bad) modulates the relationship between internalizing symptoms and learning rate adjustment to volatility—we looked at whether there was an interaction between internalizing factor scores, block type (volatile, stable) and relative outcome value (good, bad) upon learning rate. Estimates for $\alpha_{(good-bad)x(volatile-stable)}$ were credibly modulated by scores on the general factor ($\beta_g = -0.19$, 95%-HDI = [−0.3,−0.1]), *Figure 4a*, but not by scores on the anxiety-specific factor or the depression-specific factor ($\beta_a = -0.02$, 95%-HDI = [−0.11, 0.07], $\beta_d = 0.07$, 95%-HDI = [−0.04, 0.16]), *Figure 4—figure supplement 1*. In addition, scores on the general factor, but not the anxiety-specific or depression-specific factors, also credibly modulated the main effect of relative outcome value upon learning rate ($\alpha_{good-bad}$), $\beta_g = -0.21$, 95%-HDI = [−0.37, −0.04], see *Figure 4a* and *Figure 4—figure supplement 1*.

To illustrate these results, we calculated the expected learning rate for each within-subject condition associated with scores one standard deviation above or below the mean on the general factor, *Figure 4b*. Low general factor scores (shown in blue) were associated with higher learning rates following good versus bad outcomes, both when outcomes were reward-related (here a good outcome was a reward) and when they were aversive (here a good outcome was no shock delivery). Low general factor scores were also associated with a more pronounced difference in learning rate between volatile and stable task blocks following good outcomes. This was observed both when outcomes were reward-related and aversive. In contrast, high scores on the general factor (shown in red) were associated with smaller differences in learning rate between volatile and stable blocks and following good versus bad outcomes; this held across both reward and aversive versions of the task. In particular, the boost in learning from positive predictions errors under volatile conditions (i.e. from good relative outcomes in the volatile block) shown by individuals with low general factor scores was absent in individuals with high general factor scores; if anything individuals with high general factor scores' learning after positive outcomes was reduced under volatile relative to stable conditions.

To confirm that the relationship between general factor scores and the interaction of block type by relative outcome value on learning rate did not vary as a function of task version, we fit an additional model that parametrized the three-way interaction of block type (volatile, stable), relative outcome value (good, bad) and task version (reward, aversive) for learning rate. Fitting this model to participants' choice behavior confirmed that this three-way interaction was not statistically credible, nor was its modulation by general factor scores ($\beta_g$ = 0.06, 95%-HDI = [−0.06; 0.18]). The effect of general factor scores on the interaction between block type (volatile, stable) and relative outcome value (good, bad) on learning rate remained credible ($\beta_g$ = −0.21, 95%-HDI = [−0.34; −0.1]). We note that a parameter recovery analysis revealed successful recovery of the parameter representing the three-way interaction of block type, relative outcome value and task type on learning rate (*Appendix 4—figure 5*). This suggests that we did not simply fail to observe a three-way interaction due to lack of experimental power. Together, these findings support the conclusion that the negative relationship between general factor scores and increased learning rate following relative good

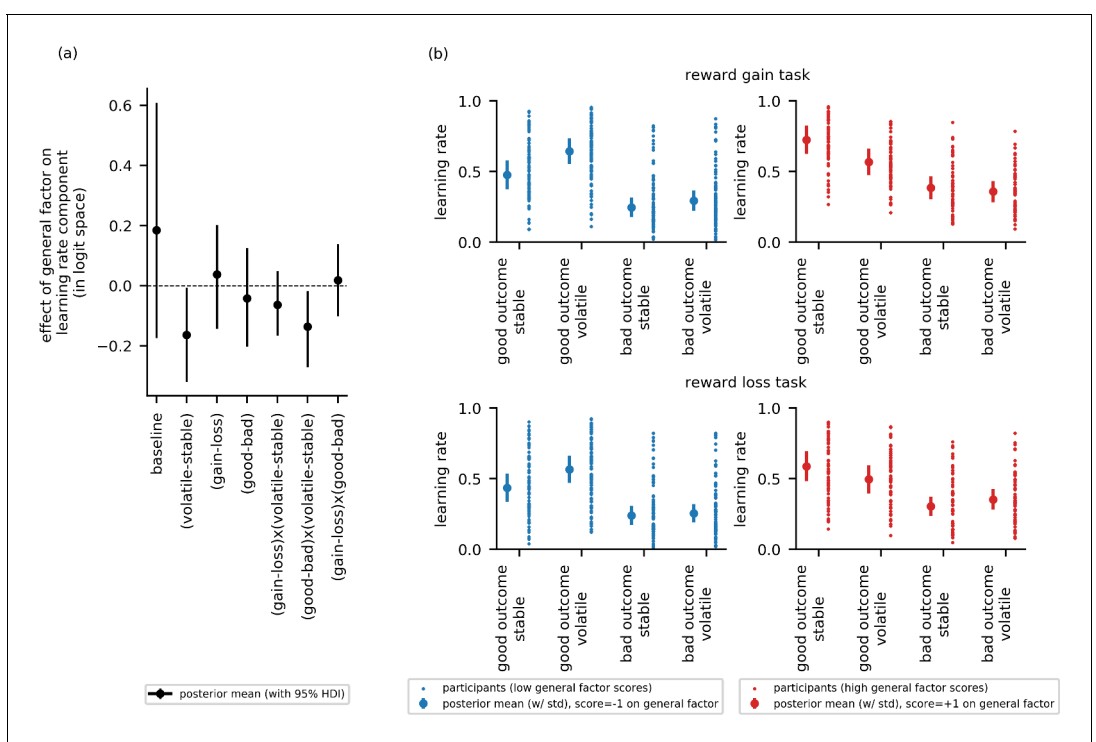

**Figure 5.** Experiment 2: Effect of the general factor scores on learning rate (online sample, n = 147). Panel (**a**) shows posterior means and 95% highest posterior density intervals (HDI) for the effect of general factor scores ($\beta_g$) on each of the learning rate components. Replicating findings from experiment 1, general factor scores credibly modulated the extent to which learning rate varied between the stable and volatile task block ($\alpha_{volatile-stable}$; $\beta_g$ = −0.16, 95%-HDI = [−0.32,−0.01]) and the extent to which this in turn varied as a function of relative outcome value ($\alpha_{(good-bad)x(volatile-stable)}$; $\beta_g$ = −0.14, 95%-HDI = [−0.27,−0.02]). (**b**) Here, we illustrate learning rate as a function of each within-subject condition and high (+1 standard deviation, shown in red) versus low (−1 standard deviation, shown in blue) scores on the general factor. As in experiment 1, participants will low general factor scores showed a boost in learning under volatile conditions following receipt of outcomes of good relative value (reward gain or no reward loss). Once again, this boost is not evident in participants with high general factor scores. Small data points represent posterior mean parameter estimates for individual participants. Large points represent the posterior mean learning rates expected for participants with scores ± 1 standard deviations above or below the mean on the general factor. Error bars represent the posterior standard deviation for these expected learning rates. As in experiment 1, there were no cross-group effect of task version (gain - loss) and no effect of general factor scores, or of anxiety- or depression- specific factor scores, on learning components involving task version (gain - loss). As in experiment 1, baseline rates of learning were highly variable. Results for anxiety and depression specific factor scores are shown in *Figure 5—figure supplement 2*.

The online version of this article includes the following figure supplement(s) for figure 5:

**Figure supplement 1.** Cross-group results from experiment 2 for effects of block type (volatile, stable), task version (reward, aversive), and relative outcome value (good, bad) on learning rate.

**Figure supplement 2.** Effect of depression-specific and anxiety-specific factors on learning rate and its components (data from experiment 2).

outcomes in volatile environments did not differ credibly as function of task version (reward, aversive).

## Experiment 2

We fit the behavioral model from experiment 1 to an independent online sample of participants, in order to test whether the reward gain findings replicated, and whether the findings for punishment would be replicated using loss of reward in place of primary aversive outcomes (shock). Specifically, we predicted that general factor scores would inversely correlate with adaptation of learning rate to volatility, that this would be observed to a greater extent following outcomes of positive relative value (reward gain or no reward loss) and that this would hold across task version (reward gain or reward loss).

One-hundred and seventy-two participants were recruited from Amazon's Mechanical Turk platform to participate in Experiment 2 (see Materials and methods: Experiment 2). We excluded participants who missed 10 or more responses in either the reward gain or reward loss version of the volatility task (the task is detailed further below); this left 147 participants (n = 147; 65 females) whose data were included in the analyses. Participants filled out the Spielberger State-Trait Anxiety Inventory (STAI), the Mood and Anxiety Symptoms Questionnaire (MASQ), and the Beck Depression Inventory (BDI), showing similar distributions of scores across these three measures to participants in Experiment 1 (see *Appendix 1—table 1* and *Appendix 1—table 3*). Participants' scores on the three internalizing latent factors were calculated using the factor loadings estimated in experiment 1 for items from this subset of measures. As can be seen in *Figure 1*, participants in experiment 2 (denoted by dark gray *o*s) showed a similar range of scores on the three factors to participants in experiment 1 despite no attempt being made to specifically recruit individuals who met diagnostic criteria for anxiety or depressive disorders.

As noted above, participants only completed three of the original questionnaires to reduce fatigue, incomplete datasets and participant drop-out. To check that factor scores could be reliably estimated using only items from this subset of the original questionnaires, we calculated factor scores using factor loadings estimated in experiment 1 and either the full or reduced set of questionnaire items. For this analysis, we pooled the questionnaire data from experiment 1 (n = 86) together with the online dataset used for the confirmatory factor analysis (n = 199). We correlated the factor scores calculated using either the full or reduced item set. This revealed little change in scores for the general factor (r = 0.97) or the depression-specific factor (r = 0.97). Scores on the anxiety-specific factor were only moderately correlated when using loadings from the full versus reduced set of items (r = 0.41); this likely reflects the omission of the PSWQ from the reduced questionnaire set. These differences reflect those seen between prior studies in the literature, where loadings on the general and depression factors are fairly consistent, but the anxiety factor can either reflect symptoms of anxious arousal, worry or both, with this varying as a function of measures included (*Clark et al., 1994*; *Steer et al., 1999*; *Steer et al., 2008*; *Brodbeck et al., 2011*). For the general factor and depression-specific factor, the high correlations between factor scores obtained using the full and reduced item set puts us in a strong position to draw conclusions across experiments 1 and 2. This is less true for the anxiety-specific factor. Our hypotheses center on the general factor; however, we include all scores on all three factors in the hierarchical model fitting procedure for consistency with experiment 1.

Each participant completed two versions of the probabilistic decision-making under volatility task (*Browning et al., 2015*): a reward gain and a reward loss task. The reward gain task was closely based on the in-lab reward task, differing only in the timing of intra- and inter-trial intervals, which were shortened slightly for the online version, and in the calculation of monetary bonuses (see Materials and methods). The reward loss task was parallel to the reward gain task, except that participants were allocated points at the start of the task and tried to avoid losing their points throughout the task. In this version of the task, outcome magnitudes corresponded the amount of points that would be subtracted from the point total. Participants' general level of performance on these online versions of the volatility tasks, as indexed by average magnitude of outcomes received across trials, was broadly similar to that observed in experiment 1 (see *Appendix 5—figure 1*).

## Experiment 2 results: general factor scores are linked to reduced adjustment of learning rate to volatility across both reward loss and reward gain. As in experiment 1, this was primarily observed following better-than-expected outcomes

At a group level, learning rates were credibly higher following good relative outcomes (reward, no loss) versus bad relative outcomes (no reward, loss) $(\alpha_{good-bad})$, $\mu$ = 0.52, 95%-HDI = [0.42,0.71]. There was no effect of task version on learning rate $(\alpha_{gain-loss})$, $\mu$ = 0.13, 95%-HDI = [−0.02,0.28]. In contrast to experiment 1, there was also no credible group-level effect of block type on learning rate $(\alpha_{volatile-stable})$, $\mu$ = 0.04, 95%-HDI = [−0.1,0.17], see *Figure 5—figure supplement 1*.

Although there was no effect of block type at the group level, in line with predictions, there was a credible interaction of block type by scores on the general factor: that is, the difference in learning rate between volatile and stable blocks, $\alpha_{volatile-stable}$, was inversely correlated with scores on the general factor $(\beta_g = -0.16$, 95%-HDI = [−0.32,−0.01]), *Figure 5a*. As in experiment 1, individuals with low general factor scores showed greater adjustment of learning rate to volatility, learning faster in the volatile block than the stable block, than participants with high general factor scores. Also as in experiment 1, neither depression nor anxiety-specific factor scores credibly modulated adjustment of learning rate to volatility $(\alpha_{volatile-stable})$: $\beta_a = -0.03$, 95%-HDI = [−0.14,0.08]; $\beta_d = 0.01$, 95%-HDI = [−0.14,0.12], *Figure 5—figure supplement 2*.

Once again, our second question of interest was whether deficits in adaptation of learning rate to volatility linked to elevated internalizing symptomatology would be domain general or domain specific. We examined if the relationship between scores on the three internalizing factors and the effect of block type (volatile, stable) on learning rate varied as a function of task version (reward gain, reward loss). As in experiment 1, none of the corresponding three-way interactions were statistically credible: general factor by block type by task version: $\beta_g = -0.06$, 95%-HDI = [−0.17, 0.05]; anxiety-specific factor by block type by task version: $\beta_a = 0.01$, 95%-HDI = [−0.09,0.1]; or depression-specific factor by block type by task version: $\beta_d = -0.03$, 95%-HDI = [−0.13,0.09]. In addition, scores on the three internalizing factors did not credibly modulate the main effect of task version (reward gain, reward loss) upon learning rate: $\beta_g = -0.04$, 95%-HDI = [−0.14,0.20]; $\beta_a = -0.15$, 95%-HDI = [−0.28,0.0]; $\beta_d = 0.01$, 95%-HDI = [−0.14,0.16].

Our third question of interest was whether deficits in adaptation of learning rate to volatility linked to elevated internalizing symptoms would vary depending on whether outcomes were better or worse than expected. Here, we replicated the finding from experiment 1. Specifically, general factor scores modulated the interaction of block type (volatile, stable) and relative outcome value (good, bad) upon learning rate, $\alpha_{(volatile-stable)x(good-bad)}$, $\beta_g = -0.14$, 95%-HDI = [−0.27,−0.02], *Figure 5a*. As in experiment 1, individuals with low scores on the general internalizing factor showed greater increases in learning rates in the volatile relative to stable task block following better-than-expected versus worse-than-expected outcomes. This boost in learning from positive predictions errors under volatile conditions was absent in individuals with high general factor scores; as in experiment 1, if anything, individuals with high general factor scores' learning after positive outcomes was reduced under volatile relative to stable conditions, *Figure 5b*. In contrast to these findings, neither scores on the anxiety-specific factor nor on the depression-specific factor credibly modulated the interaction of block type by relative outcome value on learning rate $(\beta_d = 0.08$, 95%-HDI = [−0.02,0.18]; $\beta_a = 0.01$, 95%-HDI = [−0.09,0.1]), *Figure 5—figure supplement 2*.

These findings, alongside those from experiment 1, indicate that a failure to boost learning when contingencies are volatile, especially following a better than expected outcome, is a shared characteristic of both anxiety and depression. We note that, unlike in experiment 1, scores on the general factor were not associated with increased overall learning for good versus bad outcomes, $\alpha_{good-bad}$ (i.e. a main effect of relative outcome value, independent of its interaction with block type), $\beta_g = -0.04$, 95%-HDI = [−0.20, 0.13], *Figure 5a*.

### Other model parameters

The behavioral model contained five other parameters (inverse temperature $\omega$, mixture weight for probability versus magnitude $\lambda$, choice kernel update rate $\eta$, choice kernel inverse temperature $\omega_k$, subjective magnitude $r$). We did not have a priori hypotheses pertaining to the relationship between

these parameters and scores on the three internalizing factors. Hence, we looked for statistically credible effects that replicated across both experiments 1 and 2.

Participants as a group relied on outcome probability more during attempts to obtain reward (reward gain) than during attempts to avoid punishment (shock in experiment 1, reward loss in experiment 2); the 95% posterior intervals excluded zero for the difference in mixture weight by task version for both experiment 1 (mixture weight; $\lambda_{reward-aversive}$, $\mu$ = 0.26, 95%-HDI = [0.01,0.49]) and for experiment 2 (mixture weight; $\lambda_{gain-loss}$, $\mu$ = 0.29, 95%-HDI = [0.02,0.55]). Excluding the learning rate results already presented, we did not observe any statistically credible associations between parameter values and scores on the three internalizing factors that replicated across both experiments 1 and 2.

## Discussion

We examined how adaptation of probabilistic decision-making to contingency volatility varied as a function of internalizing symptomology. Through bifactor analysis of item-level responses to standardized questionnaire measures of anxiety and depression, we estimated participants' scores on a general latent factor of internalizing symptomatology as well as their scores on anxiety-specific and depression-specific factors. We modeled participants' performance on alternate versions of a probabilistic decision-making under volatility task using a hierarchical Bayesian framework and population-level parameters that captured variance attributable to each of the three internalizing factors. In experiment 1, participants including individuals diagnosed with GAD and MDD performed reward gain and aversive versions of the task. In experiment 2, participants recruited online from Amazon Mechanical Turk performed reward gain and reward loss versions of the task. Across both experiments, we observed that high scores on the general factor were associated with reduced adjustment of learning rate to volatility. No parallel relationship was observed for scores on the anxiety or depression-specific factors. These findings are in line with the contention that impoverished adaptation of learning to contingency volatility is linked to the common component of anxiety and depressive symptomatology.

An important, logical, next question concerns the generality of this impairment. In the current study, we addressed this by examining whether impoverished adaptation of learning to volatility in individuals with high scores on the general factor holds for both rewarding and punishing outcomes. In experiment 1, we examined performance on parallel versions of the task using financial reward gain or electrical stimulation (shock) as outcomes; in experiment 2, we compared task performance when outcomes entailed financial reward gain or financial reward loss. Neither experiment found any evidence to suggest that the relationship between impoverished adaptation of learning rate to volatility and scores on the general internalizing factor was modulated by outcome domain. Across all three versions of the task, individuals with low general factor scores showed greater adaptation of learning rate to volatility than individuals with high general factor scores. This suggests that the impairment in adjusting learning to volatility linked to elevated internalizing symptoms generalizes across both reward and punishment learning.

It has previously been suggested that anxiety might be especially associated with altered learning regarding punishment (*Lissek et al., 2005*), whereas depression might be especially associated with altered learning regarding reward (*Elliott et al., 1997*; *Steele et al., 2007*). However, recent reviews have suggested that evidence for the latter, at least, is far from clear cut (*Robinson and Chase, 2017*). In the case of the volatility task used here, two previous studies found elevated trait anxiety to be linked to impoverished learning rate adjustment when outcomes involved punishment (electric stimulation or financial loss; *Browning et al., 2015*; *Pulcu and Browning, 2017*). For outcomes involving reward gain, the correlations between trait anxiety and learning rate adaptation to volatility were not credibly different from zero but also not credibly different to that observed with punishing outcomes (*Browning et al., 2015*; *Pulcu and Browning, 2017*). The small sample size of these prior studies means that only effects of moderate size were detectable. In addition, as touched on in the Introduction, the measure of anxiety used (the STAI trait scale) does not provide a particularly pure measure of anxiety-specific affect, also showing moderate-to-high correlations with measures of depression. In the current study, our larger sample size and bifactor analysis of internalizing symptoms enables us to better clarify whether anxiety-specific or depression-specific symptoms are linked to differential adaptation of learning rate to volatility as a function of outcome domain.

Considering anxiety first, we found no evidence to support a relationship between anxiety-specific symptoms and enhanced learning following punishing versus rewarding outcomes or to suggest a differential relationship between anxiety-specific symptoms and adaptation of learning to volatility when outcomes are punishing versus rewarding. We note that omission of PSWQ administration in experiment 2 limited the range of anxiety symptoms that could be captured in that experiment. However, it was experiment 1 that critically used electrical stimulation as punishing outcomes (this, as opposed to reward loss, is arguably the version of punishment where anxiety-specific effects are most expected) and this revealed no credible relationship between anxiety-specific symptoms and differences in learning rate as a function of task version (reward, aversive) or volatility by task version. Further, supplementary analyses using residual scores on the PSWQ or MASQ-AA, controlling for variance explained by the general factor, in place of scores on the anxiety-specific factor also reveal no credible relationship between anxiety-specific symptoms and differences in learning rate as a function of task version (reward, aversive) or volatility by task version (see *Appendix 6—figure 1* and *Appendix 6—figure 2*).

There was also no evidence to support a relationship between depression-specific symptoms and differential adaptation of learning to volatility as a function of outcome domain. These findings, together with those for the general factor reviewed above, suggest that the deficit in adapting learning rate to volatility is linked to symptom variance common to anxiety and depression and is general to both learning about reward and learning about punishment. We note that, in experiment 1, there was a trend toward depression-specific factor scores being linked to higher learning rates, in general, in the aversive versus reward version of the task. This did not reach significance and no equivalent effect was apparent in experiment 2. This might however be worth further exploration in future work.

Our third question of interest was whether impairment in adapting learning to volatility might differentially impact learning from better-than-expected versus worse-than expected outcomes. In the task used here, one shape predicts receipt of reward gain, loss or shock at 75–80% and the other at 20–25%; these contingencies are either stable or volatile depending on the block (following *Behrens et al., 2007* and *Browning et al., 2015*, we used slightly more extreme probabilities in the volatile block to balance performance between conditions). Since outcome probability is always greater than 0 and less than 1, outcomes of relative positive value (reward gain, no shock, no reward loss) will be better than expected, to a varying extent, and should generate a positive prediction error. Similarly, outcomes of relative negative value (no reward gain, shock, reward loss) will be worse than expected, to a varying extent, and should generate a negative prediction error. It is widely accepted that both positive and negative prediction errors are used to inform our estimates of outcome probability—that is, we learn from both better-than-expected and worse-than-expected outcomes. However, learning need not be symmetric. Here, across both experiments, we observed that individuals with low scores on the general internalizing factor mainly showed a volatility-related boost to learning from positive prediction errors. This selective boost to learning from positive prediction errors when contingencies were volatile was not shown by individuals with high scores on the general internalizing factor.

In the current task, asymmetry in learning from positive versus negative prediction errors does not confer a performance advantage given that both outcome types are equally fully informative— receiving a good outcome following selection of shape 1 means shape 2 would have resulted in a bad outcome and vice versa. However, asymmetric learning from good versus bad outcomes under volatility might be valuable in many real-world situations. Conceivably, when the world changes, there might normally be fewer ways of performing well than performing badly, so learning more substantially from the former could be advantageous (*Huys et al., 2015*). As such, enhanced learning from positive prediction errors when the world is volatile might reflect a learning style acquired over time by individuals with low general factor scores and applied by default in new situations, such as the current tasks.

Given the ubiquity of volatility in the real-world, our findings suggest that the development of interventions aimed at assisting participants to better distinguish contingency noise from contingency volatility, and to adjust decisions accordingly, might be useful in treatment of both anxiety and depression. An important question is whether impairment in adapting learning rate to volatility is specific to internalizing psychopathology or might represent an even more general deficit, perhaps one than also characterizes individuals with high levels of externalizing or psychotic

symptomatology. We hope to address this in future work. Understanding this will enable us to more broadly identify patients likely to benefit from cognitive interventions aimed at improving decision-making under different sources of uncertainty, in particular contingency noise versus contingency volatility. A second important future avenue entails research aimed at increasing our understanding of, and development of interventions for, deficits in decision-making uniquely linked to anxiety or depression-specific symptoms. The hope is that such interventions might valuably complement ones targeted at deficits linked to the general component of internalizing psychopathology.

In summary, our findings provide support for the contention that impoverished adjustment of learning rate to volatility is broadly linked to internalizing psychopathology. Bifactor modeling of internalizing symptoms and hierarchical Bayesian modeling of task performance enabled us to establish that this deficit is linked to symptoms shared by both anxious and depressed individuals, that is, to scores on a general internalizing factor, with no additional relationship being observed for anxiety or depression specific symptoms. The relationship between general factor scores and impaired adaptation of learning to volatility was observed in a sample including participants with anxiety and depressive disorders as well as in an online sample where no diagnostic interviews or pre-screening on symptoms was conducted. Further, this relationship held regardless of whether outcomes entailed receipt of reward, shock or reward loss. This speaks to the generality of the deficit in question. Intriguingly, individuals with low general factor scores were found to predominantly boost learning under volatility following better-than-expected outcomes. In other words, they learnt more from positive than from negative prediction errors under conditions of high contingency volatility. It is possible that in everyday life, a generally adaptive strategy when the world is rapidly changing is to learn from successful actions. The ability to do this could potentially confer resilience when faced with environments characterized by second-order uncertainty. In contrast, for individuals with high levels of internalizing symptoms, the inability to adjust behavior depending on whether unexpected outcomes are likely due to contingency noise or to contingency volatility might well comprise decision-making across multiple settings; this could explain findings that self-reported intolerance of uncertainty is elevated in individuals with both anxiety and depressive disorders. More generally, we believe that greater specification of the manner in which deficits in decision-making, and other cognitive functions, map onto latent dimensions of psychopathology will play an important role in the advancement of personalized medicine, allowing for empirically based stratification of individuals who participate in treatment trials, prediction of individuals at risk of clinically significant psychopathology, and development of interventions aimed at remediating core cognitive and computational deficits.

## Materials and methods

### Experiment 1

#### Participants

Potential participants of both sexes, between the ages of 18 and 55, were recruited continuously from the local community for a period of one year and 6 months between March 2015 and August 2016. Advertisements in local newspapers and on local mailing lists (e.g. Oxford University mailing lists) together with flyers at local primary care practices and geographically constrained Facebook advertisements were used to recruit participants with generalized anxiety disorder (GAD) and major depressive disorder (MDD). Diagnoses were determined using the research version of the Structured Clinical Interview for DSM-IV-TR (SCID) administered by trained staff and supervised by an experienced clinical psychologist. We excluded participants if they were currently receiving pharmacological treatment or had been prescribed psychotropic medication or taken non-prescribed psychoactive drugs (i.e. street drugs) within the past 3 months. Participants reporting a history of neurological disease or meeting diagnostic criteria for PTSD, OCD, bipolar disorder, schizophrenia or other psychotic disorders, or substance abuse were also excluded. In parallel, we recruited a healthy control group screened using the SCID to ensure they did not meet diagnostic criteria for any DSM-IV-TR Axis I disorder. Here, participants were also excluded if they reported a history of neurological disease or usage of psychoactive drugs (legal or illegal) within the last 3 months.

One-hundred and eight individuals came in for the SCID screening session. Of these, 42 individuals did not meet inclusion criteria and 8 individuals declined to participate in the subsequent

experimental sessions. Our final participant sample (n = 58) comprised 12 participants who met diagnostic criteria for GAD, 20 participants who met diagnostic criteria for MDD (three of whom had a secondary diagnosis of GAD), and 26 healthy control participants.

We also included within our final sample an additional 30 participants who had been recruited from the local community to perform the same tasks in the context of an fMRI study. These participants showed broadly the same age-range and sex ratio as the patient and control groups (see *Appendix 1—table 1*). Here, potential participants were excluded if they reported a prior diagnosis of neurological or psychiatric illness other than GAD or MDD. In addition, participants reporting usage of psychoactive medication or street-drugs were also excluded.

We excluded data from either the reward gain or aversive version of the probabilistic decision-making under volatility task if there was equipment malfunction or if a participant reported after the session that they did not understand the task. In experiment 1, we excluded data from the aversive version of the volatility task from eight participants (three participants with MDD, one participant with GAD, and four control participants). Data from the reward gain version of the volatility task were excluded for six participants (two participants with GAD, one participant with MDD, two control participants, and one community member participant). Only two participants (both control subjects) had data excluded from both tasks. These exclusions left 86 participants in total. Power calculations indicated a sample size of 75 or higher would give 95% power to obtain effect sizes similar to that observed in our earlier work relating adaptation of learning in the aversive version of the volatility task to trait anxiety (*Browning et al., 2015*). See *Appendix 1—table 1* for participant details by task.

## Experimental procedure

Experiment 1 was approved by the Oxford Central University Research Ethics Committee (CUREC) and carried out at the Wellcome Centre for Integrative Neuroimaging (WIN) within the John Radcliffe Hospital in compliance with CUREC guidelines. Written informed consent was obtained from each participant prior to participation. Participants recruited by community advertisement into the GAD and MDD patient groups or into the healthy control group were initially screened by phone. This was followed by an in-person screening session during which informed consent was obtained and the Structured Clinical Interview for DSM-IV-TR (SCID) was administered. Individuals meeting criteria for inclusion in the study were invited back for two additional sessions. During the second session, participants completed standardized self-report measures of anxiety and depression and then completed the aversive (shock) version of the volatility task. During the third session, participants completed the reward gain version of the volatility task. The second and third sessions were separated by at least 1 day and no more than 1 week. All three sessions were conducted within the Nuffield Department for Clinical Neurosciences at the John Radcliff Hospital. Participants were paid at a fixed rate of £20 per session and were also given a bonus of up to £10 based on their performance in the reward gain version of the volatility task.

To increase the number of participants and to fill in the spread of symptoms, 30 additional community-recruited participants (aged between 18 and 40 years, 14 females) were included in the sample for experiment 1. These participants were not administered the SCID, but any individuals reporting a history of psychiatric or neurological conditions were excluded as were individuals on psychotropic medication or taking illegal psychotropic agents. These participants completed the aversive and reward gain versions of the volatility task during two fMRI scanning sessions conducted a week apart in the Wellcome Centre for Integrative Neuroimaging at the John Radcliffe Hospital. Questionnaires were administered at the beginning of each of these sessions.

## Self-report measures of internalizing symptoms

Participants completed standardized self-report measures of anxiety and depression. Measures included the Spielberger State-Trait Anxiety Inventory (STAI form Y; *Spielberger et al., 1983*), the Beck Depression Inventory (BDI; *Beck et al., 1961*), the Mood and Anxiety Symptoms Questionnaire (MASQ; *Clark and Watson, 1995*; *Watson and Clark, 1991*), the Penn State Worry Questionnaire (*Meyer et al., 1990*), and the Center for Epidemiologic Studies Depression Scale (CESD; *Radloff, 1977*). In addition, we administered the 80-item Eysenck Personality Questionnaire (EPQ;

*Eysenck and Eysenck, 1975*) to be able to include items from the Neuroticism subscale in our bifactor analysis.

## Exploratory bifactor analysis

A bifactor analysis was conducted on item level responses (n = 128) to the MASQ anhedonia subscale, the MASQ anxious arousal subscale, the STAI trait subscale, the BDI, the CESD, the PSWQ, and the EPQ neuroticism (N) subscale. Item responses were either binary (0–1), quaternary (0–4), or quinary (0–5). Response categories that were endorsed by fewer than 2% of participants were collapsed into the adjacent category to mitigate the effects of extreme skewness. Reverse-scoring was implemented prior to inclusion of items in the bifactor analysis to facilitate interpretation of factor loadings. Polychoric correlations were used to adjust for the fact that categorical variables cannot have correlations across the full range of −1 to 1 (*Jöreskog, 1994*).

We determined the number of dimensions to use in the bifactor analysis based on theoretical considerations, visually inspecting the scree plot and conducting a parallel analysis, which compares the sequence of eigenvalues from the data to their corresponding eigenvalues from a random normal matrix of equivalent size (*Horn, 1965*; *Humphreys and Montanelli, 1975*; *Floyd and Widaman, 1995*). Parallel analysis was conducted using the 'fa.parallel' function from the Psych package in R. This procedure simulates a correlation matrix out of random variables drawn from a normal distribution, with the number of variables and the number of samples matched to the actual dataset. Eigenvalue decomposition is then applied to this simulated correlation matrix to estimate the magnitudes of eigenvalues that would be expected due to chance alone. These eigenvalues are plotted as a dotted line labeled 'random data' in *Figure 1—figure supplement 1*. Eigenvalue decomposition is also applied to the correlation matrix from the actual data and these eigenvalues are also plotted in *Figure 1—figure supplement 1* and labeled 'actual data'. Only three eigenvalues obtained from actual data lie above random data line and are hence reliably distinguishable from noise. Parallel analysis can also be conducted by randomizing the rows of the actual data matrix rather than drawing new random variables from a normal distribution. This procedure also supports a three factor solution for our dataset.

We first conducted an oblique factor analysis; following this we conducted a higher order factor analysis on the lower order factor correlations to extract a single higher order factor (i.e. a general factor). This procedure is known as Schmid-Leiman (SL) orthogonalization (*Schmid and Leiman, 1957*). Both steps were conducted using the 'omega' function from the Psych package in R. An alternative exploratory bifactor method, Jennrich-Bentler Analytic Rotations (*Jennrich and Bentler, 2011*), was also applied to the data. The factor scores calculated by the two methods were highly correlated (r = 0.96 for the general factor, r = 0.91 for the depression factor, and r = 0.96 for the anxiety factor). We chose to use SL orthogonalization, as the Jennrich-Bentler rotations attributed less unique variance to depression symptoms, potentially resulting in less power to detect depression-specific effects.

Factor scores for each participant were calculated using the Anderson-Rubin method (*Anderson and Rubin, 1956*), which is a weighted-least squares solution that maintains the orthogonality of the general and specific factor scores. A confirmatory bifactor analysis was conducted in a separate online participant sample (see below).

## The probabilistic decision-making under volatility task

Each participant completed both a reward gain version and an aversive version of the probabilistic decision-making under volatility task. These two tasks were as previously described in *Browning et al., 2015*. The two versions of the task had parallel structures. On each trial, participants were presented with two shapes on either side of the computer screen and were instructed to choose one. Each shape was probabilistically associated with either receipt of reward (in the reward gain version of the task) or receipt of shock (in the aversive version of the task). Participants were instructed that, on each trial, one of the two shapes would result in receipt of reward (or shock). They were also instructed that in making their decision whether to choose one shape or the other, they should consider both the probability of receipt of reward (or shock) and the reward (or shock) magnitude should it be received. Outcome magnitude (1-99) was randomly selected for each shape and changed from trial to trial. The same random sequence of magnitudes was used for all

participants who had the same block order (i.e. stable block or volatile block first) for a given version of the task. The sequence of magnitudes differed between the reward and aversive task versions.

For the reward gain version of the volatility task, reward points gained were converted to a monetary bonus that was paid out at the end of the experiment. The bonus ranged from £0 to £10 and was calculated by dividing the total sum of outcome magnitudes received by the maximum amount possible. For the aversive version of the volatility task, shock magnitude (1-99) corresponded to different intensities of electric stimulation. This mapping, or calibration, was conducted for each participant prior to performance of the task using the same procedure as reported by *Browning et al., 2015*. During calibration, participants reported their subjective experience of pain using a 10-point scale, on which 1 was defined as 'minimal pain' and 10 as 'worst possible pain'. Participants were told that the highest level of stimulation they would receive was a '7' on this scale and that for this they should select the highest level that they were willing to tolerate receiving up to 20 times during the task. The amplitude of a single 2 ms pulse of electrical stimulation was increased from zero until participants reported feeling a sensation that they rated as 1/10. The amplitude of the single pulse was then kept at this level while the number of 2 ms pulses delivered in a train was increased. The number of pulses was increased until the participant reported a subjective intensity of 7/10. If the participants reported a 7/10 before the number of pulses reached 8 or did not report a 7/10 by 48 pulses, the amplitude of the pulse corresponding to the 1/10 level was adjusted. Participants also completed 14 trials during which the intensity of electrical shock was randomly varied by changing the number of pulses delivered in a train between one and the number required to produce a report of 7/10. Participants' subjective pain ratings of these different levels of shock were fitted to a sigmoid curve. The single pulse reported as a 1/10 and the train of multiple pulses reported as a 7/10 formed the lowest (1) and highest (99) magnitudes for an outcome that a participant could receive during the task. The sigmoid curve was used map the outcome magnitudes in between 1 and 99 to numbers of pulses.

Each task was divided into a stable and volatile block of trials, each 90 trials long. In the stable block, one shape was associated with reward (or shock) receipt 75% of the time and the other shape was associated with reward (or shock) receipt 25% of the time. In the volatile block, the shape with a higher probability (80%) of resulting in reward (or shock) receipt switched every twenty trials; we used 80% as opposed to 75% to balance difficulty between the volatile and stable blocks (following *Behrens et al., 2007* and *Browning et al., 2015*). The order of task blocks by task version was counterbalanced across participants such that an equal number of participants received the volatile block of each task first, the stable block of each task first, the stable block of the reward task and the volatile block of the aversive task first, or the stable block of the aversive task and the volatile block of the reward task first. Participants were not told that the task was divided into two blocks.

## Computational modeling of task performance
### Decomposing parameters into parameter components

We fitted participants' choice behavior using alternate versions of simple reinforcement learning models. We sought to parsimoniously yet flexibly capture differences in choice behavior associated with task version (reward gain or aversive) and block type (volatile, stable) and to be able to address if learning rate varied as a function of relative outcome value. For the reward gain version of the task, good outcomes corresponded to the receipt of a reward and bad outcomes corresponded to its omission. For the aversive task, good outcomes corresponded to the omission of electric stimulation and bad outcomes corresponded to its delivery. While the magnitude of prediction errors varied across trials, good outcomes were always associated with positive prediction errors and bad outcomes with negative prediction errors.

We decomposed core model parameters, in particular learning rate, into components that captured the effects of block type (volatile, stable), task version (aversive, reward) and relative outcome value (good, bad) as well as the two-way interactions of these effects. As an example, in *Equation 2*, we show the division of learning rate $\alpha$ into a baseline learning rate $\alpha_{baseline}$, a difference in learning rates between the volatile and stable blocks $\alpha_{volatile-stable}$, a difference in learning rates between the reward gain and aversive versions of the volatility tasks $\alpha_{reward-aversive}$, a difference in learning rates between trials following good and bad outcomes $\alpha_{good-bad}$, and the two-way interactions of those

differences. We also explored the benefit of including three-way interactions, see model comparison.

$$
\begin{aligned}
\alpha = \mathrm{logistic}(&\alpha_{baseline} + \alpha_{(reward-aversive)}\chi_{(reward-aversive)} \\
&+ \alpha_{(volatile-stable)}\chi_{(volatile-stable)} \\
&+ \alpha_{(good-bad)}\chi_{(good-bad)} \\
&+ \alpha_{(volatile-stable)x(reward-aversive)}\chi_{(volatile-stable)x(reward-aversive)} \\
&+ \alpha_{(volatile-stable)x(good-bad)}\chi_{(volatile-stable)x(good-bad)} \\
&+ \alpha_{(reward-aversive)x(good-bad)}\chi_{(reward-aversive)x(good-bad)})
\end{aligned}
\tag{2}
$$

We use the term 'parameter components' to distinguish these elements from composite parameters, such as the overall learning rate ($\alpha$). The variable $\chi_{(volatile-stable)}$ takes on a value of 1 when the trial is in the volatile block and a value of $-1$ when the trial is in the stable block. A logistic transform was applied to constrain the learning rate to be between [0,1]. This transform was also applied to other parameters that were constrained to be between 0 and 1. A logarithmic transform was used for parameters that were constrained to be positive, such as inverse temperature. The parameter ranges are specified for each parameter in the model construction section.

## Hierarchical bayesian estimation of parameters

A hierarchical Bayesian procedure was used to estimate distributions over parameters for each reinforcement learning model, and to estimate how the potential dependences of these parameters on task version (reward versus aversive), block type (volatile versus stable) and relative outcome value (good versus bad) varied as a function of individual differences in internalizing symptomology. Data from all participants was used to fit each model. Specifically, each parameter component, such as the difference in learning rates between volatile and stable blocks ($\alpha_{volatile-stable}$, was assigned an independent population-level prior distribution that was shared across participants. The mean for each population-level distribution was specified as a linear model with an intercept ($\mu$), to represent the overall mean of the parameter component across participants, along with weights ($\beta_g, \beta_d, \beta_a$) for the participants' scores on each of the three factors ($X_g, X_d, X_a$) (g = general factor, d = depression-specific factor, a = anxiety-specific factor). The variance ($\sigma^2$) for the population-level distribution was also estimated separately for each parameter component. Note that there are different population-level parameters ($\mu, \beta_g, \beta_d, \beta_a, \sigma^2$) for each parameter component, but we omit this detail for notational ease. As an example, the population-level model for the learning rate difference between volatile and stable blocks ($\alpha_{volatile-stable}$) is given by *Equation 3*.

$$
\alpha_{volatile-stable} \sim Normal\big(\mu + \beta_g X_g + \beta_d X_d + \beta_a X_a, \sigma^2\big)
\tag{3}
$$

Models were fit using PyMC3 (*Salvatier et al., 2016*), a Python Bayesian statistical modeling software package. The hyperpriors assigned to these population-level parameters ($\mu, \beta_g, \beta_d, \beta_a$) were uninformative Normal(0,10). The hyperpriors for the population variances, $\sigma^2$, were Cauchy(2.5). A Hamiltonian Monte-Carlo method was used to sample from the full posterior. Four chains were run with 200 tuning steps and 2000 samples each. Visual inspection of the traces as well as Gelman−Rubin statistics ($R$) were used to assess convergence (*Gelman and Rubin, 1992*). There were no population-level parameters with $R$ values greater than 1.1 (most were below 1.01). There were only 8 out of the 2236 participant-level parameters (from two participants) with $R$ values greater than 1.1, and these were for $\eta$ and $\omega_k$, which were not the focus of the main analysis. The marginal posterior distributions for the population-level parameters ($\mu, \beta_g, \beta_d, \beta_a$) were used to assess statistical significance of population-level effects. Population-level parameters with a 95% highest posterior density (HDI) intervals that did not contain zero were deemed to be statistically credible.

## Stage-wise model construction

To find a model that was sufficiently flexible to capture important aspects of participants' behavior yet was not overfitted to the data, we performed a stage-wise model construction procedure. At each stage, we either added a computational component to the model or modified an existing component. We compared the enriched model to the best model from the previous stage using an

approximate form of leave-one-out cross-validation, which uses Pareto smoothed importance sampling (PSIS-LOO; *Vehtari et al., 2017*). Our baseline model, as well as the potential modifications, were informed by prior work (*Lau and Glimcher, 2005*; *Behrens et al., 2007*; *Ito and Doya, 2009*; *Li and Daw, 2011*; *Berns and Bell, 2012*; *Akaishi et al., 2014*; *Browning et al., 2015*; *Donahue and Lee, 2015*; *Mkrtchian et al., 2017*; *Aylward et al., 2019*). In total, 13 alternative models were assessed. The model with the lowest PSIS-LOO was selected as the winning model and used to make inferences about the relationships between task performance and internalizing symptoms. Each model was estimated according to the same hierarchical Bayesian procedure (detailed in the preceding section). In each case, participants' scores on the three internalizing factors were included in the estimation procedure.

## Stage 1: Additive versus multiplicative influence of outcome probability and magnitude (model #1 versus model #2)

In the first stage, we compared a model (#1) that assumes that participants combine outcome probability and outcome magnitude multiplicatively during decision making to a model (#2) that assumes that participants combine them additively. The multiplicative model #1 is similar to the model used in our previous work (*Browning et al., 2015*), and the additive model #2 is similar to models that have been found to fit well by other groups (e.g. *Donahue and Lee, 2015*).

In both models, the probability ($p_t$) that a good outcome would result from choosing shape 1 and not shape 2, is updated on a trial-by-trial basis using a Rescorla-Wagner rule (*Equation 4a and 5a*). The learning rate ($\alpha \in [0, 1]$) determines how much the estimate is revised by the prediction error (i.e. the difference between the previous estimate $p_{t-1}$ and the most recent outcome $O_{t-1}$). The outcome is coded such that $O_{t-1} = 1$ if shape 1 is chosen and followed by a good outcome (i.e. delivery of reward or absence of electric stimulation) or if shape 2 is chosen and followed by a bad outcome (i.e. absence of reward or delivery of electric stimulation). $O_{t-1} = 0$ codes for the opposite cases in each task.

For model #1, the outcome probability estimate is adjusted using a risk parameter ($\gamma \in [0.1, 10]$) to capture the relative importance of magnitude versus probability to choice. The expected value is calculated, multiplying the outcome probability and outcome magnitude for each shape separately ($M1_t$ and $M2_t$ for shapes 1 and 2, respectively), before taking the difference in expected value between shapes (*Equation 4c*). In contrast, in model #2, the differences in magnitude and probability are calculated separately for each shape first and are then combined as an additive mixture (*Equation 5b*). The mixture parameter ($\lambda \in [0, 1]$) in this model determines the relative importance of outcome magnitude and outcome probability, albeit in a different way to $\gamma$ in model #1. In both models, a softmax choice rule with an inverse temperature parameter ($\omega \in R^+$) is then used to specify how deterministically (or noisily) participants made choices as a function of the combined value of probability and magnitude on each trial (*Equation 4d and 5c*).

Equations 4a-d (Model #1).

(a)(updating probability estimates).

$$p_t = p_{t-1} + \alpha(O_{t-1} - p_{t-1}) \tag{4a}$$

(b)(risk adjusted probabilities).

$$p_t' = \min(\max((p_t - 0.5)^\gamma + 0.5, 0), 1) \tag{4b}$$

(c)(difference in expected value).

$$v_t = p_t' M1_t - \left(1 - p_t'\right)M2_t \tag{4c}$$

(d)(softmax action selection).

$$P(C_t = 1) = \frac{1}{1 + \exp(-\omega v_t)} \tag{4d}$$

Equations 5a-c (Model #2).

(a)(updating probability estimates).

$$p_t = p_{t-1} + \alpha(O_{t-1} - p_{t-1}) \tag{5a}$$

(b)(mixture of probability and magnitude).

$$v_t = \lambda[p_t - (1 - p_t)] + (1 - \lambda)[M1_t - M2_t] \tag{5b}$$

(c)(softmax action selection).

$$P(C_t = 1) = \frac{1}{1 + \exp(-\omega v_t)} \tag{5c}$$

The three parameters for model #1 ($\alpha$, $\gamma$, $\omega$) and the three parameters for model #2 ($\alpha$, $\lambda$, $\omega$) were each divided into four components: a baseline, a difference between volatile and stable blocks, a difference between the reward and aversive task, and the interaction of these two factors. We used these components in all the models considered given the importance of the experimental factors of block type (volatile, stable) and task version (reward, aversive) to our a priori hypotheses. For models #1 and #2, this resulted in twelve parameterized components estimated for each participant for each model.

PSIS-LOO was substantially lower for the additive model #2 than for the multiplicative model #1 (difference in PSIS-LOO = −1,637; std. error of difference = 241). Therefore, model #2 was carried onto the next stage of model comparison. This finding is in line with observations of separate striatal representations for outcome magnitude and probability (*Berns and Bell, 2012*). Additive models, as opposed to expected value models, have also been found to provide a better fit than expected value models in a probabilistic reversal learning task in non-human primates (*Donahue and Lee, 2015*).

## Stage 2: Influence of relative outcome value (model #2 versus models #3-#6)

We next investigated decomposing parameters according to whether the outcome on a given trial was better than or worse than expected, that is whether there was a positive or negative prediction error (*Frank et al., 2007*; *Cox et al., 2015*; *Eldar et al., 2016*). Specifically, we allowed the parameters in model #2 to differ on trials following good versus bad outcomes and for this difference in relative outcome value to interact with task version (reward versus aversive) and block type (volatile versus stable). As a reminder, good outcomes were defined to be the receipt of a reward in the reward gain version of the volatility task and an omission of electric stimulation in the aversive version of the volatility task.

In model #3, we used this additional decomposition for the learning rate parameter alone. This substantially improved PSIS-LOO relative to model #2 (difference in PSIS-LOO = −614; std. error of difference = 126). In model #4, we allowed inverse temperature $\omega$ and the mixture weight $\lambda$, instead of learning rate, to differ for good and bad outcomes; this also improved PSIS-LOO relative to model #2 (difference in PSIS-LOO = −122; std. error of difference = 50), but to a lesser extent than observed for model #3. In model #5, we allowed all three parameters to differ between good and bad outcomes and this achieved an even better PSIS-LOO relative to model #2 (difference in PSIS-LOO = −701; std. error of difference = 142). In model #6, we added the triple interaction between block type, task version, and relative outcome value; this resulted in a slightly higher (worse) PSIS-LOO relative to model #5 (difference in PSIS-LOO = 23; std. error of difference = 9). Therefore, model #5 was carried onto the next stage of model comparison. (Note that the equations for model #5 are identical to model #2, because the breakdown of composite parameters into parameter components is omitted from the equations to reduce notational clutter).

## Stage 3: Nonlinear effects of magnitude difference (model #5 versus model #7)

We next considered the possibility that participants might treat differences in outcome magnitudes non-linearly (i.e. that they make decisions on the basis of subjective rather than objective outcomes magnitudes), and that the degree of this nonlinearity might differ between rewarding and aversive outcomes. Model #7 uses a scaling parameter ($r \in [0.1, 10]$) to capture this potential non-linearity (*Equation 6b*; note that the sign for the difference in this equation was temporally removed before exponentiating and then added back again). The scaling parameter ($r$) was divided into a baseline

component and a component for the difference between the reward and aversive versions of the volatility task. This decomposition was chosen given the likely possibility that participants treat differences in reward magnitudes and differences in shock magnitudes differently. (Further division of this parameter was not performed to reduce the number of model comparisons). Adding this parameter improved PSIS-LOO relative to model #5 (difference in PSIS-LOO = −308; std. error of difference = 104). Therefore, model #7 was carried forward.

Equations 6a-c (Model #7).
(a) (updating probability estimates).

$$p_t = p_{t-1} + \alpha(O_{t-1} - p_{t-1}) \tag{6a}$$

(b) (mixture of probability and magnitude).

$$v_t = \lambda[p_t - (1 - p_t)] + (1 - \lambda)[M1_t - M2_t]^r \tag{6b}$$

(c) (softmax action selection).

$$P(C_t = 1) = \frac{1}{1 + \exp(-\omega v_t)} \tag{6c}$$

## Stage 4: Accounting for lapses in attention (model #7 versus model #8)

We next tested the inclusion of a lapse parameter ($\epsilon \in [0, 1]$; model #8; *Equation 7c*) to allow for the possibility that participants occasionally make unintended choices due to lapses of attention or motor errors. The lapse parameter was divided into a baseline component and a component for the difference between reward and aversive versions of the volatility task. Further divisions by experimental condition were not tested given that both the lapse parameter and the inverse temperature parameter similarly aim to capture noise in participants' choices and the inverse temperature parameter was already divided into seven different components. Model #8, which included the lapse parameter, had a slightly worse PSIS-LOO than Model #7 (difference in PSIS-LOO = 31; std. error of difference = 43). Therefore, Model #7 was retained as the best model and carried forward.

Equations 7a-c (Model #8).
(a) (updating probability estimates).

$$p_t = p_{t-1} + \alpha(O_{t-1} - p_{t-1}) \tag{7a}$$

(b) (mixture of probability and magnitude).

$$v_t = \lambda[p_t - (1 - p_t)] + (1 - \lambda)[M1_t - M2_t]^r \tag{7b}$$

(c) (softmax action selection with lapse).

$$P(C_t = 1) = (1 - \epsilon)\frac{1}{1 + \exp(-\omega v_t)} + \epsilon/2 \tag{7c}$$

## Stage 5: Separate probability estimates for each shape (model #7 versus models #9-#10)

We next allowed for the possibility that participants maintain two separate probability estimates, one for shape 1 and one for shape 2. Although participants were instructed that these two probabilities were yoked (i.e. p and 1-p), it was possible that they treated them independently. Stimulus-specific (or action-specific) probability estimates are commonly found in models, such as Q-learning (*Li and Daw, 2011*; *Mkrtchian et al., 2017*; *Aylward et al., 2019*). Hence, we compared model #7 to two different models that update stimulus-specific probability estimates.

Model #9 updates the two probability estimates using a Rescorla-Wagner rule, similarly to model #7. On each trial, only one of these two estimates is updated, that is, the one corresponding to the shape chosen by the participant on that trial. In this model, $O1_{t-1} = 1$ if shape 1 is followed by a good outcome, whereas $O2_{t-1} = 1$ if shape 2 is followed by a good outcome. When bad outcomes occur, these variables take on a value of zero. After the update, both estimates decay toward 50% (*Equation 8c-d*) using a decay rate ($\delta \in [0, 1]$). The decay parameter consisted of a baseline component and a component for the difference between reward and aversive versions of the volatility task. The decay and learning rate act in opposite directions on probability estimates, which can make it

difficult to separately estimate their effects. To mitigate this estimation issue, we did not allow decay to differ by block type (volatile versus stable) or by relative outcome value (good versus bad).

Equations 8a-c (Model #9).

(a) (updating probability estimates, shape 1).

$$p_t = p_{t-1} + \alpha(O1_{t-1} - p_{t-1}) \tag{8a}$$

(b) (updating probability estimates, shape 2).

$$q_t = q_{t-1} + \alpha(O2_{t-1} - q_{t-1}) \tag{8b}$$

(c) (decay probability estimate, shape 1).

$$p_t' = (1-\delta)p_t + (\delta)0.5 \tag{8c}$$

(d) (decay probability estimate, shape 2).

$$q_t' = (1-\delta)q_t + (\delta)0.5 \tag{8d}$$

(e) (mixture of probability and magnitude).

$$v_t = \lambda\left[p_t' - q_t'\right] + (1-\lambda)[M1_t - M2_t]^r \tag{8e}$$

(f) (softmax action selection).

$$P(C_t = 1) = \frac{1}{1 + \exp(-\omega v_t)} \tag{8f}$$

Model #10 uses a Beta-Bernoulli Bayesian model to update the outcome probability estimates for each shape. Outcome probabilities are estimated by updating the counts of good or bad outcomes that followed the choice of each shape (**Equations 9a–d**). The update parameter ($\alpha \in [0, 10]$) acts similarly to a learning rate. Only one of these counts is updated on each trial, that is, the one for the chosen shape and the outcome received. Otherwise the values of these counts decay towards zero (**Equations 9a–d**). The outcome probabilities for each shape are calculated as the mean of the Beta distribution using **Equations 9e–f**.

Equations 9a-c (Model #10).

(a) (updating good outcome counts, shape 1).

$$a_t = \delta a_{t-1} + \alpha \tag{9a}$$

(b) (updating bad outcome counts, shape 1).

$$b_t = \delta b_{t-1} + \alpha \tag{9b}$$

(c) (updating good outcome counts, shape 2).

$$c_t = \delta c_{t-1} + \alpha \tag{9c}$$

(d) (updating bad outcome counts, shape 2).

$$d_t = \delta d_{t-1} + \alpha \tag{9d}$$

(e) (calculating probability estimates, shape 1).

$$p_t = \frac{a_t + 1}{a_t + b_t + 2} \tag{9e}$$

(f) calculating probability estimates, shape 2.

$$q_t = \frac{c_t + 1}{c_t + d_t + 2} \tag{9f}$$

(g) (mixture of probability and magnitude).

$$v_t = \lambda[p_t - q_t] + (1 - \lambda)[M1_t - M2_t]^r \tag{9g}$$

(h) (softmax action selection).

$$P(C_t = 1) = \frac{1}{1 + \exp(-\omega v_t)} \tag{9h}$$

Neither model #9 nor model #10 improved PSIS-LOO relative to model #7 (difference in PSIS-LOO = 223 for model #9; std. error of difference = 72; and difference in PSIS-LOO = 171 for model #10; std. error of difference = 71; respectively). Therefore, model #7 was retained as the best model.

## Stage 6: Including a choice kernel (model #7 versus model #11)

In the sixth stage, we allowed for the possibility that participants tended to repeat (or avoid repeating) past choices independently of the recent outcomes received (*Lau and Glimcher, 2005*; *Ito and Doya, 2009*; *Akaishi et al., 2014*). Model #7 was augmented to include a choice kernel ($k_t$), which acts like a moving average for the participant's recent choices. The choice kernel is updated using an update rate ($\eta \in [0, 1]$; *Equation 10c*), which effectively determines how many previous choices are contained in the value of the choice kernel on the current trial. The impact of the choice kernel on choice was determined by an additional inverse temperature parameter ($\omega_k \in R^+$; *Equation 10d*). The update rate ($\eta$) consisted of only the baseline component and the inverse temperature parameter ($\omega_k$) was divided into a baseline component and a component for the difference between reward and aversive versions of the volatility task. Further division of these two parameters was not performed, because the Monte-Carlo Markov chains used for parameter estimation failed to converge when allowing further dependences on experimental condition (i.e. task version for the update rate or block type for update rate and inverse temperature) or relative outcome value (good versus bad); this suggests that there is insufficient information in the data for discriminating different values of these parameters across these components.

Equations 10a-d (Model #11).

(a) (updating probability estimates).

$$p_t = p_{t-1} + \alpha(O_{t-1} - p_{t-1}) \tag{10a}$$

(b) (mixture of probability and magnitude).

$$v_t = \lambda[p_t - (1 - p_t)] + (1 - \lambda)[M1_t - M2_t]^r \tag{10b}$$

(c) (updating choice kernel).

$$k_t = k_{t-1} + \eta(C_{t-1} - k_{t-1}) \tag{10c}$$

(d) (softmax action selection).

$$P(C_t = 1) = \frac{1}{1 + exp(-(\omega v_t + \omega_k[k_t - (1 - k_t)]))} \tag{10d}$$

Model #11 improved PSIS-LOO over model #7 (difference in PSIS-LOO = −117; std. error of difference = 42) and was therefore carried forward to the next stage.

## Stage 7: Adding a choice kernel to models from Stage 5 (model #11 versus models #12 and #13)

Finally, we examined adding the choice kernel to models from stage 5 (model #9 and model #10). Model #12 and model #13 were identical to models #9 and model #10, respectively, except for the inclusion of a choice kernel. The equations for model #12 and #13 are omitted because they can be directly obtained by replacing either *Equation 8f* or *Equation 9h* with *Equations 10c-d*. Neither of these two models improved PSIS-LOO over model #11 (difference in PSIS-LOO = 179 for model #12; std. error for difference = 71; and difference in PSIS-LOO = 144 for model #13; std. error for difference = 63); therefore, model #11 was retained as the best fitting model.

## Winning model

As the winning model, model #11 is used in the main manuscript (see *Equations 10a-d*). For ease of reference, we provide the full list of parameter components again here. The learning rate ($\alpha$), the mixture parameter ($\lambda$), and the inverse temperature parameter ($\omega$) were each broken down into seven components: a baseline, three main effects: block type (volatile versus stable), task version (reward versus aversive), relative outcome value (good versus bad), and the three two-way interactions of these effects. The inverse temperature ($\omega_k$) for the choice kernel and the subjective magnitude parameter ($r$) each had a baseline component and were allowed to differ between the reward and aversive versions of the task, but not between volatile and stable blocks or between trials following good or bad outcomes. The update rate ($\eta$) for the choice kernel had a single baseline component that did not vary by task version, block type or relative outcome value.

The results of model comparison for experiment 1 are summarized in *Appendix 3—table 1*.

### Model parameter recovery

We conducted a parameter recovery analysis to check that model #11's parameters were identifiable. Subject-specific posterior means for each component of each parameter from model #11 (e.g. $\alpha_{baseline}$, $\alpha_{volatile-stable}$, etc.) were used to simulate new choice data. The winning model was then re-fit to each of these simulated datasets. The original parameter components estimated from the actual dataset (referred to as 'ground truth' parameters) were correlated with the newly estimated parameter components (referred to as 'recovered' parameters) for each simulated dataset. An example of one simulated dataset is given in *Appendix 4—figure 1* for learning rates and in *Appendix 4—figure 2* for the other parameters. This procedure was repeated for 10 simulated datasets.

We also examined the robustness of estimates for population-level parameters ($\mu, \beta_g, \beta_a, \beta_d$) by looking at the variability in their values across the simulated datasets. Variability across datasets reflects sensitivity to noise in participants' choices, which is estimated by the fitted values for the two inverse temperatures in the model. The population-level parameters for each simulated dataset are shown in *Appendix 4—figure 3* for learning rate and in *Appendix 4—figure 4* for the other model parameters.

We also fit an additional model that parametrized the three-way interaction of block type (volatile, stable), relative outcome value (good, bad), and task version (reward, aversive) for learning rate. We conducted a parameter recovery analysis for this model that paralleled that described above. Specifically, 10 additional datasets were simulated using the subject-specific parameter components from this model, the model was refit to these simulated data, and the recovered parameters were correlated with the ground truth parameters. An example of one simulated dataset for this analysis is given in *Appendix 4—figure 5*.

### Model reproduction of basic features of choice behavior

As a final check, we simulated data from the winning model to see if it would reproduce basic qualitative features of participants' actual choice behavior. The number of trials on which a participant stays with the same choice or switches to the other choice is one qualitative feature that our model should be able to reproduce even though it was not optimized to do so. In each of the simulated datasets described in the previous section, we summed the number of trials on which each simulated participant switched from one choice to the other. Each simulated participant corresponded to an actual participant whose estimated parameter values were used to generate the data for that simulated participant. Therefore, we can examine the correlation between the number of switch trials made by the simulated participants and those made by the actual participants. The actual number and the simulated number of switch trials was highly correlated across participants; see Appendix 7.

## Confirmatory bifactor analysis

A new sample of participants was recruited and administered the battery of anxiety and depression questionnaires used in experiment 1 to test the reproducibility of the factor structure estimated in experiment 1.

## Participants

Three hundred and twenty-seven participants (203 women; mean age = 21.1 years) were recruited from UC Berkeley's psychology research pool and asked to fill out the same battery of anxiety and depression questionnaire measures as used in Experiment 1. This questionnaire session was completed online in a single session, using Qualtrics (*Qualtrics, 2014*). Experimental procedures were approved by the University of California-Berkeley Committee for the Protection of Human Subjects and carried out in compliance with their guidelines. Participants checked a box online to indicate consent prior to participation. Participants received course credit for participation. Participants were excluded from the confirmatory bifactor analysis if their dataset contained one or more missing responses; 199 complete data-sets were obtained (see *Appendix 1—table 2* for participant details).

## Measures of internalizing symptoms

Participants completed the same standardized questionnaire measures of anxiety, depression and neuroticism as listed under experiment 1.

## Confirmatory bifactor analysis

For the confirmatory bifactor analysis, all of the individual item-level questions were allowed to load on the general factor. Additionally, each item was also allowed to load onto either the anxiety-specific factor or the depression-specific factor. This assignment was determined by whether the item had a loading greater than 0.2 on that specific factor in the exploratory bifactor analysis conducted within experiment 1. Fifty-five items had loadings greater than 0.2 on the depression-specific factor, 35 items had loadings greater than 0.2 on the anxiety-specific factor, and the remaining items had loadings less than 0.2 for both the depression- and anxiety-specific factors; no items had loadings greater than 0.2 on both the depression- and anxiety-specific factors. After item assignment, factor loadings were re-estimated. Diagonally weighted least squares estimation was used, because it is more appropriate for ordinal data and less sensitive to deviations from normality than maximum likelihood estimation (*Li, 2016*). This procedure was conducted using the Lavaan package in R and specifying that an orthogonal solution should be obtained. Quality of fit was determined by the comparative fit index (CFI) and the root mean square error of approximation (RMSEA).

## Experiment 2

### Participants

One hundred and seventy-two participants (74 females) were recruited from Amazon's Mechanical Turk. Using qualifications on the Mechanical Turk platform, participants were restricted to be from the United States and were required to be 18 years of age or older. Participants' specific ages were not recorded. Twenty-five participants were excluded from the online dataset for having greater than 10 missed responses in each task, leaving 147 participants for analysis.

### Experimental procedure

This experiment was approved by the University of CaliforniaBerkeley Committee for the Protection of Human Subjects and carried out in compliance with their guidelines. One hundred and seventy-two participants were recruited using Amazon's Mechanical Turk platform. Participants viewed an online version of the consent form and were asked to click a box to indicate their consent. They were then redirected to an externally hosted website to take part in the experiment. On the externally hosted website, participants first completed standardized self-report measures of anxiety and depression and then completed two alternate versions of the volatility task: a reward gain and a reward loss task. These alternate versions were modified from those in experiment 1 to be suitable for online administration. The reward gain task and reward loss task were completed in the same session. Participants were required to take a 5-min break after filling out the questionnaires and a second 5-min break before completing the second task.

Participants were paid a fixed amount of $8 for completing the experimental session, which took approximately 1–1.5 hr. Participants could also earn a bonus between $0 and $3 depending on their performance. At the start of the experimental session, participants were told that their cumulative performance across both versions of the volatility task would be compared to other participants'

performance and a bonus of $3 will be awarded to participants that score in the top 5%, $1 to those in the top 10%, and $0.25 to the top 50%.

## Self-report measures of internalizing symptoms
Participants in experiment 2 completed the Spielberger State-Trait Anxiety Inventory (STAI form Y; *Spielberger et al., 1983*), the Beck Depression Inventory (BDI; *Beck et al., 1961*), the Mood and Anxiety Symptoms Questionnaire (MASQ; *Clark and Watson, 1995*; *Watson and Clark, 1991*). Participants showed similar distributions of scores on these measures to participants in experiment 1; see *Appendix 1—table 1* and *Appendix 1—table 3* for the means and standard deviations of questionnaire scores for participants in experiment 1 and experiment 2, respectively.

## Estimation of latent factor scores
Factor scores were calculated using the 80 items from the STAI trait scale, the BDI and the MASQ. Factor scores were calculated using the loadings (i.e. factor structure) estimated in experiment 1 and regressing participants responses from experiment 2 onto those loadings using a weighted least-squares regression (*Anderson and Rubin, 1956*). Prior to the calculation of factor scores, item responses were normalized across datasets (across the data from participants in experiment 1, the online UC Berkeley participant dataset used for the confirmatory factor analysis and data from participants in experiment 2). We also used data from the first two of these datasets to compare the factor scores obtained using the STAI, MASQ, and BDI versus the full set of questionnaires.

## Online version of the probabilistic decision-making under volatility task
The online versions of the probabilistic decision-making under volatility task comprised a reward gain and a reward loss version of the task. The structure of the task was kept as close to the in-lab versions as possible. On each trial, participants were presented with two shapes, one on either side of the computer screen, and were instructed to choose one or the other. Each shape was probabilistically associated with the gain or loss of reward; the nature of the outcome (reward gain or reward loss) depending on the version of the task. When making their choice, participants were instructed to consider both the magnitude of the potential gain or loss associated with each shape, which was shown to participants inside each shape and varied across trials, and the probability that each shape would result in reward gain or reward loss. The outcome probability could be learned across trials, using the outcomes received. Participants were instructed that, on each trial, only one of the two shapes would result in the gain (or loss) of reward if chosen, while the other shape would result in no gain (or no loss) if chosen.

In the reward gain version of the task, the magnitude of potential reward gain associated with each shape, on each trial, varied from 1 to 99 reward points that would be added to the participants' point total if choice of that shape resulted in reward receipt. In the reward loss version of the task, the magnitudes illustrated corresponded to 1 to 99 points that would be *subtracted* from the participants' point total if choice of the given shape resulted in reward loss. On trials where participants chose the shape not associated with a gain (or loss), zero points were added (or subtracted) from the participants' point total. In the reward gain task, participants started with a total of 0 points, while in the reward loss task, participants started with 5000 points. Outcome magnitudes for the two shapes were varied across trials; the sequences of outcome magnitudes for the reward gain task and for the reward loss task were the same as those used for the reward gain task and the aversive task in experiment 1, respectively. At the end of both tasks, the total number of points remaining was summed and used to determine the monetary bonus.

Each task was divided into a stable and volatile block of trials, each 90 trials long. In the stable block, one shape was associated the gain (or loss) of reward 75% of the time and the other shape was associated with the gain (or loss) of reward 25% of the time. In the volatile block, the identity of shape with a higher probability of resulting in a gain or loss (80% probability) switched every 20 trials. The identity of the first block, stable or volatile, was randomized across participants. Participants were not told that the task was divided into two blocks.

The in-lab and online versions of the volatility tasks differed in certain details. In the in-lab versions, participants were given 8 s to respond, while in the online version, they were given 6 s. Additionally, the time between choice and outcome presentation was shortened from 4 s (in-lab) to 1 s

(online). The time following outcome presentation and before the next trial was also shortened from 4 s (in-lab) to 1 s (online). The shapes used in the two experiments also differed. In-lab, the two shapes were either a circle or a square and textured as Gabor patches. Online, the two shapes were both circles, one red and one yellow.

## Computational modeling of task performance

Participants' choice data from the two online versions of the probabilistic decision-making task were modeled using the best-fitting model (#11) from experiment 1. The same hierarchical Bayesian procedure was used to fit model #11 as described previously (see Materials and methods: Experiment 1). As in experiment 1, participants' scores on the three internalizing factors were entered into the population-level prior distributions for model parameters. Hamiltonian Monte-Carlo methods were used to sample from the posterior distributions over the model parameters. The resulting posterior distributions and their 95% highest density intervals (HDI) for the population-level parameters were used to test hypotheses regarding the relationship between internalizing symptoms and task performance.

## Checking model fit

For consistency, we sought to use the same model to fit participants' data from experiment 2 as that used in experiment 1. It was possible that differences in procedure (online versus in lab, timing changes, use of reward loss in place of shock) might impact model fit. Hence, we sought to validate that the best-fitting model from experiment 1 (model #11) was also a good fit to data from online participants in experiment 2. To address this, we fit all 13 models to the online data and compared fits across models. Model #11 had the second lowest PSIS-LOO (*Appendix 3—table 2*). Model #12 (which differs from model #11 only in the use of separate learning rates for each shape) had a slightly lower PSIS-LOO. However, the difference in PSIS-LOO for models #11 and #12 was within one standard error (difference in PSIS-LOO = 43; SE = 49). In contrast, model #11's PSIS-LOO was more than two standard errors better than model #12 in experiment 1 (difference in PSIS-LOO = 179, SE = 71). Furthermore, the average PSIS-LOO across both experiments was lower for model #11 (PSIS-LOO = 38,543) than for Model #12 (PSIS-LOO = 38,611). Hence, if we seek to retain one model across both experiments, model #11 is the better choice.

As in experiment 1, we also checked that model #11 could reproduce basic qualitative features of the data. The posterior means for each participant's model parameter components were used to simulate new choice data. We summed the number of trials on which each simulated participant switched from one choice to the other. Each simulated participant corresponded to an actual online participant whose estimated parameter values were used to generate the data for that simulated participant. As in experiment 1, the actual number and the simulated number of switch trials was highly correlated across participants, see Appendix 7.

## Fitting behavioral models using alternate population parameters

To supplement the main analyses, we repeated the hierarchical modeling of task performance using the same winning model (#11) but replacing the two specific factors' scores (entered, together with scores on the general factor, at the population-level during model estimation) with residual scores on either the PSWQ scale and MASQ-AD subscale or the MASQ-AA and MASQ-AD subscale (residuals obtained after regressing out variance explained by scores on the general factor). These provided alternate indices of anxiety-specific and depression-specific affect. In particular, the analyses using the MASQ-AA subscale enable us to ensure we are not missing effects of anxiety linked to variance captured by this subscale that is not captured by the latent anxiety-specific factor extracted from the bifactor analysis (see final section of the results and *Appendix 6—figure 1* and *Appendix 6—figure 2*).

## Acknowledgements

This research was supported by grants from the European Research Community (GA 260932) and the National Institute of Mental Health (R01MH091848). PD and CG are funded by the Max Planck Society. PD is also funded by the Alexander von Humboldt foundation. The authors thank Andrea

Reinecke for assistance with Structured Clinical Diagnostic Interviews and Michael Browning, Sirius Boessenkool and Emily Witt for assistance with recruitment, experimental set up and data collection.

## Additional information

### Funding

| Funder | Grant reference number | Author |
|---|---|---|
| European Research Council | GA 260932 | Sonia J Bishop |
| National Institute of Mental Health | R01MH091848 | Sonia J Bishop |
| Alexander von Humboldt Foundation | | Peter Dayan |
| Max Planck Society | | Christopher Gagne Peter Dayan |
| National Institutes of Health | R01MH124108 | Sonia J Bishop |

The funders had no role in study design, data collection and interpretation, or the decision to submit the work for publication.

### Author contributions

Christopher Gagne, Conceptualization, Data curation, Software, Formal analysis, Visualization, Methodology, Writing - original draft, Project administration, Writing - review and editing; Ondrej Zika, Conceptualization, Data curation, Formal analysis, Methodology, Project administration, Writing - review and editing; Peter Dayan, Formal analysis, Supervision, Methodology, Writing - original draft, Writing - review and editing; Sonia J Bishop, Conceptualization, Formal analysis, Supervision, Funding acquisition, Methodology, Writing - original draft, Project administration, Writing - review and editing

### Author ORCIDs

Christopher Gagne (iD) https://orcid.org/0000-0003-2241-5285
Ondrej Zika (iD) http://orcid.org/0000-0003-0483-4443
Peter Dayan (iD) https://orcid.org/0000-0003-3476-1839
Sonia J Bishop (iD) https://orcid.org/0000-0001-7833-3030

### Ethics

Human subjects: Informed consent was obtained for all participants. Procedures for experiment 1 were approved by and complied with the guidelines of the Oxford Central University Research Ethics Committee (protocol numbers: MSD-IDREC-C2-2012-36 and MSD-IDREC-C2-2012-20). Procedures for experiment 2 were approved by and complied with the guidelines of the University of California-Berkeley Committee for the Protection of Human Subjects (protocol ID 2010-12-2638).

### Decision letter and Author response

Decision letter https://doi.org/10.7554/eLife.61387.sa1
Author response https://doi.org/10.7554/eLife.61387.sa2

## Additional files

### Supplementary files

• Transparent reporting form

## Data availability

The data and code used to create the figures and fit the hierarchical Bayesian models to the data is available on a Github repository associated with the first author and this repository has been shared on the Open Science Framework (https://osf.io/8mzuj/).

The following dataset was generated:

| Author(s) | Year | Dataset title | Dataset URL | Database and Identifier |
|---|---|---|---|---|
| Christopher G | 2020 | Impaired adaptation of learning to contingency volatility in internalizing psychopathology | https://osf.io/8mzuj | Open Science Framework, 8mzuj |

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

## Appendix 1

### Basic demographic details

In this section, we provide demographic details for participant samples for experiment 1 (n = 86), see *Appendix 1—table 1*; the confirmatory factor analysis (n = 199), see *Appendix 1—table 2*; and experiment 2 (n = 147), see *Appendix 1—table 3*.

**Appendix 1—table 1.** Basic demographic details for participants in Experiment 1.

| Participant recruitment group | Major Depressive Disorder (MDD) | Generalized Anxiety Disorder (GAD) | Healthy Controls | Unselected Community Sample |
|---|---|---|---|---|
| *Participants* Total N (N for reward task, N for aversive task) | 20 (19, 17) | 12 (10, 11) | 24 (22, 19) | 30 (29, 30) |
| *Female* | 10 (10, 8) | 11 (9, 10) | 16 (14, 13) | 13 (12, 13) |
| *Age* mean ± sd (for reward task, for aversive task) | 31 ± 10 (31 ± 10, 28 ± 10) | 32 ± 9 (31 ± 9, 32 ± 9) | 27 ± 6 (27 ± 6, 28 ± 6) | 27 ± 5 (27 ± 5, 27 ± 5) |
| *STAI* mean ± sd (for reward task, for aversive task) | 59 ± 6 (59 ± 7, 58 ± 6) | 58 ± 9 (60 ± 9, 57 ± 9) | 40 ± 12 (41 ± 12, 41 ± 12) | 36 ± 12 (36 ± 11, 36 ± 12) |
| *BDI* mean ± sd (for reward task, for aversive task) | 24 ± 9 (25 ± 9, 23 ± 9) | 20 ± 11 (22 ± 10, 20 ± 11) | 7 ± 7 (7 ± 7, 7 ± 8) | 6 ± 8 (5 ± 7, 6 ± 8) |
| *MASQ-AD* mean ± sd (for reward task, for aversive task) | 80 ± 10 (81 ± 10, 79 ± 10) | 74 ± 16 (75 ± 17, 73 ± 16) | 55 ± 18 (56 ± 18, 56 ± 19) | 50 ± 20 (48 ± 18, 50 ± 20) |
| *MASQ-AA* mean ± sd (for reward task, for aversive task) | 28 ± 7 (28 ± 7, 28 ± 7) | 33 ± 10 (34 ± 11, 34 ± 10) | 21 ± 4 (21 ± 4, 20 ± 2) | 22 ± 6 (22 ± 6, 22 ± 6) |
| *PSWQ* mean ± sd (for reward task, for aversive task) | 62 ± 14 (61 ± 14, 60 ± 14) | 76 ± 9 (75 ± 9, 75 ± 9) | 52 ± 13 (54 ± 12, 51 ± 14) | 42 ± 15 (42 ± 15, 42 ± 15) |
| *CESD* mean ± sd (for reward task, for aversive task) | 30 ± 9 (30 ± 9, 28 ± 8) | 30 ± 14 (32 ± 14, 30 ± 14) | 12 ± 8 (12 ± 8, 11 ± 8) | 10 ± 11 (9 ± 10, 10 ± 11) |
| *EPQ-N* mean ± sd (for reward task, for aversive task) | 18 ± 3 (18 ± 3, 17 ± 3) | 19 ± 4 (19 ± 3, 18 ± 4) | 10 ± 6 (11 ± 6, 10 ± 6) | 10 ± 6 (10 ± 6, 10 ± 6) |
| *General Factor* mean ± sd (for reward task, for aversive task) | 1.1 ± 0.8 (1.1 ± 0.9, 1.0 ± 0.8) | 1.3 ± 1.0 (1.5 ± 1.0, 1.4 ± 1.0) | −0.3 ± 0.8 (−0.2 ± 0.8, −0.4 ± 0.7) | −0.2 ± 0.8 (−0.3 ± 0.7, −0.2 ± 0.8) |
| *Depression-Specific Factor* mean ± sd (for reward task, for aversive task) | 0.8 ± 1.0 (0.9 ± 1.0, 0.8 ± 1.1) | −0.0 ± 0.8 (−0.1 ± 0.8, −0.1 ± 0.7) | 0.1 ± 1.2 (0.1 ± 1.2, 0.3 ± 1.2) | −0.2 ± 0.9 (−0.2 ± 0.9, −0.2 ± 0.9) |
| *Anxiety-Specific Factor* mean ± sd (for reward task, for aversive task) | −0.5 ± 1.1 (−0.5 ± 1.1, −0.6 ± 1.2) | 0.8 ± 0.9 (0.5 ± 0.8, 0.7 ± 0.9) | −0.2 ± 0.9 (−0.1 ± 0.8, −0.1 ± 0.9) | −0.3 ± 1.1 (−0.3 ± 1.1, −0.3 ± 1.1) |

STAI = Spielberger State-Trait Anxiety Inventory (form Y; *Spielberger et al., 1983*) BDI = Beck Depression Inventory (*Beck et al., 1961*) MASQ-AD/MASQ AA=anhedonic depression and anxious arousal subscales for the Mood and Anxiety Symptoms Questionnaire (*Clark and Watson, 1995*; *Watson and Clark, 1991*) PSWQ = Penn State Worry Questionnaire (*Meyer et al., 1990*) CESD = Center for Epidemiologic Studies Depression Scale (*Radloff, 1977*) EPQ-N = the Neuroticism subscale for the 80-item Eysenck Personality Questionnaire (*Eysenck and Eysenck, 1975*). The two healthy control participants whose data were excluded from both tasks are omitted from this table.

**Appendix 1—table 2.** Basic demographic details of participants who provided questionnaire data for the confirmatory factor analysis.

| **Number of participants (Total N)** | **199** |
| --- | --- |
| *Female* (N) | 120 |
| *Age* (mean ± sd) | 21 ± 4 |
| *STAI* (mean ± sd) | 44 ± 9 |
| *BDI* (mean ± sd) | 7 ± 6 |
| *MASQ-AD* (mean ± sd) | 54 ± 15 |
| *MASQ-AA* (mean ± sd) | 24 ± 8 |
| *PSWQ* (mean ± sd) | 57 ± 13 |
| *CESD* (mean ± sd) | 24 ± 8 |
| *EPQ-N* (mean ± sd) | 6 ± 4 |
| *General Factor* (mean ± sd) | −0.1 ± 1.0 |
| *Depression-Specific Factor* (mean ± sd) | 0.4 ± 1.0 |
| *Anxiety-Specific Factor* (mean ± sd) | −0.1 ± 1.0 |

STAI = Spielberger State-Trait Anxiety Inventory (form Y; *Spielberger et al., 1983*) BDI = Beck Depression Inventory (*Beck et al., 1961*) MASQ-AD/MASQ AA=anhedonic depression and anxious arousal subscales for the Mood and Anxiety Symptoms Questionnaire (*Clark and Watson, 1995*; *Watson and Clark, 1991*) PSWQ = Penn State Worry Questionnaire (*Meyer et al., 1990*) CESD = Center for Epidemiologic Studies Depression Scale (*Radloff, 1977*) EPQ-N = the Neuroticism subscale for the 80-item Eysenck Personality Questionnaire (*Eysenck and Eysenck, 1975*).

**Appendix 1—table 3.** Basic demographic details for participants in Experiment 2.

| **Participants (Total N)** | **147** |
| --- | --- |
| *Female* (N) | 64 |
| *Age* | Not recorded; required to be 18 years or older. |
| *STAI* (mean ± sd) | 43 ± 13 |
| *BDI* (mean ± sd) | 11 ± 12 |
| *MASQ-AD* (mean ± sd) | 63 ± 18 |
| *MASQ-AA* (mean ± sd) | 23 ± 8 |
| *General Factor* (mean ± sd) | −0.1 ± 0.9 |
| *Depression-Specific Factor* (mean ± sd) | −0.2 ± 0.9 |
| *Anxiety-Specific Factor* (mean ± sd) | 0.1 ± 1.0 |

STAI = Spielberger State-Trait Anxiety Inventory (form Y; *Spielberger et al., 1983*); BDI = Beck Depression Inventory (*Beck et al., 1961*) MASQ-AD/MASQ AA=anhedonic depression and anxious arousal subscales for the Mood and Anxiety Symptoms Questionnaire (*Clark and Watson, 1995*; *Watson and Clark, 1991*).

## Appendix 2

### Additional factor analyses

We compared the fit of the bifactor model, as assessed by confirmatory factor analysis in a novel dataset (n = 199), with that of three other models: a two-correlated factors models, a general factor only model, and a higher order factor model. Quality of fit was determined by the comparative fit index (CFI) and the root mean square error of approximation (RMSEA). These analyses were conducted using the Lavaan package in R.

As with the bifactor model, we conducted confirmatory factor analyses for each of these alternate models using the CFA dataset (n = 199). As for the bifactor model CFA, this entailed specifying the assignment of items to factors and then re-estimating the loadings.

For the two-correlated factors model, item-to-factor assignments were obtained by first using exploratory factor analysis (EFA) to fit this model to the data from experiment 1. The exploratory fitting procedure was the same as the first stage of the main bifactor analysis, that is polychoric correlations were calculated and an oblique rotation (i.e. oblimin) was applied. Examination of factor loadings revealed that one factor loaded predominantly onto anxiety-related items and the other factor loaded predominantly onto depression-specific items. The two factors were strongly correlated (r = 0.5), reflecting the high degree of correlation (r > 0.52) amongst depression- and anxiety-related questionnaires (e.g. STAI, BDI, etc.) in our dataset. Items with loadings < 0.2 on both specific factors in the exploratory analysis were fixed to have loadings equal to zero for confirmatory analysis. Items with loadings > 0.2 were then re-estimated during the confirmatory analysis using diagonally weighted least squares estimation. This is same procedure carried out for the main bifactor analysis (see Materials and methods: Confirmatory Bifactor Analysis).

The only difference between the two-correlated factors model and the bifactor model in the exploratory factor analysis is that the latter involves Schmid-Leiman orthogonalization following the oblique rotation in order to partition shared versus unique variance. Because this orthogonalization does not change the flexibility of the model (*Reise, 2012*), the two models fit equally well in the dataset they were estimated on (i.e. the experiment 1 dataset). Comparing these models using confirmatory factor analysis in the independent online sample (n = 199), showed a better fit for the bifactor model (root-mean-square error of approximation RMSEA = 0.065) than for the correlated factors model (RMSEA = 0.083).

For the general factor only model, item assignments were taken directly from the main bifactor model (as fit in the EFA conducted on questionnaire data from experiment 1), simply excluding assignments for the specific factors. Conversely, the higher order factor model used the same specific-factor item assignments as the bifactor model (as fit in the EFA) but excluded general factor assignments; here, a higher order factor was allowed to explain correlations between the two specific factors but was not allowed to load directly onto items themselves. In the independent online sample used for the CFA (n = 199), the full bifactor model showed a better fit (RMSEA = 0.065) than either the general factor only model (RMSEA = 0.110), or the higher order three factor model (RMSEA = 0.084).

# Appendix 3

## Model comparison tables

In this section, we include summaries of the model comparison results for experiment 1 (*Appendix 3—table 1*) and experiment 2 (*Appendix 3—table 2*); for a full description of the models compared and the comparison procedure see Materials and methods: Stage-wise Model Construction.

**Appendix 3—table 1.** Model comparison table for Experiment 1.

Thirteen models were fit to participants' choice data from experiment 1. Models were fit hierarchically and compared using leave-one-out cross validation error approximated by Pareto smoothed importance sampling (PSIS-LOO; values shown in right-most column). The model with the lowest PSIS-LOO was selected as the winning model and used to make inferences about the relationships between task performance and internalizing symptoms.

| Model Number | Parameters | # of Parameter Components | PSIS-LOO |
|---|---|---|---|
| Model #1 | $\alpha,\ \gamma, \omega^1$ | 12 | 27,801 |
| Model #2 | $\alpha,\ \lambda, \omega$ | 12 | 26,164 |
| Model #3 | $\alpha^{gb},\ \lambda, \omega$ | 15 | 25,550 |
| Model #4 | $\alpha,\ \lambda^{gb}, \omega^{gb}$ | 18 | 26,042 |
| Model #5 | $\alpha^{gb},\ \lambda^{gb}, \omega^{gb}$ | 21 | 25,462 |
| Model #6 | $\alpha^{gb},\ \lambda^{gb}, \omega^{gb}$ | 24 | 25,486 |
| Model #7 | $\alpha^{gb},\ \lambda^{gb}, \omega^{gb},\ r^{ra\ only}$ | 23 | 25,154 |
| Model #8 | $\alpha^{gb},\ \lambda^{gb}, \omega^{gb},\ r^{ra\ only},\ \epsilon^{ra\ only}$ | 25 | 25,185 |
| Model #9 | $\alpha^{gb},\ \lambda^{gb}, \omega^{gb},\ r^{ra\ only},\ \delta$ | 27 | 25,377 |
| Model #10 | $\alpha^{gb},\ \lambda^{gb}, \omega^{gb},\ r^{ra\ only},\ \delta$ | 27 | 25,325 |
| Model #11 ** | $\alpha^{gb},\ \lambda^{gb}, \omega_v^{gb}, r^{ra\ only},\ \omega_k^{ra\ only},\ \eta^{baseline}$ | **26** | **25,037** |
| Model #12 | $\alpha^{gb},\ \lambda^{gb}, \omega_v^{gb}, r^{ra\ only},\ \omega_k^{ra\ only},\ \eta^{baseline},\ \delta$ | 32 | 25,216 |
| Model #13 | $\alpha^{gb},\ \lambda^{gb}, \omega_v^{gb}, r^{ra\ only},\ \omega_k^{ra\ only},\ \eta^{baseline},\ \delta$ | 32 | 25,181 |

[1]:Unless otherwise stated, each parameter is divided into four parameter components: a shared baseline parameter across blocks and tasks, and differences in the parameter between stable and volatile blocks (volatile-stable), between different task versions (reward-aversive) and an interaction of those differences (reward-aversive)x(volatile-stable).

gb: For each parameter with this superscript, three additional parameter components were added for the relative value of previous outcome (good-bad) and the interactions of relative outcome value with block type (volatile-stable)x(good-bad) and task version (reward-aversive)x(good-bad).

ra only: For each parameter with this superscript, only differences in the parameters between the reward and aversive task versions (reward-aversive) were included.

baseline: For each parameter with this superscript, only one single baseline parameter was used, across both task versions and volatile and stable blocks.

**Indicates best fitting model.

**Appendix 3—table 2.** Model comparison table for Experiment 2.

The same 13 models fit to participants' choice data from experiment 1 were also fit to participants' choice data from experiment 2. Models were fit hierarchically and compared using leave-one-out cross validation error approximated by Pareto smoothed importance sampling (PSIS-LOO). Model #12 (which differs from model #11 only in the use of separate learning rates for each shape) had a slightly lower PSIS-LOO than model #11. However, the difference in PSIS-LOO for Models #11 and #12 was within one standard error (difference in PSIS-LOO = 43; SE = 49). In contrast, model #11's PSIS-LOO was more than two standard errors better than model #12 in experiment 1 (difference in

PSIS-LOO = 179, SE = 71). Hence, if we seek to retain one model across both experiments, model #11 is the better choice.

| Model Number | Parameters | # of Parameter Components | PSIS-LOO |
|---|---|---|---|
| Model #1 | $\alpha,\ \gamma, \omega$[1] | 12 | 57,520 |
| Model #2 | $\alpha,\ \lambda, \omega$ | 12 | 54,002 |
| Model #3 | $\alpha^{gb},\ \lambda, \omega$ | 15 | 52,918 |
| Model #4 | $\alpha,\ \lambda^{gb}, \omega^{gb}$ | 18 | 53,755 |
| Model #5 | $\alpha^{gb},\ \lambda^{gb}, \omega^{gb}$ | 21 | 52,758 |
| Model #6 | $\alpha^{gb},\ \lambda^{gb}, \omega^{gb}$ | 24 | 52,769 |
| Model #7 | $\alpha^{gb},\ \lambda^{gb}, \omega^{gb},\ r^{gl\ only}$ | 23 | 52,139 |
| Model #8 | $\alpha^{gb},\ \lambda^{gb}, \omega^{gb},\ r^{gl\ only},\ \epsilon^{gl\ only}$ | 25 | 52,136 |
| Model #9 | $\alpha^{gb},\ \lambda^{gb}, \omega^{gb},\ r^{gl\ only},\ \delta$ | 27 | 52,083 |
| Model #10 | $\alpha^{gb},\ \lambda^{gb}, \omega^{gb},\ r^{gl\ only},\ \delta$ | 27 | 52,169 |
| Model #11 | $\alpha^{gb},\ \lambda^{gb}, \omega_v^{gb}, r^{gl\ only},\ \omega_k^{gl\ only},\ \eta^{baseline}$ | 26 | 52,048 |
| Model #12 | $\alpha^{gb},\ \lambda^{gb}, \omega_v^{gb}, r^{gl\ only},\ \omega_k^{gl\ only},\ \eta^{baseline},\ \delta$ | 32 | 52,005 |
| Model #13 | $\alpha^{gb},\ \lambda^{gb}, \omega_v^{gb}, r^{gl\ only},\ \omega_k^{gl\ only},\ \eta^{baseline},\ \delta$ | 32 | 52,084 |

[1]:Unless otherwise stated, each parameter is divided into four parameter components: a shared baseline parameter across blocks and tasks, and differences in the parameter between stable and volatile blocks (volatile-stable), between different task versions (gain-loss) and an interaction of those differences (gain-loss)x(volatile-stable).

gb: For each parameter with this superscript, three additional parameter components were added for the relative value of previous outcome (good-bad) and the interactions of this difference with block type (volatile-stable)x(good-bad), and task version (gain-loss)x(good-bad). gl only: For each parameter with this superscript, only differences in the parameters between the reward gain and reward loss task versions (gain-loss) were included.

baseline: For each parameter with this superscript, only one single baseline parameter was used, across both task versions and volatile and stable blocks.

## Appendix 4

### Model parameter recovery results

Using estimated parameter values for the winning model (model #11), we simulated 10 new datasets for each of Experiment 1's participants (n = 86), see Materials and methods. By fitting the model to these simulated datasets, we can compare the ground truth parameter values with the parameters recovered from the simulated data. The recovered parameters strongly correlated with the ground truth parameter values. For learning rate components, the mean correlation was r = 0.88 (std = 0.13) across all simulated datasets, see *Appendix 4—figure 1* for an example dataset. Across all parameters, the mean correlation across simulated datasets was r = 0.76 (std = 0.15), see *Appendix 4—figure 2* for an example dataset. These analyses indicates that individual model parameters were recoverable as desired.

We also examined the robustness of our estimates for population-level parameters ($\mu, \beta_g, \beta_a, \beta_d$) by looking at the variability in their values across the simulated datasets. Variability across datasets reflects sensitivity to noise in participants' choices, which is estimated by the fitted values for the two inverse temperatures in the model. The population-level parameters for each simulated dataset are shown in *Appendix 4—figure 3* for learning rate and in *Appendix 4—figure 4* for the other model parameters.

As described in the main text, in Experiment 1, we also fit an additional model that parametrized the three-way interaction of block type (volatile, stable), relative outcome value (good, bad), and task version (reward, aversive) for learning rate. We conducted a parameter recovery analysis for this model that paralleled that described above (see also Materials and methods). This confirmed that the three way interaction of block type, relative outcome value, and task type on learning rate could be successfully recovered. The average correlation between ground truth and recovered parameters values for this triple interaction, across 10 simulated datasets, was r = 0.86 (std = 0.10). Parameter recovery results for all learning rate components from an example simulated dataset are given in *Appendix 4—figure 5*.

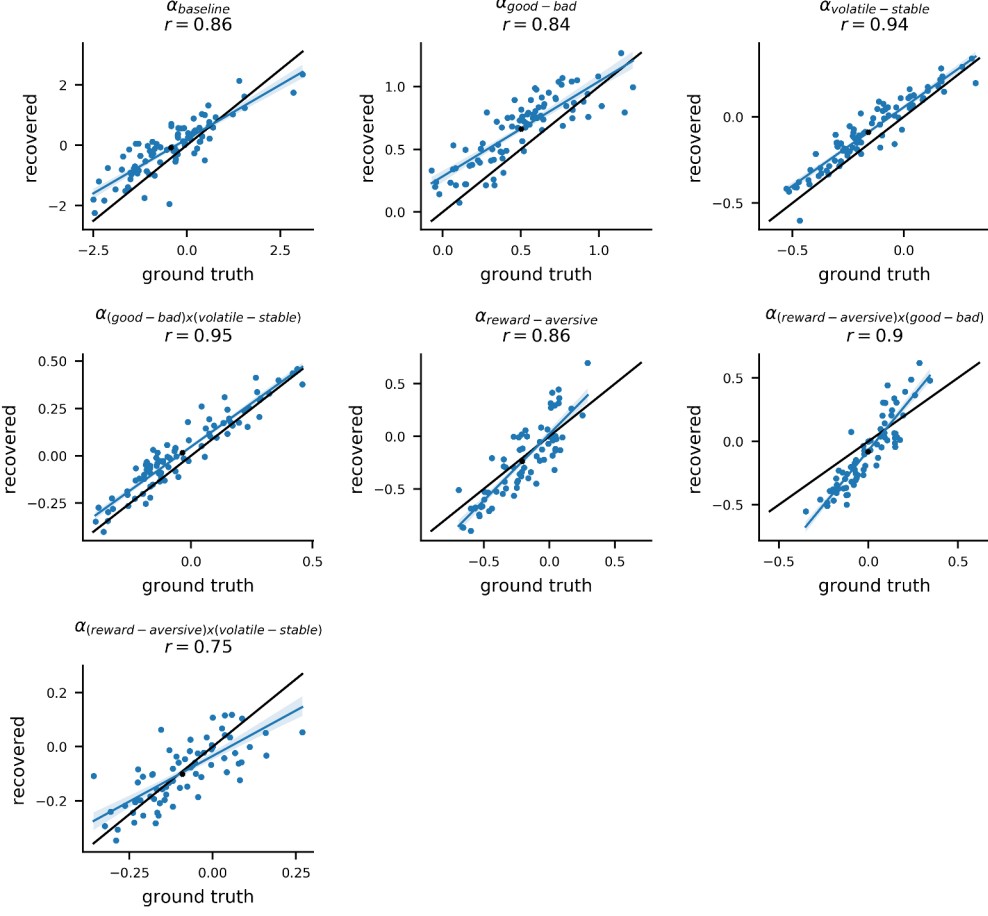

**Appendix 4—figure 1.** Recovery of individual-level learning rate parameters. Posterior mean parameter estimates for participants from experiment 1 were used to simulate new choice data from the winning model (#11). The model was then re-fit to each of these simulated datasets. The original parameters estimated from the actual dataset (referred to as 'ground truth' parameters) were correlated with the newly estimated parameters (referred to as 'recovered' parameters) for each simulated dataset. An example dataset is shown here. Each panel shows the ground truth and recovered posterior means for a separate component of the composite learning rate parameter. The x-axis corresponds to original 'ground truth' parameter values and the y-axis corresponds to the recovered parameter values; each datapoint represents an individual participant. The average correlation between ground truth and recovered parameters values for learning rate components, across 10 simulated datasets, was r = 0.88 (std = 0.13).

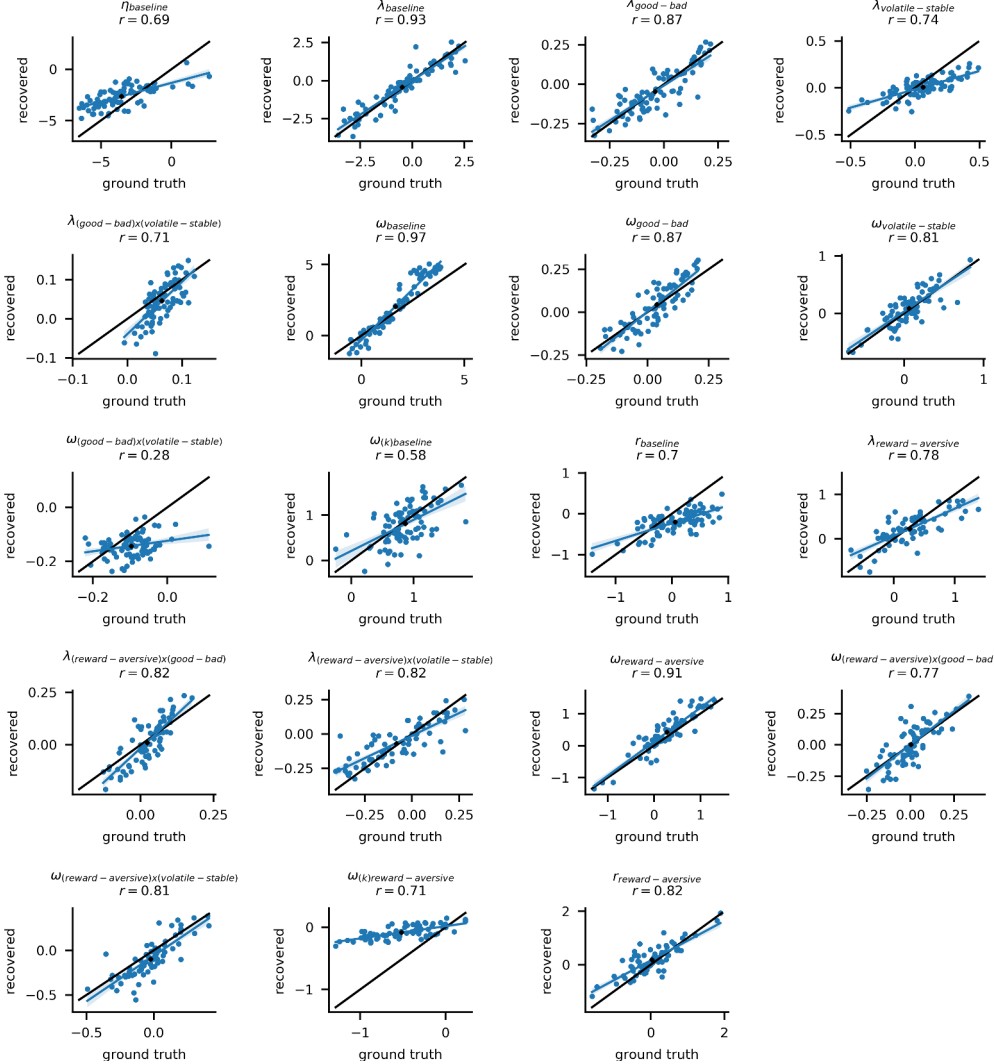

**Appendix 4—figure 2.** Recovery of other model parameters. Posterior mean parameter estimates for participants from experiment 1 were used to simulate new choice data from the winning model (#11). As in *Appendix 4—figure 1*, we show the results of parameter recovery for one example simulated dataset; here, we present data for parameters other than learning rate. Each panel shows the ground truth and recovered posterior means for a separate model parameter component. The x-axis corresponds to original 'ground truth' parameter values and the y-axis corresponds to the recovered parameter values; each datapoint represents an individual participant. The average correlation between ground truth and recovered parameters values across the 10 datasets for other (non-learning rate) parameters was r = 0.76 (std = 0.15).

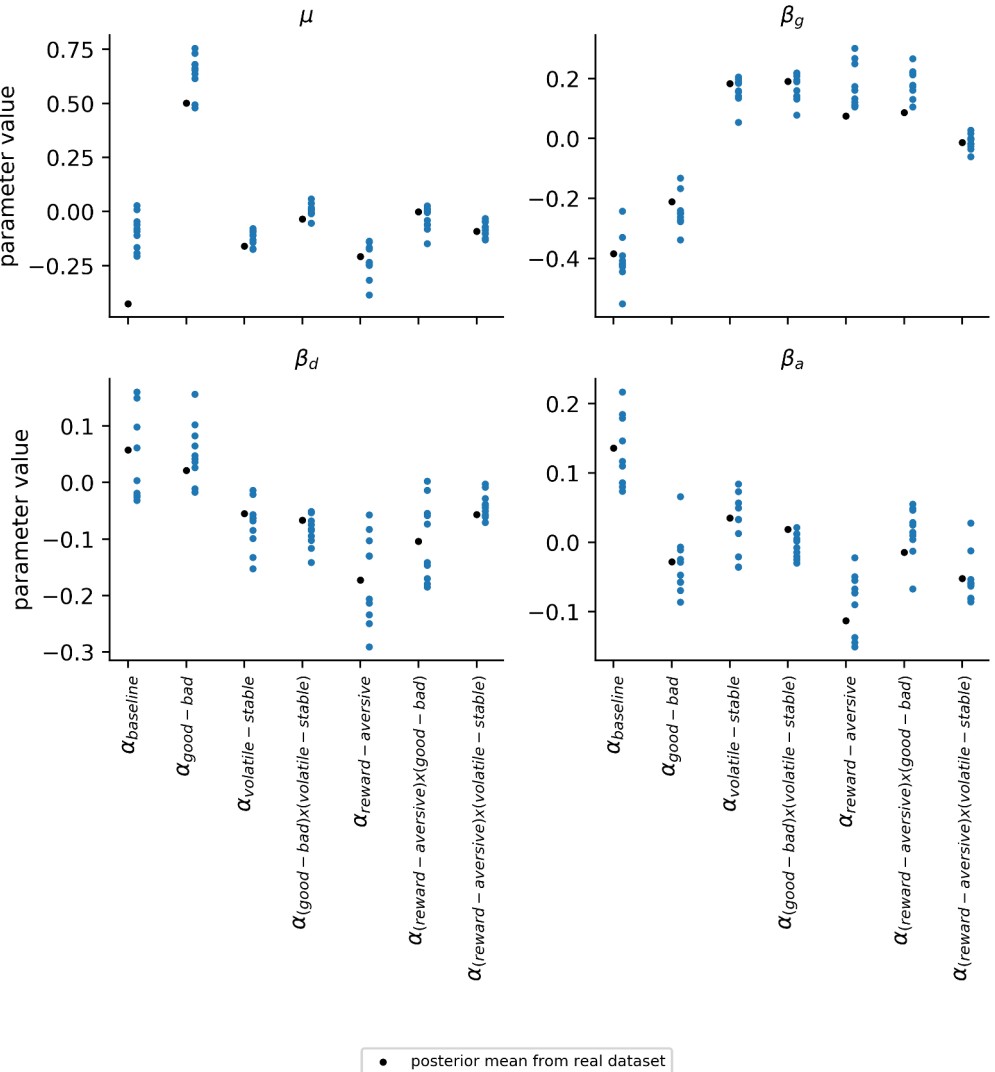

**Appendix 4—figure 3.** Variability of population-level learning rate parameters across simulated datasets. The robustness of the estimates for the population-level parameters ($\mu$, $\beta_g$, $\beta_a$, $\beta_d$) was explored by examining the variability in parameter values across the 10 simulated datasets (blue data points). Population-level parameters corresponding to learning rate components are shown in this figure. The simulated datasets used for this analysis were the same as those used for *Appendix 4—figure 1* and *Appendix 4—figure 2*. Since each of the 10 simulated datasets uses the same ground-truth parameters for generating data (black data points), differences across these datasets reflect an estimate for the amount of noise in participants' choices; this choice noisiness is captured by the two inverse temperatures in the model. Consistency across datasets and proximity to the ground-truth parameters indicates a robustness to this type of noise.

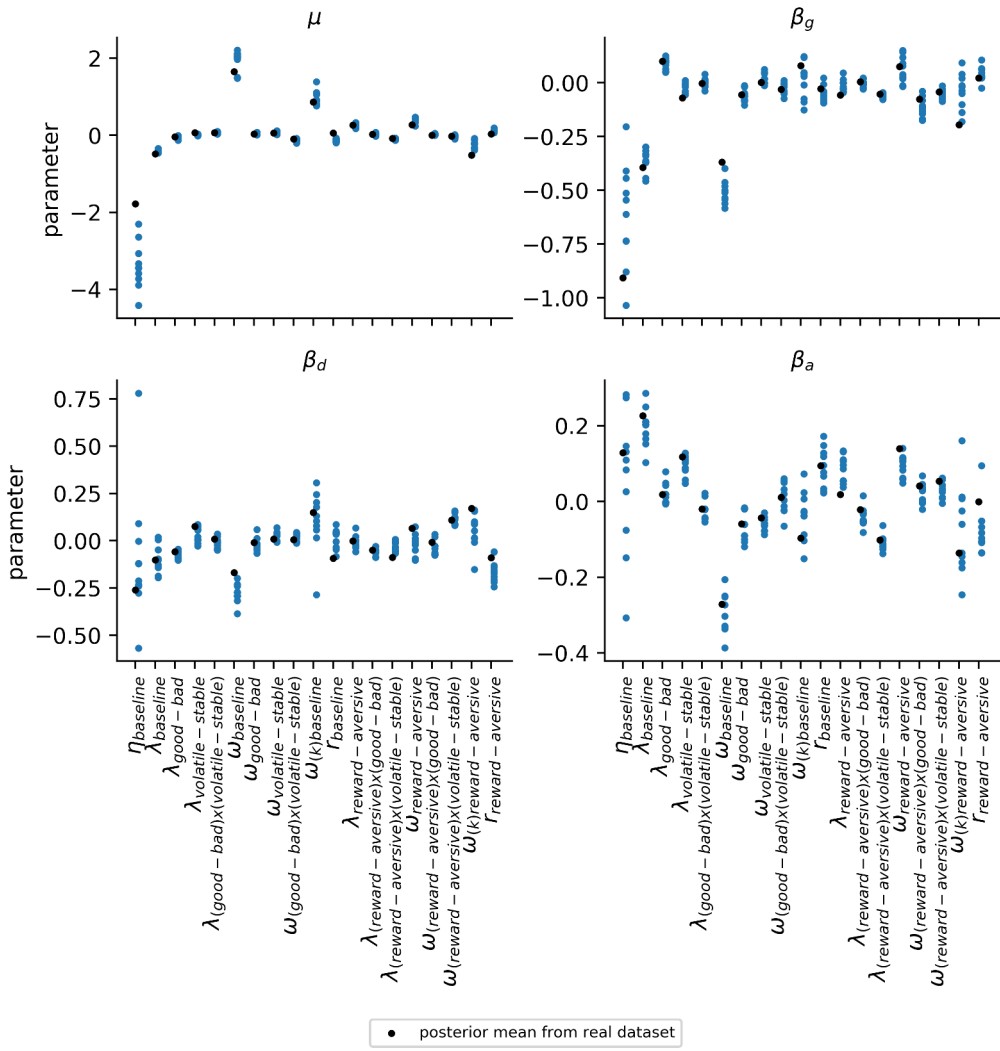

**Appendix 4—figure 4.** Variability of population-level parameters across simulated datasets for other parameters. The robustness of the estimates for the population-level parameters ($\mu$, $\beta_g$, $\beta_a$, $\beta_d$) was explored by examining the variability in parameter values across the 10 simulated datasets (blue data points). Population-level parameters corresponding to all other parameter components aside from those for learning rate are shown in this figure. The simulated datasets used for this analysis were the same as those used for *Appendix 4—figure 1* and *Appendix 4—figure 2*. Since each of the 10 simulated datasets uses the same ground-truth parameters for generating data (black data points), differences across these datasets reflect an estimate for the amount of noise in participants' choices; this choice noisiness is captured by the two inverse temperatures in the model. Consistency across datasets and proximity to the ground-truth parameters indicates a robustness to this type of noise. Apart from the baseline component of each parameter, simulated parameter component ranges are relatively narrow and predominantly encompass the parameter values estimated from the actual dataset (black data points).

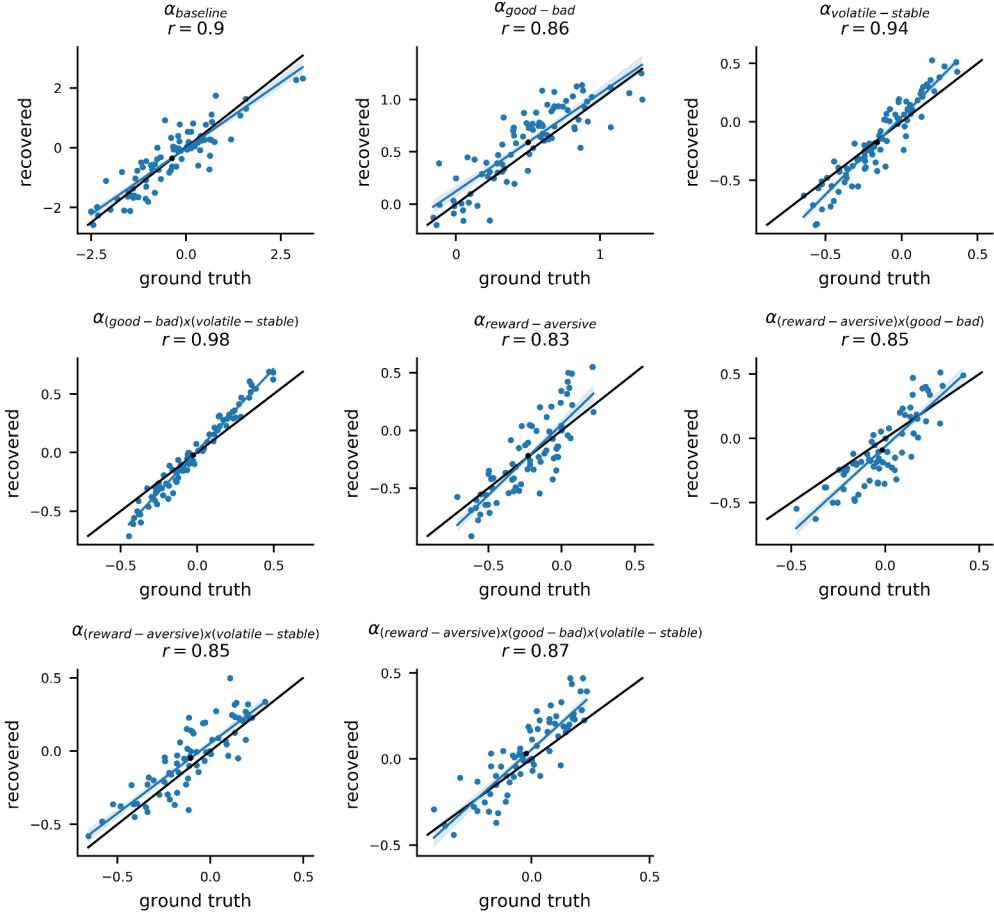

**Appendix 4—figure 5.** Recovery of individual-level learning rate parameters in the thee-way interaction model. In experiment 1, we additionally fit a model that included the three-way interaction of block type (volatile, stable), relative outcome value (good, bad), and task version (reward, aversive) for learning rate. This model was identical to the winning model (#11) except for the inclusion of the three-way interaction and was used to confirm that the relationship between general factor scores and the interaction of block type by relative outcome value on learning rate did not vary as a function of task version. Posterior means for each participants' model parameters were used to simulate new choice data from the model. The model was then re-fit to each of these simulated datasets. The original parameters estimated from the actual dataset (referred to as 'ground truth' parameters) were correlated with the newly estimated parameters (referred to as 'recovered' parameters) for each simulated dataset. An example dataset is shown here. Each panel shows the ground truth and recovered posterior means for a separate component of the composite learning rate parameter. The x-axis corresponds to original ground truth parameter values and the y-axis corresponds to the recovered parameter values; each datapoint represents an individual participant. The high correlation between the ground truth and recovered parameters was high, even for the triple interaction (bottom right panel), indicating good parameter recoverability. The average correlation between ground truth and recovered parameters values for this triple interaction, across 10 simulated datasets, was r = 0.86 (std = 0.10).

# Appendix 5

## Task performance

In this section, we examine general task performance across experiment 1 and experiment 2. This enables us to compare the general level of performance in the reward gain version of the task across both experiments and to compare the general level of performance in the aversive (shock) version of the task (as conducted in experiment 1) to that in the reward loss version of the task (as conducted in experiment 2). Performance was measured using the average outcome magnitude received across trials, that is, the mean reward gained or lost and the mean level of electrical stimulation (shock) received. Higher average magnitudes indicate better performance in the reward gain version of the task; lower average magnitudes indicate better performance in the reward loss and aversive (shock) versions of the task. If anything, general level of performance was closer between the two punishment versions of the task: aversive (shock), reward loss than between the two reward gain versions of the task.

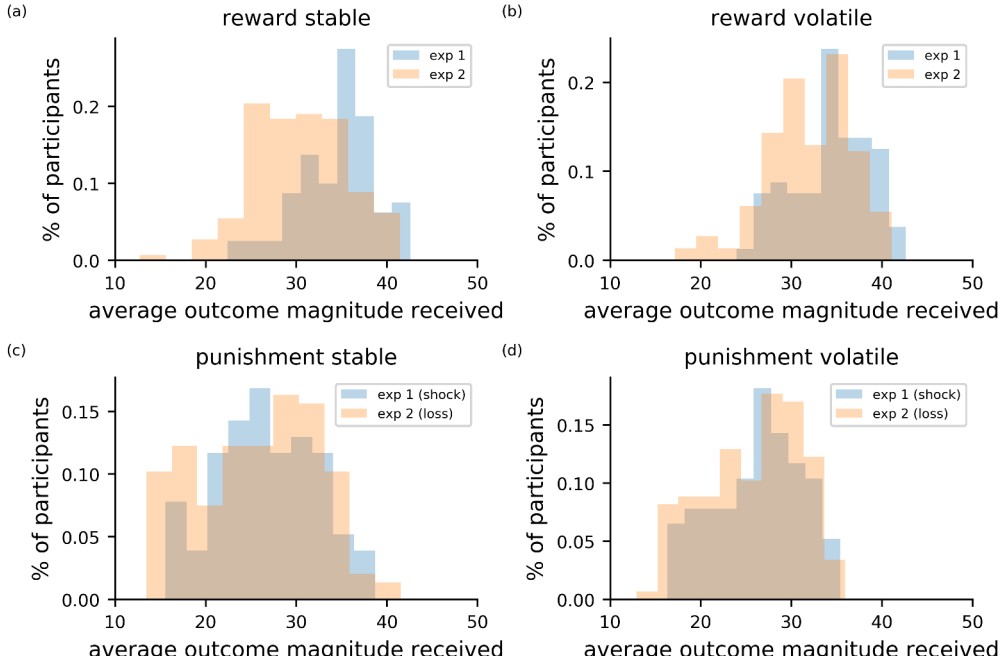

**Appendix 5—figure 1.** Comparison of task performance between experiment 1 and experiment 2. The four panels depict the performance of participants in each block (stable left column; volatile right column) and in each task (reward top row; punishment bottom row). Data from experiment 1 is shown in blue; data from experiment 2 is shown in orange. To assess performance, the magnitudes of outcomes received were averaged across trials. Higher average magnitudes for the reward condition indicates better performance. Higher average magnitudes for the loss and shock outcomes indicates worse performance.

## Appendix 6

## Alternate modeling of anxiety- and depression-specific effects upon behavior

As detailed in the main manuscript, the depression-specific factor was strongly correlated with residual scores on the MASQ-AD (after regressing out variance explained by scores on the general factor) and the anxiety-specific factor was strongly correlated with residual scores on the PSWQ (created in an equivalent fashion, see *Figure 1—figure supplement 2*). Using residual scores on these two scales together with scores on the general factor provides a way to approximate the bifactor model that is arguably more interpretable for clinicians who are familiar with these measures. Accordingly, we conducted an alternative hierarchical model of participants' task performance in experiment 1, entering individual scores on the general factor along with residual scores on the MASQ-AD and PSWQ into the population-level prior distributions for model parameters (i.e. as population-level covariates). As expected, this produced extremely similar results to the main bifactor model. The relationship between general factor scores and the effect of volatility and volatility by relative outcome value on learning rate remained statistically credible. No credible effects on learning rate were observed for residual MASQ-AD or PSWQ scores (*Appendix 6—figure 1*).

We used a similar approach to investigate if symptoms uniquely linked to the anxious arousal scale of the MASQ (MASQ-AA) and not captured by the general factor might explain variance in participants' behavior. Scores on the MASQ-AA were not strongly correlated with the anxiety-specific factor from the bifactor analysis and nearly half the variance in these scores remained unexplained after regressing out variance explained by the general factor. Hence the proposed analysis enables us to ensure that we are not missing a relationship between anxiety and learning rate that might be linked to symptoms uniquely captured by the MASQ-AA subscale. We hence fitted a hierarchical model to participants' task performance with participants scores on the general factor and residual scores on the MASQ-AA and the MASQ-AD (after variance explained by scores on the general factor were regressed out) entered as population-level covariates. We modeled data from both experiment 1 and experiment 2 in this manner as the MASQ was given in both cases. Both analyses replicated the general factor effects reported above (for experiment 1 see *Appendix 6—figure 1*, for experiment 2 see *Appendix 6—figure 2*). Residual scores on the MASQ-AA subscale did not credibly influence adaptation of learning rate to volatility or the effect on learning rate of the interaction of block type (volatile, stable) by relative outcome value or task version. There was also no relationship between MASQ-AA scores and differential learning as a function of task type (aversive versus reward gain in experiment 1 and reward loss versus reward gain in experiment 2). In experiment 2, elevated residual scores on the MASQ-AA scores were associated with greater learning following outcomes of relative positive versus negative value (*Appendix 6—figure 2*). However, no equivalent relationship was observed in experiment 1 (*Appendix 6—figure 1*). We hence caution against weight being given to this observation prior to any replication of this result.

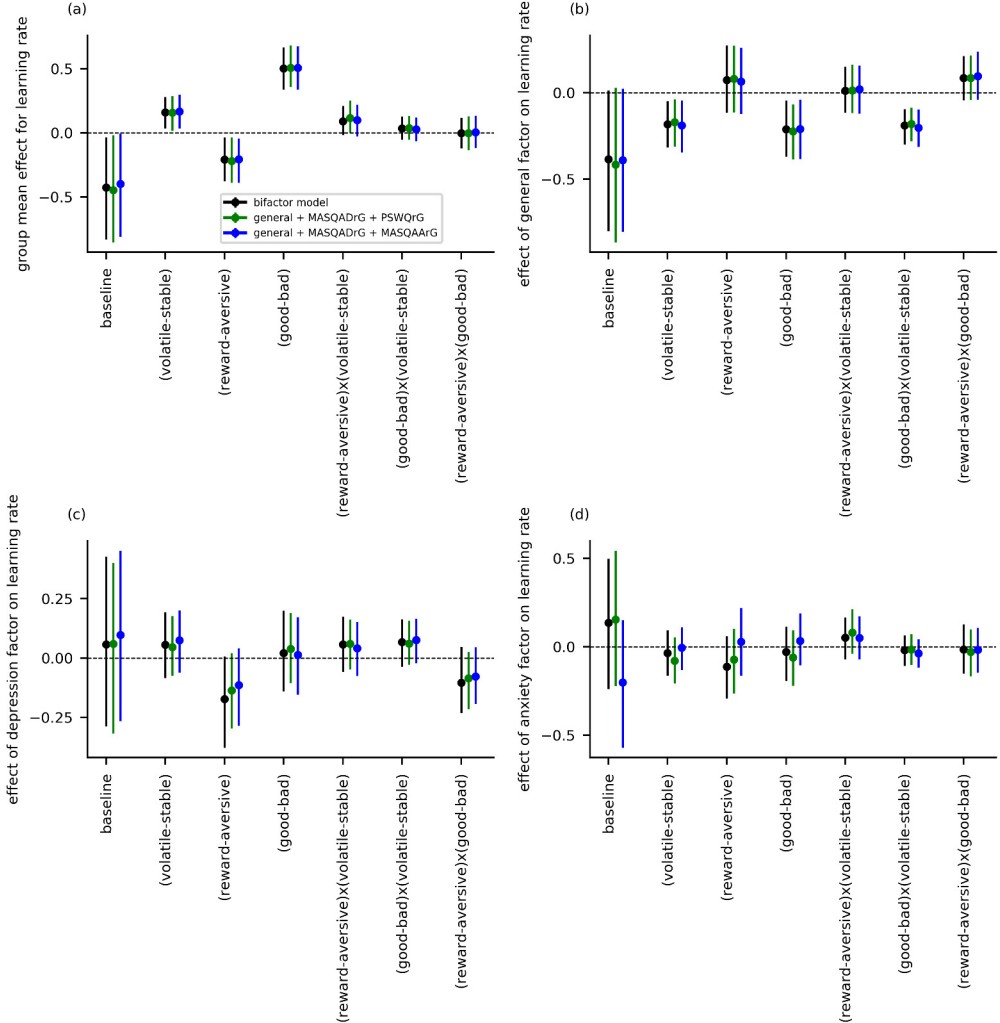

**Appendix 6—figure 1.** Learning rate parameters for experiment 1 data as estimated using alternate population-level parameters for specific effects of anxiety and depression. Two alternative models were fit to the behavioral data from experiment 1, in addition to the main bifactor model. For the first alternative model, population-level parameters entered comprised scores on the general factor and residual scores on the MASQ anhedonia subscale and the Penn-State Worry Questionnaire (PSWQ). These residual scores were created by removing variance from scores for the MASQ and PSWQ explainable by scores for the general factor; as such these scores provide alternative depression-specific and anxiety-specific symptom measures. This model is abbreviated as 'general + MASQADrG + PSWQrG'. For the second alternative model, residual PSWQ scores were replaced by residual scores for the MASQ anxious arousal subscale. This enables us to investigate whether anxiety-related symptoms uniquely captured by the MASQ-AA influence learning rate. This model is abbreviated as 'general + MASQADrG + MASQAArG'. The main model is labeled simply as 'bifactor model'. Both alternative models yielded general factor learning rate effects that were consistent with the main model (panel b). No additional effects were observed for the depression or anxiety subscales (panels c-d).

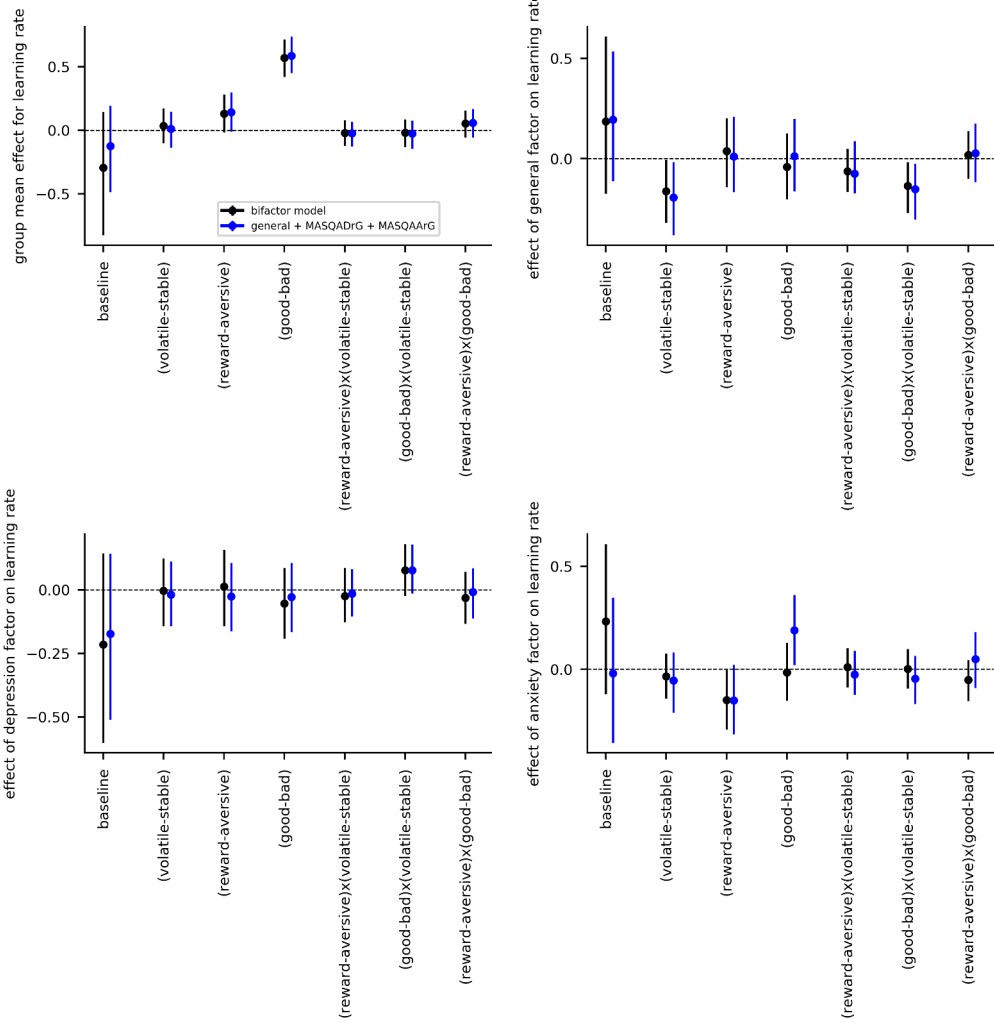

**Appendix 6—figure 2.** Learning rate parameters for experiment 2 data as estimated using alternate population-level parameters for specific effects of anxiety and depression. In addition to the main bifactor model, an additional alternative model was also fit to the behavioral data from experiment 2. In this model (the second alternate model described in *Appendix 6—figure 1*), population-level parameters entered comprised scores on the general factor and residual scores on the MASQ anhedonia subscale and the MASQ anxious arousal subscale (having regressed out variance explainable by general factor scores). This model is abbreviated as 'general + MASQADrG + MASQAArG'. As in *Appendix 6—figure 1*, the main model is labeled simply as 'bifactor model'. The alternative model yielded general factor learning rate effects that were consistent with the main model (panel b). No additional effects were observed for residual scores on the MASQ-AD subscale (panel c). Elevated residual scores on the MASQ-AA subscale were linked to increased learning after outcomes of relative positive value (good - bad) but did not modulate adaptation of learning rate to volatility or the interaction of volatility and relative outcome value (good - bad). We note, no equivalent findings were observed for MASQ-AA in experiment 1 (see *Appendix 6—figure 1*). We could not fit the 'general + MASQADrG + PSWQ' model also described in *Appendix 6—figure 1* to this dataset as participants were not administered the PSWQ questionnaire.

## Appendix 7

### Model reproduction of basic features of choice behavior

Data were simulated using the winning model (#11) to check that it could reproduce basic qualitative features of participants' actual choice behavior in experiment 1 and experiment 2.

The number of trials on which a participant stays with the same choice or switches to the other choice is a qualitative feature that our model should be able to reproduce even though it was not optimized to do so. Using the datasets simulated for parameter recovery analyses, we summed the number of trials on which each simulated participant switched from choosing one shape to the other (see Materials and methods). Each simulated participant corresponded to an actual participant whose estimated parameter values were used to generate the data for that simulated participant; we next summed actual switch trials. The actual number and the simulated number of switch trials was highly correlated across participants (r(84)>0.88 for all conditions and datasets). We give an example dataset in *Appendix 7—figure 1*. We repeated this analysis for experiment 2. Here again, the actual number and the simulated number of switch trials was highly correlated across participants (r(145)>0.82 for all conditions and datasets); see *Appendix 7—figure 2* for an example dataset.

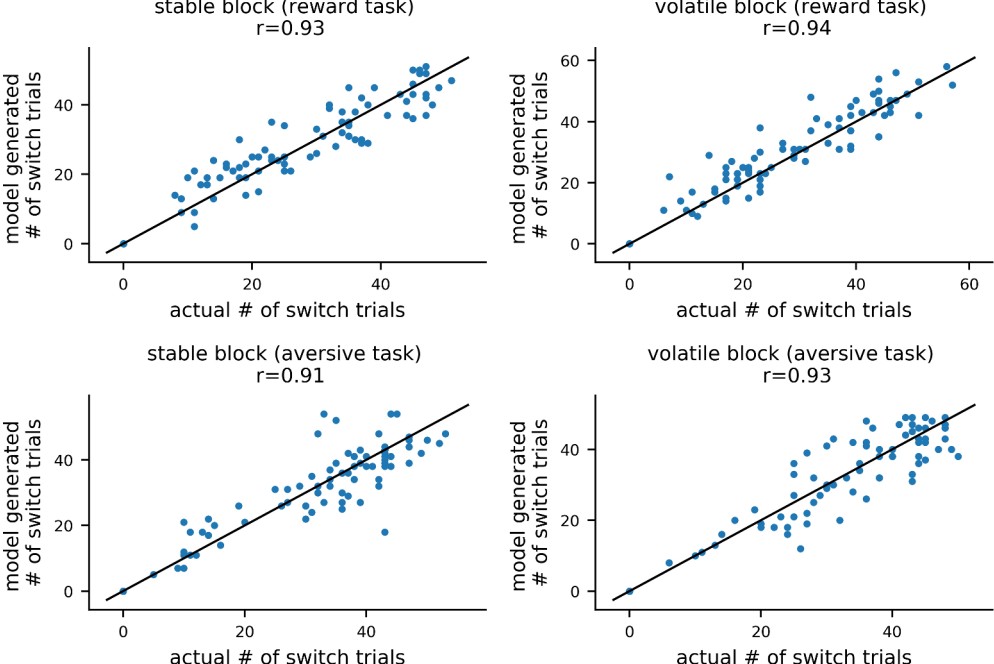

**Appendix 7—figure 1.** Comparison of actual and model generated numbers of switch trials in experiment 1. For each participant, we calculated the number of trials on which they switched choice of shape. As described under parameter recovery, each participant's posterior means for each of model #11 parameters were used together with model #11 to simulate 10 new datasets. For each of these simulated datasets, the number of switch trials was computed and correlated with the actual number of switch trial for the corresponding participant. This is shown here for an example dataset, with switch trials for each combination of task version (reward, aversive) and block type (volatile, stable) shown in a separate panel. Mean correlations between actual and generated switch trials were high (rs >0.88 across the 4 conditions and 10 datasets), demonstrating that the model can reproduce a basic qualitative feature of participants' choice behavior.

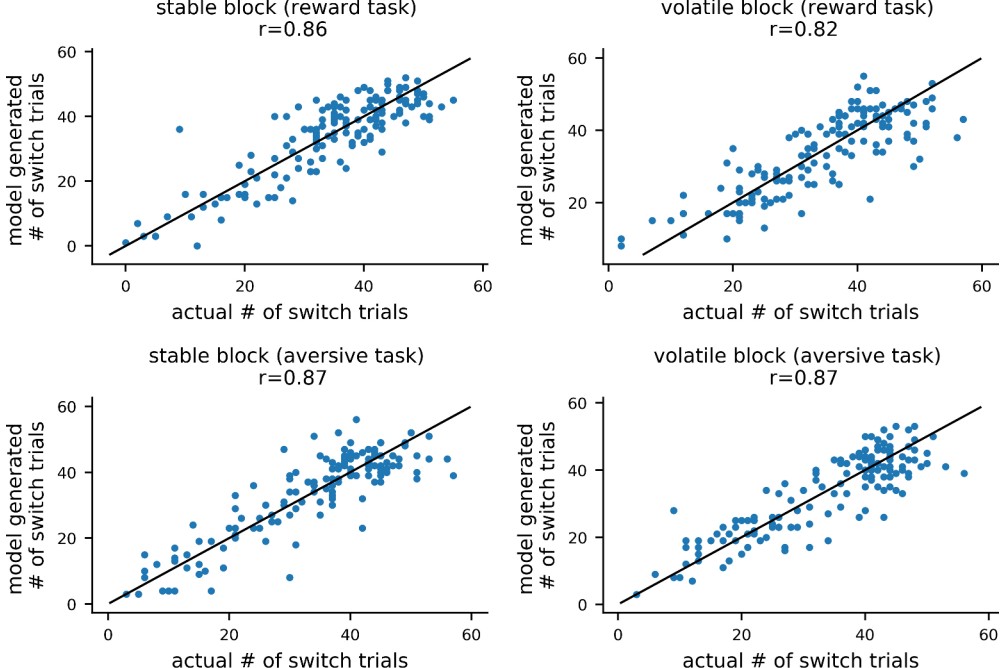

**Appendix 7—figure 2.** Comparison of actual and model generated numbers of switch trials in experiment 2. For each participant, we calculated the number of trials on which they switched choice of shape. As described under parameter recovery, each participant's posterior means for each of model #11 parameters were used together with model #11 to simulate 10 new datasets. For each of these simulated datasets, the number of switch trials was computed and correlated with the actual number of switch trial for the corresponding participant. This is shown here for an example dataset, with switch trials for each combination of task version (reward, aversive) and block type (volatile, stable) shown in a separate panel. Mean correlations between actual and generated switch trials were high (rs >0.80 across the 4 conditions and 10 datasets), demonstrating that the model can reproduce a basic qualitative feature of participants' choice behavior.

