## [Decision Letter]

**Acceptance summary:**

This paper examines the relationship between psychopathology and decision-making adaptation to changes in contingency volatility in the environment. Previous work from the senior author has shown that individuals with high trait anxiety were less able to appropriately adjust their learning rate to the level of volatility in the context of aversive learning. The current paper extends this previous work in several ways: first, it takes a transdiagnostic approach to symptoms across both anxiety and depression; second, it conducts a very comprehensive model comparison to select the most appropriate behavioral model; third, it includes both aversive and appetitive learning, to test for generality vs. domain specificity of any finding; finally, it also compares the effects of positive and negative prediction errors across the reward and punishment domains. The authors report impaired adaptation of learning rate – in both the reward and punishment domains – as a function on a general factor of internalizing psychopathology, rather than anxiety specifically.

**Decision letter after peer review:**

Thank you for submitting your article "Impaired adaptation of learning to contingency volatility as a transdiagnostic marker of internalizing psychopathology" for consideration by *eLife*. Your article has been reviewed by three peer reviewers, and the evaluation has been overseen by a Reviewing Editor, Dr Shackman, and Senior Editor, Joshua Gold. The following individuals involved in review of your submission have agreed to reveal their identity: Argyris Stringaris and Samuel J Gershman.

The Reviewing Editor has drafted this decision to help you prepare a revised submission.

Summary

This paper examines relations between psychopathology and decision-making adaptation to changes in contingency volatility in the environment. Previous work from the senior author has shown that individuals with high trait anxiety were less able to appropriately adjust their learning rate to the level of volatility in the context of aversive learning. The current paper extends this previous work in several ways: first, it takes a transdiagnostic approach to symptoms across both anxiety and depression; second, it conducts a very comprehensive model comparison to select the most appropriate behavioral model; third, it includes both aversive and appetitive learning, to test for generality vs. domain specificity of any finding; finally, it also compares the effects of positive and negative prediction errors across the reward and punishment domains. The authors report impaired adaptation of learning rate – in both the reward and punishment domains – as a function on a general factor of internalizing psychopathology, rather than anxiety specifically.

The reviewers and I found several strengths to the report:

An admirable paper. The topic is of great interest, the care to replicate is refreshing to see, and the modeling is meticulous. We think it could be an important contribution to the field.

Overall, we were impressed with this study. The modeling and analysis was carried out carefully, and is comprehensive almost to the point that I worry the main ideas are obscured by layers of model variations (though I think it's okay as is). The new results add nuance to the emerging literature on computational modeling of anxiety and depression. They highlight the need for a transdiagnostic understanding of these disorders with respect to learning in volatile environments. We also appreciated that the authors made an effort to establish the generality of their findings across multiple populations and task variants, with relatively large sample sizes.

This is a very interesting and well written paper. It has several strengths, including a hierarchical Bayesian approach to the model fitting, careful model comparisons, and simulations to validate the winning model, as well as the use of both reward and punishment tasks, which is quite rare in this kind of studies. Results from a laboratory experiment were also largely replicated in an online study.

This will be a nice addition to the literature on learning under uncertainty and how such learning is affected in psychopathology.

Major/general suggestions

1) Bifactor Modeling. Noted by several reviewers.

a) More needs to be done in terms of ensuring that the modeling of the clinical data is appropriate and to demonstrate that the authors are in a position to draw some of the inferences they are drawing.

b) A lot in this paper hinges on the appropriateness of the bifactor modeling. I will try to explain. The paper is an extension of previous work that the authors have done in anxiety. The main innovation here is the extension to include a) depression; b) a negative affectivity or general factor resulting from the bifactor modeling. The latter, broadly, refers to a model of (typically) clinical symptoms where their common variance is accounted for by a general factor, and other latent factors (in this case anxiety and depression) retain their unique variance (and are thus uncorrelated to each other). Such models have similarities to other hierarchical symptom models. They have enjoyed a recent resurgence for a variety of reasons. The problem with these models is that they are pretty flexible and can easily be mis-specified. Given how much in this paper depends on the right choice of model, I would encourage the authors to do more to demonstrate that a bivariate model is indeed appropriate and superior to more parsimonious alternative. In particular, I am concerned about two things: first, the loadings for the general factor are relatively low, which is an indication that it may not really be a superordinate factor at all. Second, I am concerned about the fact that the authors do not seem to be comparing their model to a simpler structure (apologies if I have missed this) or alternative specifications (e.g. a super-ordinate model). I strongly suggest they do so and use appropriate metrics, such as the NLM. A good overview of the problems with inference and interpretation of such models is by Markon et al., Ann Rev Clin Psych 2019 and references therein, particularly Eid et al., 2017. Editor: See also recent simulation studies by Irwin Waldman's group and the HiTOP consortium (Roman Kotov)

c) Likewise, another reviewer wondered, "Is the bifactor analysis necessary? What happens is a simple factor analysis, or PCA, are used? Are the first three eigenvectors similar to the ones assumed in the bifactor analysis?"

2) Power Analysis. It would be useful if the authors gave an indication of how much power they would have in order to detect meaningful effect sizes for some of the two- and particularly three-way interactions in the paper. It may be appropriate to highlight this as a limitation of the study.

3) Model Validation. Given the concerns that have been expressed about the problems with LOO (being unrepresentative and often anti correlated with the training dataset, see Poldrack, 2019 JAMA Psych), the authors should either provide a rationale for selecting LOO or recompute using k-folds.

4) Differences Across Experiments. How do the authors explain the discrepancy between the two experiments with respect to the block type (volatile vs. stable) effect? Is it possible that the populations differed in their factor scores? From eye-balling Figure 1, this explanation seems implausible, but it would be useful to address this question statistically.

---

## [Author Response]

Major/general suggestions1) Bifactor Modeling. Noted by several reviewers.[…] In particular, I am concerned about two things: first, the loadings for the general factor are relatively low, which is an indication that it may not really be a superordinate factor at all.

During the development and refinement of standardized questionnaire measures of psychopathology such as those used here (e.g. STAI, BDI, MASQ, etc.), reliability and validity are typically assessed at the level of the scale or occasionally subscale (see Bieling et al., 1998; additional references added below). Individual item responses tend to be noisy, and aggregated scores at the scale-level effectively average out much of this noise. Given this, we reasoned that we should not be overly concerned about the nominal magnitude of loadings on individual items, if the factor scores derived from the loadings explain substantial variance in questionnaire (sub)scale summary scores and if the correlations between factor scores and questionnaire (sub)scale summary scores support factor content validity (i.e. point to the general factor capturing variance across depression and anxiety measures and point to the anxiety specific and depression specific factors capturing additional variance unique to anxiety or depression measures, respectively).

To examine this, we correlated the scores on the general and two specific factors extracted from the bifactor analysis with the scale / sub-scale scores that are typically calculated for each of the questionnaires administered (e.g. scores on the MASQ anhedonia and anxious arousal subscales, etc.). Although the bifactor analysis and these scores are based on the same items, the bifactor analysis was fit without specifying which items belonged to which questionnaires. Therefore, correlations between scores on the factors extracted from the bifactor analysis and scores on the questionnaire (sub)scales can be used to assess content validity of the extracted factors independently and to determine how much variance these factors can explain at the scale / subscale level. The correlations between factor scores and questionnaire scores has been added to Figure 1 as a new panel c.

In Figure 1C, it can be seen the general factor scores correlate strongly with all of the scale and subscale scores (r>0.6). This indicates that although the individual item loadings were moderately sized, the general factor scores derived from them explain a substantial amount of variance at the scale-level. Moreover, the general factor explains variance across questionnaires that are commonly considered depression-specific (e.g. BDI, CESD) and those considered anxiety-specific (e.g. the PSWQ and MASQ-AA, i.e. the anxious arousal subscale). These correlations provide strong evidence that there is indeed a common (superordinate) factor in these data.

A second important observation from Figure 1C is that the two specific factors each correlate strongly with only one or two specific scales or subscales. The depression-specific factor scores correlate strongly with the MASQ anhedonia subscale (r=0.74; MASQ-AD), and moderately with the STAI depression subscale (r=0.5, STAIdep). The anxiety-specific factor scores correlate strongly with the PSWQ (r=0.76). This confirms that specific factors are indeed selective to specific symptoms, as intended, and further suggests that they explain a good amount of variance that is not captured by the general factor (given the specific factors are orthogonal to the general factor).

We have added the corresponding text to the manuscript.

“As an additional check of construct validity, participants’ scores on these factors were correlated with summary (scale or subscale) scores for the standardized questionnaires administered, Figure 1C. […] The latter conclusion can be drawn since the specific factors are orthogonal to the general factor and therefore their correlations with scale and subscale scores reflect independently explained variance.”

We also conducted a supplementary analysis in which we regressed participants’ general factor scores against these scale and subscale scores and calculated residual scores for each (sub)scale. We then correlated participants’ residual scores for each scale with their original (non-residualized) scale scores. The extent to which the correlation drops beneath ‘1’ gives a measure of variance (i.e. the square root of this correlation difference) explained by the general factor (see new Figure 1—figure supplement 2). The two subscales with the highest amount of residual variance are the MASQ-AD and the PSWQ. We then considered how much of this residual variance can be explained by the specific factors from the bifactor model. It turns out that almost all of it can, as intended, be explained by the depressive and anxiety specific factors, respectively, r=0.95, r=0.96. This shows that much of the important variance not explained by the general factor is captured by the two specific factors.

We have added the corresponding text and figure supplement to the manuscript:

“This can be further demonstrated by regressing variance explained by scores on the general factor out of scale and subscale scores and then examining the relationship between residual scores for each scale with scores on the two specific factors. As shown in Figure 1—figure supplement 2, after removing variance explained by general factor scores, nearly all the remaining variance in PSWQ scores could be captured by the anxiety specific factor and nearly all the remaining variance in MASQ-AD scores could be captured by scores on the depression specific factor.”

Together, these analyses provide evidence that the general factor and the two-specific factors account for a large amount of shared and unique variance at the scale- and subscale-level, and moreover that they are valid constructs, acting as intended, to separate shared from unique variance in standardized measures of anxiety and depression.

Additional related modeling of task performance.

In the revised version of the manuscript, we include additional modeling of task performance using the residual scores on the MASQ-AD and either the PSWQ or MASQ-AA. This is presented in Appendix 6: Alternate modeling of anxiety- and depression-specific effects upon behavior. We explain this additional modeling here.

Given that residual scores on the MASQ-AD and PSWQ (after regressing out variance explained by scores on the general factor) are so strongly related to the two specific factors, we can use these residual scores in place of the specific factors in the hierarchical model of participants’ behavior on the volatility task. This has the didactic advantage of including well-known, standardized measures, only with variance explained by a general factor residualized out. In addition, here we are using indices that retain (or explain) 40-50% of the variance from the original measures.

As expected, a hierarchical model of task performance including participants’ scores for the general factor and residual scores for the MASQ-AD and PSWQ entered as population-level covariates produced extremely similar results to those described in the main text using the three factors obtained from the bifactor model. In Appendix 6—figure 1, we show that the modulatory effect of general factor scores on effects of volatility, relative outcome value, and volatility x relative outcome value on learning rate, as reported in the main manuscript (here shown by the black data points), were also statistically significant in this new model (as shown by the green data points). Neither residual scores on the MASQ-AD or PSWQ significantly modulated learning rate , i.e. the 95%-HDI contained zero for the main effects and two-way interactions, Appendix 6—figure 1 (green data points). We note this analysis was conducted for experiment 1 only as the PSWQ was not administered to participants in experiment 2.

A similar approach can also be used to examine the relationship between scores on the MASQ-AA subscale and task performance. Regressing out scores on the general factor also left substantial residual variance in MASQ-AA scores (see Figure 1—figure supplement 2, panel B). This is interesting, as the MASQ-AA subscale may capture a different, potentially more somatic, and less cognitive, subset of anxiety-specific symptoms to those picked up by the PSWQ (Bijsterbosch et al., 2014). Notably, the MASQ-AA subscale was also the only measure included for which residual scores (after accounting for scores on the general factor) did not show positive moderate or high correlations with one of the two specific factors. In light of this, we also fit an alternate hierarchical model to participants’ behavior that included the general factor along with residual scores for the MASQ-AA and the MASQ-AD. The learning rate results for this model, when fit to the experiment 1 dataset, are also presented in Appendix 6—figure 1. In addition, we show the results for this model when fit to the experiment 2 dataset in Appendix 6—figure 2. In both cases, modulation by general factor scores of adaptation of learning rate to volatility, especially following outcomes of positive relative value, continue to be observed (blue data points). Residual scores from the MASQ-AA (and MASQ-AD in context of these other population-level parameters) did not significantly influence adaptation of learning rate to volatility or the effect on learning rate of the interaction of block type (volatile, stable) by relative outcome value or task version. There was also no relationship between MASQ-AA or MASQ-AD residual scores and differential learning as a function of task type (aversive versus reward gain in experiment 1 and reward loss versus reward gain in experiment 2). In experiment 2 only, there was an effect of residual scores from the MASQ-AA on the overall extent of learning following outcomes of positive versus negative relative value.

We hope these additional analyses might be of interest to readers who are used to primarily administering just one or two questionnaire measures.

“Appendix 6: Alternate modeling of anxiety- and depression-specific effects upon behavior.

As detailed in the main manuscript, the depression-specific factor was strongly correlated with residual scores on the MASQ-AD (after regressing out variance explained by scores on the general factor) and the anxiety-specific factor was strongly correlated with residual scores on the PSWQ (created in an equivalent fashion), see Figure 1—figure supplement 2). […] However, no equivalent relationship was observed in experiment 1 (Appendix 6—figure 1). We hence caution against weight being given to this observation prior to any replication of this result.”

Second, I am concerned about the fact that the authors do not seem to be comparing their model to a simpler structure (apologies if I have missed this) or alternative specifications (e.g. a super-ordinate model). I strongly suggest they do so and use appropriate metrics, such as the NLM. A good overview of the problems with inference and interpretation of such models is by Markon et al., Ann Rev Clin Psych 2019 and references therein, particularly Eid et al., 2017. Editor: See also recent simulation studies by Irwin Waldman's group and the HiTOP consortium (Roman Kotov)

We thank you for raising this concern. Here, our response is in two parts—a theoretical consideration of the basis of model selection and the provision of additional models as requested.

The problems raised by Markon, 2019 and Eid et al., 2017, are very important and speak to the broader, but obviously difficult question of “what is the right basis on which to select a model?” Although we don’t think that a definitive answer to this question can be given, we do believe that the answer strongly depends on the scientific question and the theory on which that question is based. Our scientific question is largely based on the literature arguing that anxiety and depression symptoms should be conceptually separated into shared and unique variance. Given this premise, our main criterion for choosing a factor model was whether it could partition variance in a way that aligned with previous theory. In this way, we viewed the selection of a factor model for our experiment akin to the selection of a good set of regressors, in which interpretability is key. Bifactor analyses seems to be the most well supported analytic tool for partitioning the variance in the desired way (Reise, 2012; Markon, 2019), despite its potential limitations (Eid et al., 2017). Nevertheless, the fit to data is obviously important, and indeed we tried to assess this using confirmatory factor analysis on an independent dataset.

With these considerations in mind, we have edited the manuscript text to explain our choice of model better and have also detailed the fit of three alternative factor models within the Supplementary Materials.

Before turning to the additional models, we would like to address this point:

3) I strongly suggest they do so and use appropriate metrics, such as the NLM.

We thank the reviewer for pointing us to Normalized Maximum Likelihood (NML). We also find it very appealing from a theoretical standpoint. However, NML requires summing over all possible datasets and it is not clear how to construct a suitable set of all possible datasets for factor analysis. The references in Markon, 2019, discuss NML on a more theoretical level giving practical implementations for only certain types of models (e.g. logistic regression in Rissanen, 2007), but not factor analysis. We were not able to find applications of NML in a practical case like ours.

Additional models

In line with the request to consider a simpler structure, we fit a “flat” correlated two factor model. Here, one factor loaded predominantly on anxiety items and another loaded predominantly on depression items. The two factors were strongly correlated (r=0.58). This model is actually an intermediate step in the bifactor analysis, which then proceeds to extract a general factor from these two correlated factors by applying the Schmid-Leiman transformation. Since the Schmid-Leiman transformation shifts the variance partitioning rather than explaining additional variance, both models actually fit identically in the in-lab dataset (n=86) on which the factor models were estimated. (Reise, 2012, as cited by Markon, 2019, actually provides a beautiful account of the equivalence of different models, noting how the correlated k-factor model and the general plus k-specific factors bifactor model are re-parameterizations of one-another). Re-estimating both models in the confirmatory sample (n=199) showed that the correlated factors model had a slightly worse fit (RMSEA=0.083) than the bifactor model (RMSEA=0.065).

We also consider a superordinate (or higher order) factor model, as requested. This model differs from the bifactor model in that the general factor cannot directly explain correlations in individual-items, but rather explains them indirectly through the correlations between the lower level factors. We fit this model to the confirmatory data set, using the two factors from the bifactor model but removing influences of the general factor directly onto individual items. This model fit less well (RMSEA=0.083) than the bifactor model (RMSEA=0.065).

We also considered a single factor model; here, given the relatively small amount of item variance explained by the specific models, we examined if the general factor alone (as estimated in the exploratory factor analysis) provided as good a fit to the CFA dataset. This model also fit less well when re-estimated in the confirmatory sample RMSEA=0.110. Furthermore, this model cannot be used to address our scientific question.

In light of the analyses just discussed, we made the following changes to the manuscript:

We’ve added a new section “Appendix 2: Additional Factor Analyses*”* describing these additional factor analyses. We also briefly summarize them in the main manuscript:

“As outlined earlier, we applied a bifactor model to item-level symptom responses as we sought to tease apart symptom variance common to anxiety and depression versus unique to anxiety or depression. With model selection, both the extent to which a given model can address the aim of the experiment and the fit of the given model to the data are important considerations. In addition to assessing the absolute fit of the bifactor solution in the confirmatory factor analysis (CFA) dataset, we can also consider its fit relative to that of alternate models. The bifactor model reported here showed a better fit to the novel (CFA) dataset than a “flat” correlated two-factor model, a hierarchical three factor model with the higher order factor constrained to act via the lower level factors, and a unifactor model created by retaining only the general factor and neither of the specific factors (see Appendix 2: Additional Factor Analyses for further details). We note that none of these alternate models would enable us to separate symptom variance common to anxiety and depression versus unique to anxiety and depression, as desired.”

c) Likewise, another reviewer wondered, "Is the bifactor analysis necessary? What happens is a simple factor analysis, or PCA, are used? Are the first three eigenvectors similar to the ones assumed in the bifactor analysis?"

We thank the reviewer for this suggestion. We describe the results for a “simple factor analysis” (the 2-factor correlated model) above. In further response to this query, we also conducted a PCA on the item-level questionnaire data in experiment 1 and examined the top 5 components (PC’s). Scores on the first PC were extremely similar to the general factor scores (r=0.9; see Author response image 1). However, scores on subsequent PCs cannot be used to cleanly disentangle depression and anxiety symptomatology as desired to address the study’s aims. Specifically, scores on the second PC correlated positively with worry (PSWQ r=0.59) and negatively with anhedonia (MASQAD r = -0.3). Scores on the third PC correlated with anxious arousal (MASQAA; r=0.5), but also moderately with two depression measures (BDI r=0.19 and CESD r=0.18). The fourth PC correlated moderately with one depression measure BDI (r=-0.25) but not the other CESD (r=-0.03), and therefore likely captured something idiosyncratic to that measure. The fifth PC did not correlate strongly with any measure, so we stopped there in extracting principal components.

As mentioned above, this PC structure would not enable us to disentangle variance common to both anxiety and depression versus variance unique to anxiety or depression. However, we can use PC1 to demonstrate the robustness of our general factor findings. Specifically, if we fit a model to experiment 1 task data, which includes participants’ scores on PC1 along with residual scores on the MASQAD and PSWQ after having regressed out variance explicable by PC1 we obtain parallel findings to those obtained using the general factor and either the two specific factors derived from the bifactor analysis or residual scores on the MASQAD and PSWQ obtained after regressing out variance explicable by scores on the general factor. This again provides evidence that the results we obtained did not hinge on the exact specification of the bifactor model.

**Author response image 1. sa2fig1:** Information captured by top five PCs: We conducted PCA on the item-level questionnaire responses from experiment 1 (n=86). Scores on the first PC correlated highly with general factor scores (r=0.9). Scores on the second PC correlated strongly positively with PSWQ (r=0.59), and moderately negatively with MASQAD (r=-0.3),. The third PC correlated most strongly with MASQAA (r=0.5), but also correlated moderately with the BDI and CESD (r=0.19, r=0.18). The fourth PC correlated moderately with BDI (r=-0.25) but not with CESD (r=-0.03), perhaps capturing something specific to the BDI. The fifth PC did not correlate strongly with any subscale. Correlations are shown for the experiment 1 dataset only, on which the PCA and correlated factors model were estimated.

2) Power Analysis. It would be useful if the authors gave an indication of how much power they would have in order to detect meaningful effect sizes for some of the two- and particularly three-way interactions in the paper. It may be appropriate to highlight this as a limitation of the study.

We thank the reviewers for raising this important issue regarding our power to detect two-way and particularly three-way interactions.

There is clear evidence that the study was amply powered to detect two-way interactions. Firstly, we actually observed significant two-way interactions for learning rate (good-bad)x(volatile-stable) in both experiment 1 and experiment 2. Secondly, our parameter recovery simulations, which used the same sample size as our real data, were able to recover two-way interactions. Such recovery simulations can be seen as ways of estimating power in the very practical sense of “the ability to detect effects, assumed to be present, if a new dataset were to be collected”. That is, we assume the effects we observed were true, we simulate new data from these effects, and then we see how well the effects can be recovered (re-detected). Good recovery then speaks to good power. For parameter recovery of effects representing two-way interactions for learning rate see Appendix 4—figure 1.

To assess our power to detect three-way interactions, we performed two additional analyses.

The first analysis applied the type of parameter recovery procedure just discussed to the model that included the three-way interaction. We’ve included the results of this analysis as a new Appendix 4—figure 5. In the figure, it can be seen that the three-way interaction term (bottom middle panel) has similar parameter recoverability to the other learning rate effects (other panels).

In addition, we conducted a second permutation-based analysis that used a synthesis procedure to show that the absence of a three way interaction was unlikely to be because of a lack of power. To explain: in the manuscript, we report that scores on the general factor modulate the effect of volatility by relative outcome value on learning rate but that this does not vary as a function of task version (a paradigmatic absent three-way interaction). As an additional check to see if we would have been able to determine if scores on the general factor modulated the effect of volatility by relative outcome value by task version on learning rate, we conducted the following analysis. We fit the model with the three-way interaction of volatility (volatile-stable) by relative outcome (good – bad) by task version (reward, aversive) included to participants’ data from experiment 1 (see Permutation Analysis in Author response image 2 panel A). This shows the influence of scores on the general factor on the 2-way interaction of volatility x relative outcome value and the absence of any influence of general factor scores on the 3-way interaction. We then permuted participants’ posterior learning rate values from just the aversive task to randomize performance across conditions and participants for this version of the task. We next used participants’ posterior means for each learning rate component (actual for the reward version of the volatility task, and the permuted values for the aversive version of the volatility task) to generate new data for each participant; all other parameters were kept the same. We then refit the model to this simulated data and re-estimated individual and population-level parameters. By eradicating between and within subjects effects upon learning rate in the aversive version of the task in this manner, while leaving them intact in the reward version of the task, we should, if sufficiently powered, now observe that general factor scores significantly modulates the effect of volatility by relative outcome value by task version upon learning rate. This is indeed the case, βg=-0.16, 95%-HDI = [-0.32, -0.02], see Permutation Analysis in Author response image 2 panel B (the three-way interaction effect is circled on the far right side of the figure). This shows that if there had been a three-way interaction in our dataset—for example a difference across tasks in the two-way interaction that we actually observed—we would likely have had the power to detect it.

Actual and recovered posterior mean learning rates for each participant in each condition are plotted in panels C to F-f); small dots are individual participants and large dots are group-level means. Panel(c) shows the non-permuted posterior means for learning rate as a function of block type (volatile, stable) and relative outcome value (good, bad) for the reward version of the task. Panel (D shows the permuted posterior means for the aversive version of the task. Panels E and F show the corresponding recovered parameter values. It can be seen that by permuting posterior means for learning rate in the aversive version of the task, we wipe out the boost to learning rate after good outcomes under volatile conditions shown by participants with low scores on the general factor (see Figure 4B). Parameter recovery successfully reveals the presence of this effect in the reward version of the volatility task and its (manipulated) absence in the aversive version of the volatility task.

In view of the complexity and synthetic nature of this result, we elected not to include it in the paper.

We reference the results of the parameter recovery analysis in the Results section.

“We note that a parameter recovery analysis revealed successful recovery of the parameter representing the three-way interaction of block type, relative outcome value and task type on learning rate (Appendix 4—figure 5). This suggests that we did not simply fail to observe a three-way interaction due to lack of experimental power.”

3) Model Validation. Given the concerns that have been expressed about the problems with LOO (being unrepresentative and often anti correlated with the training dataset, see Poldrack, 2019 JAMA Psych), the authors should either provide a rationale for selecting LOO or recompute using k-folds.

We thank the reviewer for pointing out this interesting potential issue with leave-one-out cross validation raised by Poldrack, 2019.

First, we wanted to clarify that we do not calculate exact leave-one-out cross validation. It would have been too computationally expensive to refit all of our models that many times. It would have also been very computationally expensive to do exact k-fold cross validation. Instead, we chose to approximately calculate leave-one-out cross validation using Pareto smoothed importance sampling (PSIS; Vehtari et al., 2017). This was likely confusing because we simply abbreviated our approach as LOO, which is often done (as is enshrined in the name of the extremely popular R package for calculating this quantity). We now specify this more clearly in the manuscript and have changed our abbreviation to PSIS-LOO to avoid confusion.

We chose to use PSIS-LOO, because it has become the accepted standard for model comparison in hierarchical Bayesian model, which are often too large for exact cross-validation (Gelman et al., 2015). Vehtari et al., 2017 provides nice simulations showing that PSIS-LOO approximates the exact methods well in many situations.

We also considered another metric, WAIC, which is also commonly used to compare models in the hierarchical Bayesian setting. This metric penalizes overfitting in a different way than PSIS-LOO. Importantly, the model rankings obtained by PSIS-LOO and WAIC were identical.

Finally, we note that since our study was moderately well powered (n=86 in experiment 1, n=199 in experiment 2) we were also less concerned about the issue raised by Poldrack about leave-one-out cross-validation. His concern relates to the fact that leaving out only a single data point could induce negative correlations between the test and training set. While this could be an issue in some cases, we believe that it is likely to be more of a problem in very small datasets, in which a single observation could have a sizeable influence on its own.

We’ve added the following text to our manuscript to further justify our use of PSIS-LOO and to mention that our results hold using WAIC.

“We compared the fits of different models using Pareto smoothed importance sampling to approximate leave-one-out cross validation accuracy (PSIS-LOO; Vehtari et al., 2017). PSIS-LOO is a popular method with which to estimate out of sample prediction accuracy as it is less computationally expensive than evaluating exact leave-one-out or k-fold cross validation accuracy (Gelman et al., 2013). We note that comparing models using WAIC (Watanabe and Opper, 2010), an alternative penalization-based criterion for hierarchical Bayesian models, resulted in identical model rankings for our dataset.”

4) Differences Across Experiments. How do the authors explain the discrepancy between the two experiments with respect to the block type (volatile vs. stable) effect? Is it possible that the populations differed in their factor scores? From eye-balling Figure 1, this explanation seems implausible, but it would be useful to address this question statistically.

We thank the reviewer for this very interesting point. Neither of the specific factors were associated with differential effects of block type on learning. However, if participants in experiment 2 had higher general factor scores, on average, than participants in experiment 1, then since higher scores on the general factor are associated with smaller effects of volatility on learning, such a difference could indeed potentially explain the presence of a group level effect of block type in experiment 1 but not experiment 2. However, comparing the general factor scores across the two samples, we found that experiment 1’s participants had statistically higher general factor scores than experiment 2’s participants (mean difference 0.38; t=3.0 p=0.002). This is the opposite direction to what would be needed to explain the presence of a group-level effect of block type (volatility) in experiment 1 but not in experiment 2.

We do not have a strong alternate explanation for this discrepancy, and hence we note it but do not speculate as to its cause.

**References:**

Bieling, P. J., Antony, M. M., and Swinson, R. P. (1998). The State-Trait Anxiety Inventory, Trait version: structure and content re-examined. *Behaviour research and therapy*, *36*(7-8), 777-788.

Bijsterbosch, J., Smith, S., Forster, S., John, O. P., and Bishop, S. J. (2014). Resting state correlates of subdimensions of anxious affect. *Journal of Cognitive Neuroscience*, *26*(4), 914-926.

Cuijpers, P., Gentili, C., Banos, R. M., Garcia-Campayo, J., Botella, C., and Cristea, I. A. (2016). Relative effects of cognitive and behavioral therapies on generalized anxiety disorder, social anxiety disorder and panic disorder: A meta-analysis. *Journal of Anxiety Disorders*, *43*, 79-89.

Gelman, A., Carlin, J. B., Stern, H. S., Dunson, D. B., Vehtari, A., and Rubin, D. B. (2013). *Bayesian data analysis*. CRC press.

Knowles, K. A., and Olatunji, B. O. (2020). Specificity of trait anxiety in anxiety and depression: Meta-analysis of the State-Trait Anxiety Inventory. *Clinical Psychology Review*, 101928.

Reise, S. P. (2012). The rediscovery of bifactor measurement models. *Multivariate behavioral research*, *47*(5), 667-696.

von Glischinski, M., von Brachel, R., and Hirschfeld, G. (2019). How depressed is “depressed”? A systematic review and diagnostic meta-analysis of optimal cut points for the Beck Depression Inventory revised (BDI-II). *Quality of Life Research*, *28*(5), 1111-1118.